# Optical blood-brain-tumor barrier modulation expands therapeutic options for glioblastoma treatment

Qi Cai [1], Xiaoqing Li[2], Hejian Xiong[1], Hanwen Fan[1], Xiaofei Gao[3], Vamsidhara Vemireddy[3,4], Ryan Margolis[2], Junjie Li[2], Xiaoqian Ge[1], Monica Giannotta[5,9], Kenneth Hoyt[2], Elizabeth Maher[3,4,6], Robert Bachoo[3,4,6] ✉ & Zhenpeng Qin [1,2,7,8] ✉

The treatment of glioblastoma has limited clinical progress over the past decade, partly due to the lack of effective drug delivery strategies across the blood-brain-tumor barrier. Moreover, discrepancies between preclinical and clinical outcomes demand a reliable translational platform that can precisely recapitulate the characteristics of human glioblastoma. Here we analyze the intratumoral blood-brain-tumor barrier heterogeneity in human glioblastoma and characterize two genetically engineered models in female mice that recapitulate two important glioma phenotypes, including the diffusely infiltrative tumor margin and angiogenic core. We show that pulsed laser excitation of vascular-targeted gold nanoparticles non-invasively and reversibly modulates the blood-brain-tumor barrier permeability (optoBBTB) and enhances the delivery of paclitaxel in these two models. The treatment reduces the tumor volume by 6 and 2.4-fold and prolongs the survival by 50% and 33%, respectively. Since paclitaxel does not penetrate the blood-brain-tumor barrier and is abandoned for glioblastoma treatment following its failure in early-phase clinical trials, our results raise the possibility of reevaluating a number of potent anticancer drugs by combining them with strategies to increase blood-brain-tumor barrier permeability. Our study reveals that optoBBTB significantly improves therapeutic delivery and has the potential to facilitate future drug evaluation for cancers in the central nervous system.

World Health Organization (WHO) grade IV astrocytoma (Isocitrate Dehydrogenase, IDH, wild-type), known as glioblastoma (GBM), is the most common and aggressive primary brain tumor[1–3]. Despite standard-of-care treatment that includes maximal safe resection of the contrast-enhancing regions of T1-weighted Magnetic Resonance Imaging (MRI), fractionated radiation to 60 Gy with concurrent daily temozolomide (TMZ) followed by up to 6 months of adjuvant TMZ, the median overall survival for GBM patients remains abysmal at

[1]Department of Mechanical Engineering, the University of Texas at Dallas, Richardson, TX 75080, USA. [2]Department of Bioengineering, the University of Texas at Dallas, Richardson, TX 75080, USA. [3]Department of Internal Medicine, University of Texas Southwestern Medical Center, Dallas, TX 75390, USA. [4]Harold C. Simmons Comprehensive Cancer Center, University of Texas Southwestern Medical Center, Dallas, TX 75390, USA. [5]IFOM ETS – The AIRC Institute of Molecular Oncology, 20139 Milan, Italy. [6]Department of Neurology, University of Texas Southwestern Medical Center, Dallas, TX 75390, USA. [7]Department of Biomedical Engineering, University of Texas Southwestern Medical Center, Dallas, TX 75390, USA. [8]Center for Advanced Pain Studies, the University of Texas at Dallas, Richardson, TX 75080, USA. [9]Present address: Division of Immunology, Transplantation and Infectious Diseases, IRCCS San Raffaele Scientific Institute, 20132 Milan, Italy. ✉e-mail: robert.bachoo@utsouthwestern.edu; zhenpeng.qin@utdallas.edu

15 months[4]. One obstacle in conventional therapies is the inability to achieve adequate drug concentrations in the brain due to the protective blood-brain barrier (BBB). Formed by a tight-junction (TJ) protein complex and adherens junctions between the brain microvascular endothelial cells and modulated by surrounding stromal cells (pericytes and astrocytes), the BBB excludes or limits the delivery of 98% of conventional chemotherapies to subtherapeutic levels[5]. Although GBM can disrupt the integrity of the BBB in the hypoxic and angiogenic core, the magnitude of this local disruption is nonuniform or insufficient to allow drug penetration in meaningful quantities[6–10]. Moreover, evidence suggests that GBM has tumor cells infiltrating into the neighboring tissue without disrupting the BBB, which subsequently drives the inevitable fatal recurrence[11]. Therefore, we need strategies to overcome the BBB, or the blood-brain-tumor barrier (BBTB), in both angiogenic core and infiltrative margins to achieve significant improvement in disease management and patient survival.

Several strategies to overcome BBTB for therapeutic delivery have been developed, including using hyperosmotic agents (mannitol), opening the TJ with a TJ modulator, and enhancing drug penetration through inhibiting drug efflux transporters or via receptor-mediated transport[6,11–13]. While these strategies may improve drug delivery to brain tumors, lack of spatial resolution, high incidences of complications, and potential for toxicity have impeded progress in clinical translation[12,13]. Focused ultrasound (FUS) combined with intravenously (i.v.) administered microbubbles is a local, minimally invasive method for transiently disrupting the BBTB and has progressed to early-phase clinical trials[14–16]. Recently, we demonstrated an optical method to increase BBB permeability reversibly[17]. Nevertheless, the poor survival with currently approved treatments for GBM and the failure of many promising results at the clinical trial stage highlight the compelling need to develop and validate treatment strategies with clinically relevant GBM models to bridge the gap between preclinical efficacy and successful clinical translation[18–20].

In this work, we report a GBM treatment approach by BBTB modulation followed by chemotherapy in clinically relevant infiltrative and angiogenic GBM models. We first provide evidence that human GBM shows intratumoral heterogeneous BBTB permeability, including both leaky and intact BBTB regions. To capture these features in preclinical mouse models, we characterize two genetically engineered mouse models (GEMMs) that show diffuse single-cell infiltration through the brain parenchyma (former, PS5A1) or a rapidly expanding angiogenic mass with limited single-cell infiltration (the latter, 73 C), respectively. These primary conditional mouse cell lines carry mutations seen in both adult and pediatric high-grade gliomas (namely, (1) Braf$^{V600E}$, INK4ab/Arf$^{-/-}$, PTEN$^{-/-}$, for PS5A1 GEMM, and (2) Braf$^{V600E}$, P53$^{-/-}$, PTEN$^{-/-}$, for 73 C GEMM). Together these two GEMMs represent a reasonable facsimile of the GBM tumor-stromal phenotype seen in the clinical setting. We subsequently apply pulsed laser stimulation of tight junction-targeted gold nanoparticles (AuNPs) to reversibly modulate the BBTB permeability (optoBBTB) in the GEMMs and show the brain delivery of an oncology drug paclitaxel (Taxol). Taxol is among the most widely used oncology drug because of its proven efficacy in multiple cancer subtypes, but it is abandoned for GBM treatment following its failure in early-phase clinical trials due to poor brain penetration[21–24]. Moreover, although several highly specific Braf$^{V600E}$ inhibitors have shown to be effective for treating melanoma[25], these drugs are only transiently effective with tumors (including melanoma brain metastasis) rapidly developing resistance to Braf$^{V600E}$ inhibitors[26]. Therefore, the consideration of Taxol delivery to our Braf$^{V600E}$ models is highly relevant, since Braf$^{V600E}$ inhibitor clinical trials for brain tumors are ongoing[27,28]. This study reveals that repeated cycles of optoBBTB coupled with systemic administration of Taxol suppress tumor growth (6 and 2.4- fold) by reducing tumor cell proliferation and increasing cell death, resulting in significantly improved median survival (50% and 33% increase) in the infiltrative (PS5A1) and angiogenic (73 C) models, respectively. Our investigations provide evidence of BBTB modulation and therapeutic benefits using optoBBTB in clinically relevant models.

## Results

### Characterization of diffusely infiltrative PS5A1 GEMM and angiogenic 73 C GEMM

We examined a human GBM that shows intratumoral BBTB heterogeneity and recurrence. The patient was treated with standard of care for GBM, including surgical resection and concurrent radiation (60 Gy) and TMZ, followed by 12 monthly cycles of TMZ. At the end of treatment and for 4 years of serial MR imaging, there was no evidence of recurrence (Supplementary Fig. 1a). However, within 4 months after an unchanged MR scan, the patient developed focal seizures, and a repeat MRI showed a new enhancing mass at the medial tumor margin (Supplementary Fig. 1b). Biopsy of the mass revealed a classic GBM phenotype with microvascular proliferation and tumor proliferation rate (MIB-1) of 80% (Supplementary Fig. 1c, d). These results suggest that human GBM shows infiltrative characteristics and marginal recurrence with no initial contrast enhancement (therefore intact BBTB), indicating the clinical need to establish pre-clinical GEMMs that capture these hallmarks to assess the drug efficacy and therapeutic strategies accurately.

To recapitulate these features in preclinical models, we characterized two GEMMs in terms of their BBTB integrity, tumor progression patterns, and TJ properties. These GEMMs were generated using neural-stem-cell–derived PS5A1 (Braf$^{V600E}$, INK4ab/Arf$^{-/-}$, PTEN$^{-/-}$, Supplementary Fig. 2a, c) and astrocyte-derived glioma cell line 73 C (Braf$^{V600E}$, PTEN$^{-/-}$, P53$^{-/-}$, Supplementary Fig. 2b, c)[29,30]. These cell lines were engineered to express green fluorescent protein (GFP). We first established PS5A1 and 73 C GEMMs in female nude mice (Nu/J, 002019, age 7 weeks, the Jackson Laboratory) and evaluated their BBTB permeability. Specifically, 368 nL of PS5A1 glioma cell suspension or 92 nL of 73 C glioma cells ($2 \times 10^5$/μL) was constantly injected into the mouse cortex (−1 mm, −1 mm, 0.5 mm) using a nanoinjector equipped with a glass micropipette (50 μm tip, see methods for details). The BBTB integrity of the mice during GBM progression was analyzed using i.v. injection of EZ-link biotin (660 Da) and Evans blue (66 kDa, albumin-bound). Figure 1a and c show that in PS5A1 GEMM, the dye was confined in the blood vessels in both tumor core and margin at 14-, 28-, and 42-days post injection (dpi), indicating the intact BBTB. However, 73 C GEMM showed immature dysfunctional tumor-associated vessels during disease progression. At 7 dpi, both low and high molecular weight fluorescent dyes were limited to the microvascular lumen in the tumor core region and at the margins of the expanding mass that interface with the normal brain parenchyma, also at the contralateral hemisphere (Fig. 1b, d), suggesting an intact BBTB at 7 dpi. On the contrary, at 14 and 21 dpi, following the rapid expansion of the tumor mass, there was clear evidence of dye extravasation in the tumor core regions but absent at the tumor margins. These observations suggest that the tumor core region was perfused by a microvasculature with compromised BBTB integrity. In contrast, the tumor margin region was characterized by limited infiltration into the surrounding brain parenchyma, which remained stable and had intact BBTB at the sampling time.

We subsequently investigated the tumor progression patterns in the two GEMMs. The PS5A1 tumor cells displayed a heterogeneous pattern of brain infiltration, moving along brain capillaries, through the neuropil-rich gray matter, and parallel to myelinated axons along the white matter tracts (Fig. 1a, Supplementary Fig. 3a). There was no evidence of angiogenesis with this diffusely invasive GBM model since vascular density identified by both structure (endothelial marker CD31 positive cells) and perfusion (luminal wall labeling with tomato lectin594) were similar in the tumor core and margin regions compared with that of the contralateral hemisphere (Fig. 2a, b). This data

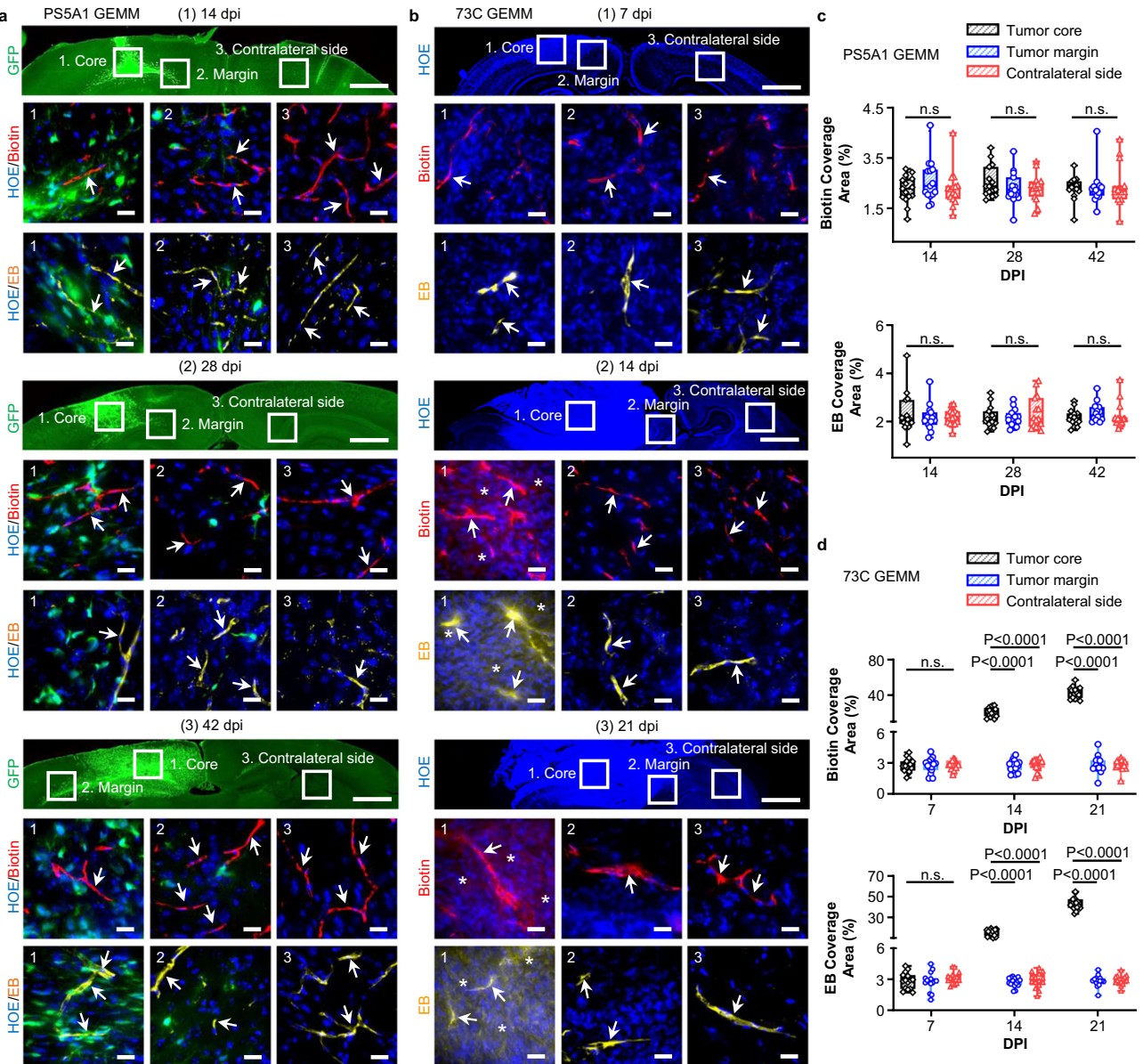

**Fig. 1 | PS5A1 GEMM has an intact BBTB, and 73 C GEMM shows heterogeneous loss of BBTB integrity during disease progression. a** Characterization of the BBTB permeability in PS5A1 GEMM using EZ-link biotin (Biotin, red, 660 Da) and Evans blue (EB, yellow, 66 kDa when bound to albumin) at 14-, 28-, and 42-days post injection (dpi). The tumor cells express GFP, and the cell nuclei are indicated by Hoechst staining (HOE, blue). The ROIs selected are (1) tumor core, (2) tumor margin, and (3) contralateral side with no tumor. The scale bars represent 1 mm in the top panel and 20 μm in the bottom panels. The blood vessels are indicated by arrows. **b** Characterization of the BBTB permeability in 73 C BBTB using EZ-link biotin (Biotin, red) and Evans blue (EB, yellow) at 7–21 dpi. The cell nuclei are indicated by Hoechst staining (HOE, blue). The ROIs selected are (1) tumor core, (2) tumor margin, and (3) contralateral side with no tumor. The blood vessels are indicated by arrows, and the dye leakage is indicated by asterisks. The scale bars represent 1 mm in the top panel and 20 μm in the middle and bottom panels. **c, d** The quantification of biotin and Evans blue coverage in PS5A1 and 73 C GEMMs by area fraction. Data are expressed as Mean ± SD. $N = 15$ images from 3 mice. Data in the box and whisker plots are given from the minima to maxima, the bounds of the box represent the 25th percentile and 75th percentile, and the middle line of the box is the median. Data were analyzed by One-way ANOVA followed by Tukey's multiple comparisons test. n.s. represents no significant difference. Source data are available as a Source Data file.

suggests that in PS5A1 GEMM, tumor growth and infiltration are promoted by co-opting the normal dense brain microvasculature for nutrient and metabolic support (Supplementary Fig. 3b).

It is well established that tumor core regions release proangiogenic signals, leading to the formation of immature and dysfunctional networks of blood vessels[31]. To verify our 73 C GEMM, we evaluated the immunostaining of junctional proteins, Claudin-5, ZO-1, VE-Cadherin, Occludin, and JAM-A with CD31-labeled endothelial cells. Notably, CD31-labeled microvascular density was significantly increased in the tumor core compared with the tumor margin and contralateral brain region (Supplementary Fig. 4a). In contrast, i.v. injection of tomato lectin594 to label perfused vessels showed a marked increase in the ratio of cell nuclei labeled with Hoechst dye (HOE) to the blood vessels (HOE/lectin) in the tumor compared with the contralateral brain (Supplementary Fig. 4b, top). However, there was no significant difference in this ratio with IHC staining of blood vessels using CD31 (HOE/CD31, Supplementary Fig. 4b, bottom). These results suggest that the perfused vessels (lectin labeled) in the 73 C tumor core need to support an increased number of tumor cells, which can lead to hypoxia and angiogenesis. Furthermore, the tumor core region contains a significant fraction of either nascent vessels that have yet to form a functional conduit and/or have formed non-functional end-vessels

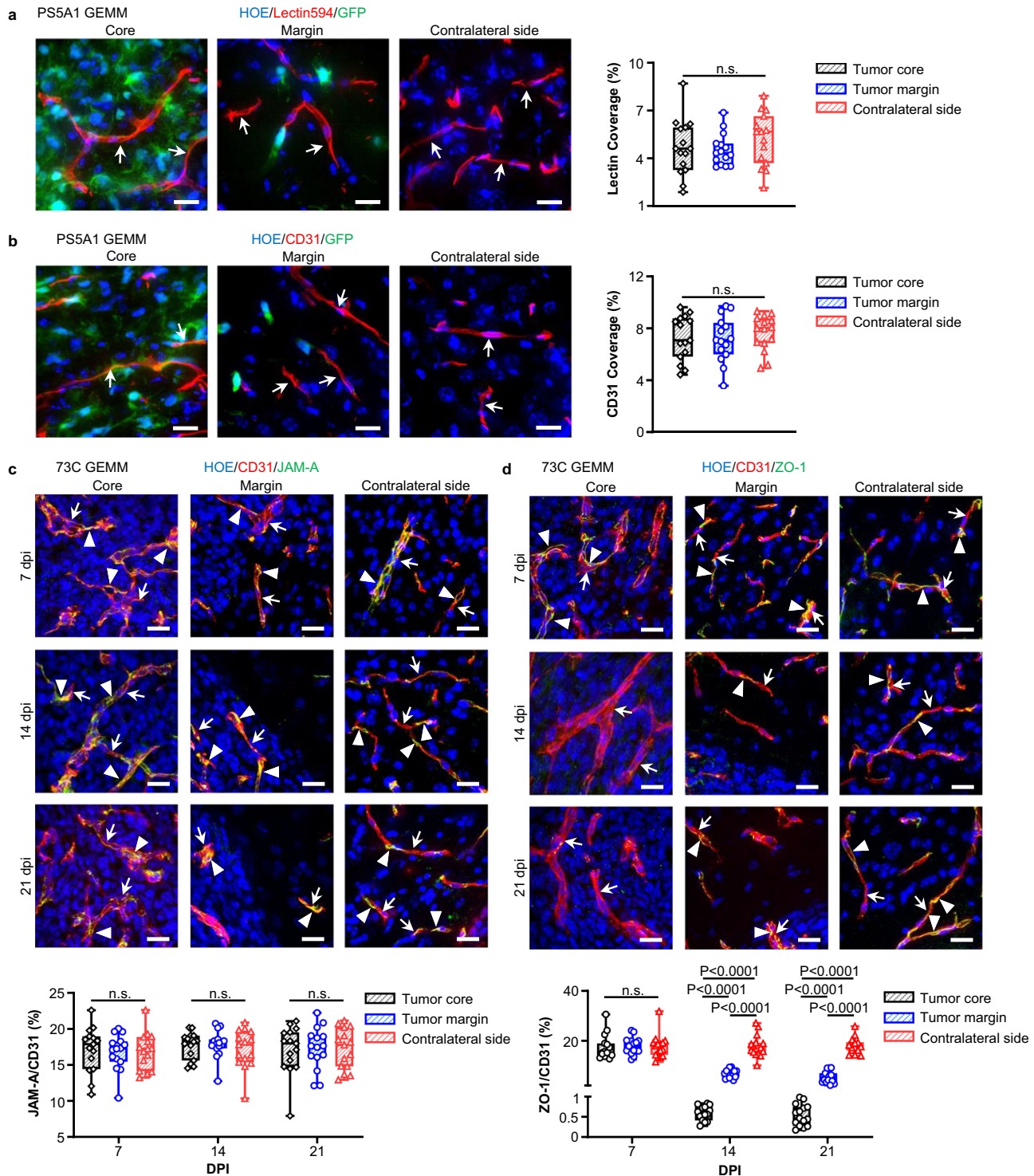

**Fig. 2 | PS5A1 GEMM shows diffuse infiltration and vascular co-option for tumor cells while 73 C GEMM shows vascular angiogenesis with loss of ZO-1 coverage.** **a** Blood vessel labeling with tomato lectin594 (indicated by arrows) in PS5A1 GEMM at 14 days post injection (dpi) and the quantification of lectin coverage by area fraction. The scale bars represent 20 μm. *N* = 15 images from 3 mice. **b** Blood vessel labeling with CD31 (indicated by arrows) in PS5A1 GEMM at 14 dpi and the quantification of CD31 coverage by area fraction. The scale bars represent 20 μm. *N* = 15 images from 3 mice. **c**, **d** IHC staining and quantification of junctional protein JAM-A and ZO-1 in 73 C GEMM at 7–21 dpi. The blood vessels are stained with CD31 (red),

and the cell nuclei are indicated by Hoechst staining (HOE, blue). The arrows indicate blood vessels, and the arrowheads indicate tight junction proteins. The scale bars represent 20 μm. The quantification of JAM-A and ZO-1 coverage on the blood vessel by area fraction. *N* = 15 images from 3 mice. In **a**–**d**, data in the box and whisker plots are given from the minima to maxima, the bounds of the box represent the 25th percentile and 75th percentile, and the middle line of the box is the median. Data were analyzed by One-way ANOVA followed by Tukey's multiple comparisons test, n.s. represents no significant difference. Source data are available as a Source Data file.

that are not perfused (CD31+ but no lectin labeling). The irregular microvasculature structure, associated with poor hemodynamics and high metabolic demands of the tumor mass, creates an environment of relative hypoxia which contributes to tumor angiogenesis and often regions of necrosis, a pathognomonic histologic feature of GBM. IHC staining of junctional proteins showed that the immunofluorescence of Claudin-5, VE-Cadherin, Occludin, and JAM-A persisted during 7–21 dpi at both tumor core and margin (Supplementary Fig. 5a, Fig. 2c). However, there was a significantly lower level of ZO-1 expression at the tumor core at 14 and 21 dpi (Fig. 2d). Further quantification of the area fraction ratio of protein over blood vessel (CD31) suggested that the relative protein coverage ratio for Claudin-5, VE-Cadherin, Occludin, and JAM-A was comparable at the tumor core, margin, and contralateral side during 7–21 dpi. Although no apparent change in the ZO-1/CD31 was observed at 7 dpi, there was a significant decrease in this ratio at the tumor core and margin at 14 and 21 dpi (Fig. 2d). We speculate that the BBTB disruption in the tumor core during disease progression is partially due to the loss of ZO-1 coverage on immature newly formed vessels.

Taken together, the characterization of the PS5A1 GEMM demonstrates that it is a suitable in vivo model for studying the challenges of drug delivery across an intact BBTB that mimic the infiltrative tumor margin found in human GBM. Moreover, our spatiotemporal analysis of tumor vasculature suggests that the 73 C GEMM features robust angiogenesis and heterogeneous BBTB with immature dysfunctional tumor-associated vessels, similar to what is seen in the core region of human GBM. These two GEMMs faithfully recapitulate the characteristics of human GBM and make them suitable surrogates for assessing drug delivery strategies.

## OptoBBTB improves drug penetration in PS5A1 and 73 C GEMMs

We next investigated whether optoBBTB improves drug penetration in the PS5A1 diffusely infiltrative GEMM. First, TJ component JAM-A targeted nanoparticles (AuNP-BV11, 50 nm) were prepared, and their physicochemical properties were characterized (Supplementary Fig. 6). Their ability to modulate BBB and safety profiles have been thoroughly investigated[17]. These nanoparticles were i.v. injected into a tumor-bearing mouse, followed by the delivery of a transcranial 532 nm picosecond laser pulse to the tumor region to stimulate the AuNPs for BBTB modulation (optoBBTB, Fig. 3a). The 532 nm picosecond laser was exploited for optoBBTB since the wavelength matches well with the surface plasmon resonance peak of the 50 nm spherical gold nanoparticles (530 nm). Fluorescent dyes or therapeutics were then delivered to assess the BBTB permeability and brain uptake. To optimize the optoBBTB, a series of nanoparticle doses and laser fluences were tested (Supplementary Table 1). We selected 18.5 μg/g of AuNP-BV11 injection followed by 40 mJ/cm² laser fluence (1 pulse) for BBTB opening since it showed high opening efficacy with minimized nanoparticle injection (Supplementary Fig. 7a). We further demonstrated that optoBBTB modulation allowed the delivery of molecules of different sizes in PS5A1 GEMM, such as EZ-link biotin (660 Da) and Evans blue/albumin (66 kDa) (Fig. 3b). The BBTB modulation was reversible and largely recovered in 1 day (Supplementary Fig. 7b). To investigate if there is a laser-induced heating effect in the tumor area, we recorded the local temperature change using a FLIR ONE Thermal Imaging Camera before and after optoBBTB on the mouse's skull. The results show that the average temperature before and after optoBBTB was 32.1 ± 0.1 °C and 32.3 ± 0.2 °C in PS5A1 GEMM (Supplementary Fig. 7c), suggesting no apparent temperature increase after optoBBTB.

Since most chemotherapy drugs are administered over multiple doses with intervals for recovery, it is important to assess the feasibility of multiple BBTB modulations for drug delivery. Taxol is a microtubule-stabilizing drug approved by the FDA for the treatment of ovarian, breast, and lung cancer, as well as Kaposi's sarcoma[21]. Following the failure of Taxol to show efficacy in an early-phase clinical

trial for GBM, further testing was abandoned. However, Taxol cannot pass through the BBB, which may partly account for the lack of clinical efficacy. To investigate the effectiveness of optoBBTB in Taxol delivery, we first demonstrated that optoBBTB using i.v. injection of AuNP-PEG with no specific targeting to TJ protein did not improve the delivery of Taxol Janelia Fluor 646 (Taxol646) into the tumor region (Fig. 3c, Supplementary Fig. 7d). Next, we performed optoBBTB with i.v. injection of AuNP-BV11, followed by the administration of Taxol646 to PS5A1 GEMM for three times with 3 days between treatments, to investigate the impact of multiple BBTB openings during tumor treatment. Figure 3d shows that the first cycle of optoBBTB at 14 dpi led to an increase in Taxol delivery and accumulation in the tumor core and margin area, as did the second and third optoBBTB cycles (18 and 22 dpi). Notably, there was no evidence of fluorescent Taxol leakage in the contralateral hemisphere, which reconfirmed the inability of this drug to pass through the normal BBB. We further analyzed the bioaccumulation and biodegradation of the gold nanoparticles in the tumor and healthy brain in PS5A1 GEMM after each optoBBTB by Inductively Coupled Plasma Mass Spectrometry (ICP-MS). The results show that there was an increased gold accumulation in the brain and the tumor, i.e., from 0.9 ± 0.5 μg/g to 4 ± 1 μg/g in the brain and from 1.3 ± 0.4 μg/g to 3.6 ± 1.3 μg/g in the tumor. No significant difference in the gold concentration was observed in the tumor and healthy brain (Supplementary Fig. 7e, Supplementary Table 2). Moreover, the slow gold clearance profile in mice was in agreement with the literature (Supplementary Fig. 7f)[32–34]. In summary, optoBBTB can be repeated and allows a multiple-cycle treatment regimen in PS5A1 GEMM that recapitulates the tumor margin histopathological characteristics.

We subsequently investigated the efficacy of optoBBTB in the 73 C GEMM. The overexpression of JAM-A in the tumor area due to the formation of angiogenic vessels made the nanoparticles attractive for enhanced GBM accumulation (Fig. 4a). ICP-MS analysis showed a >50% increase of AuNP-BV11 accumulation in the tumor compared with normal brain tissue (3.0 ± 0.5 versus 1.8 ± 0.2 μg/g in tumor and normal brain, respectively, Supplementary Fig. 8a). We further optimized the optoBBTB in the 73 C GEMM to obtain the optimal opening efficiency after single-pulse laser stimulation (Supplementary Fig. 8b, Supplementary Table 3). The highest BBTB opening level was achieved by injecting 37 μg/g of AuNP-BV11 and applying 40 mJ/cm² laser excitation (1 pulse). The BBTB recovered within 1 day, and no dye leakage into the brain was observed afterward (Supplementary Fig. 8c). Since the BBTB in 73 C GEMM remained intact at 7 dpi, optoBBTB significantly improved the delivery of both small molecules (EZ-link biotin, 660 Da) and large molecules (Evans blue, 66 kDa) after i.v. injection (Fig. 4b), while BBTB modulation using AuNP-PEG did not increase the Taxol646 delivery into the tumor (Fig. 4c, Supplementary Fig. 8d). The local temperature measurement shows that the average temperature before and after laser excitation was 32.1 ± 0.2 °C and 31.9 ± 0.1 °C (Supplementary Fig. 8e), indicating no temperature increase after optoBBTB in 73 C GEMM. Similarly, a three-cycle treatment regimen could be used for drug delivery in 73 C GEMM (Fig. 4d).

We noted that the BBTB modulation displayed a higher efficiency in the PS5A1 GEMM than in the 73 C GEMM (Figs. 3, 4, Supplementary Figs. 7, 8), although there was a significantly higher AuNP-BV11 accumulation in the tumor core of 73 C GEMM compared with PS5A1 GEMM (3.0 ± 0.5 μg/g, and 1.3 ± 0.4 μg/g, respectively, Supplementary Table 4). To increase the BBTB opening efficiency in the 73 C GEMM, we attempted to functionalize AuNPs with other vasculature targets, such as the anti-vascular endothelial growth factor 2 (VEGFR2) antibody and the anti-transferrin receptor (TfR) antibody, since VEGFR2 and TfR overexpression was observed in 73 C GEMM (Supplementary Fig. 9a, b). However, these nanoparticles did not improve BBTB opening efficiency compared with AuNP-BV11 (Supplementary Fig. 9c). To probe the mechanisms of the optoBBTB, we analyzed the changes in the irregular blood vessels in 73 C GEMM after laser stimulation

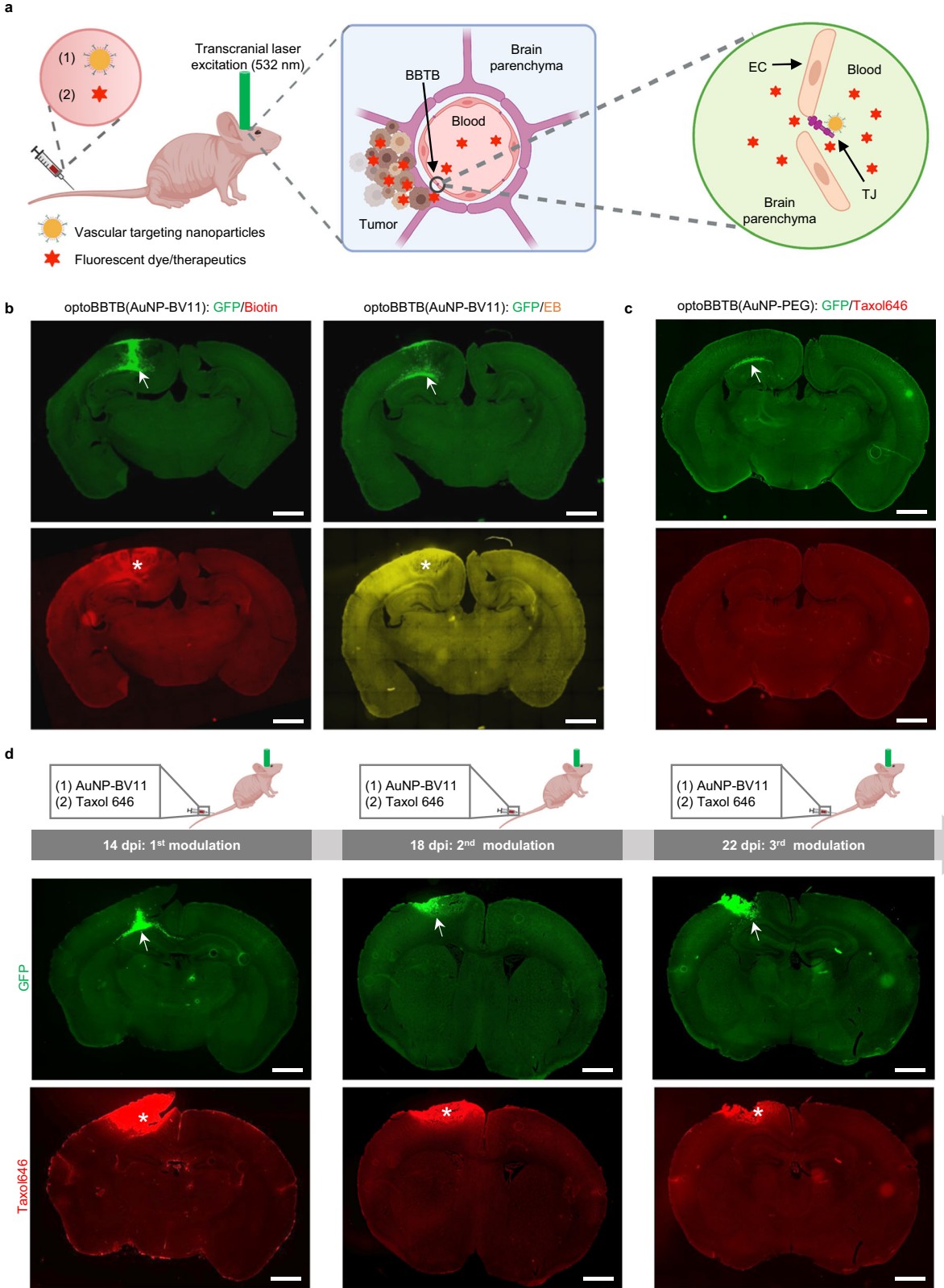

**Fig. 3 | OptoBBTB improves drug penetration to the brain in the infiltrative PS5A1 GEMM. a** Schematic illustration of optoBBTB. EC: endothelial cell. TJ: tight junction. The illustration figure was created with Biorender.com. **b** Delivery of EZ-link biotin (Biotin) and Evans blue (EB) after optoBBTB using ps laser and AuNP-BV11. The tumor is indicated by GFP fluorescent (arrows), and BBTB opening is characterized by Biotin or EB leakage (asterisks). The scale bar represents 1 mm. **c** OptoBBTB using ps laser and AuNP-PEG does not improve the delivery of fluorescent Taxol (Taxol646). The tumor is indicated by GFP fluorescent (arrow). The scale bar represents 1 mm. **d** Multiple BBTB modulations in the PS5A1 GEMM at 14-, 18-, and 22-days post injection (dpi) for fluorescent Taxol646 delivery. The tumor cells are indicated by GFP signal (arrows), and Taxol646 leakage is indicated by asterisks. The scale bar represents 1 mm. In **b, c**, three independent experiments were performed. In **d**, two independent experiments were performed. Similar results are provided in the Source Data file.

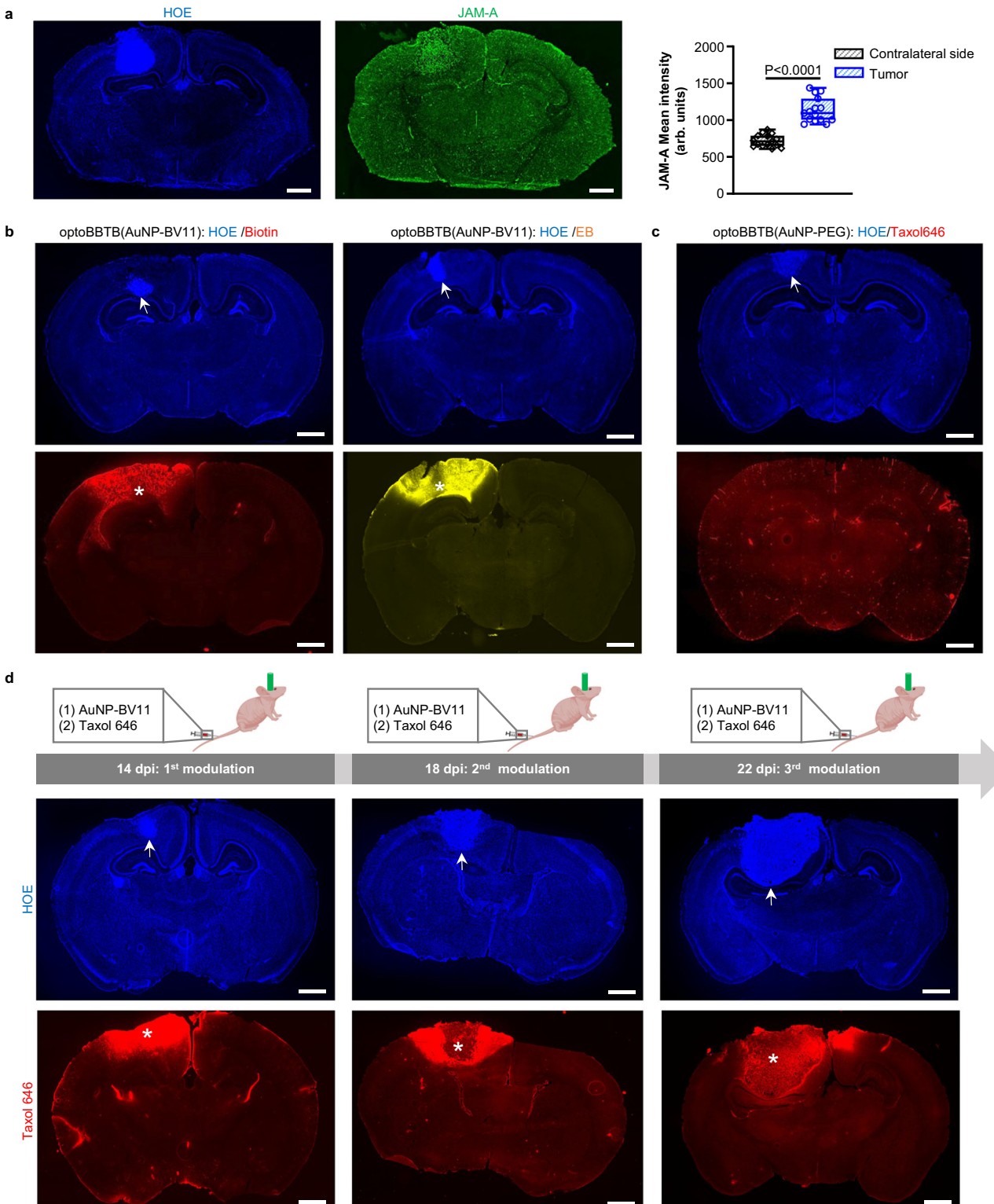

**Fig. 4 | OptoBBTB improves drug penetration to the brain in the angiogenic 73 C GEMM. a** IHC staining shows overexpression of JAM-A in the 73 C GEMM at 7 days post injection (dpi). The cell nuclei are indicated by Hoechst staining (HOE). The scale bars represent 1 mm. The quantification of JAM-A expression in the normal brain (contralateral side) and the tumor was performed by analyzing mean fluorescent intensity. $N = 15$ region of interests (ROIs) from 3 mice. Data in the box and whisker plots are given from the minima to maxima, the bounds of the box represent the 25th percentile and 75th percentile, and the middle line of the box is the median. Data were analyzed by unpaired Student's two-sided $t$-test. **b** optoBBTB with ps laser and AuNP-BV11 allows the delivery of small molecule EZ-link biotin (660 Da) and large molecule Evans blue (66 kDa, albumin-bound) to the tumor. The tumor is indicated by Hoechst staining of the cell nuclei (HOE, arrow). The scale bars represent 1 mm. **c** OptoBBTB using ps laser and AuNP-PEG does not improve the delivery of fluorescent Taxol (Taxol646). The tumor is indicated by Hoechst staining of the cell nuclei (HOE, arrow). The scale bar represents 1 mm. **d** Multiple BBTB modulations in the 73 C GEMM at 4, 8, and 12 dpi. AuNP-BV11 and Taxol646 were injected intravenously. The cell nuclei are indicated by Hoechst staining (HOE, arrow), and Taxol646 leakage is indicated by asterisks. The scale bars represent 1 mm. In **b**–**d**, three independent experiments were performed and similar results are provided in the Source Data file. Source data are available as a Source Data file.

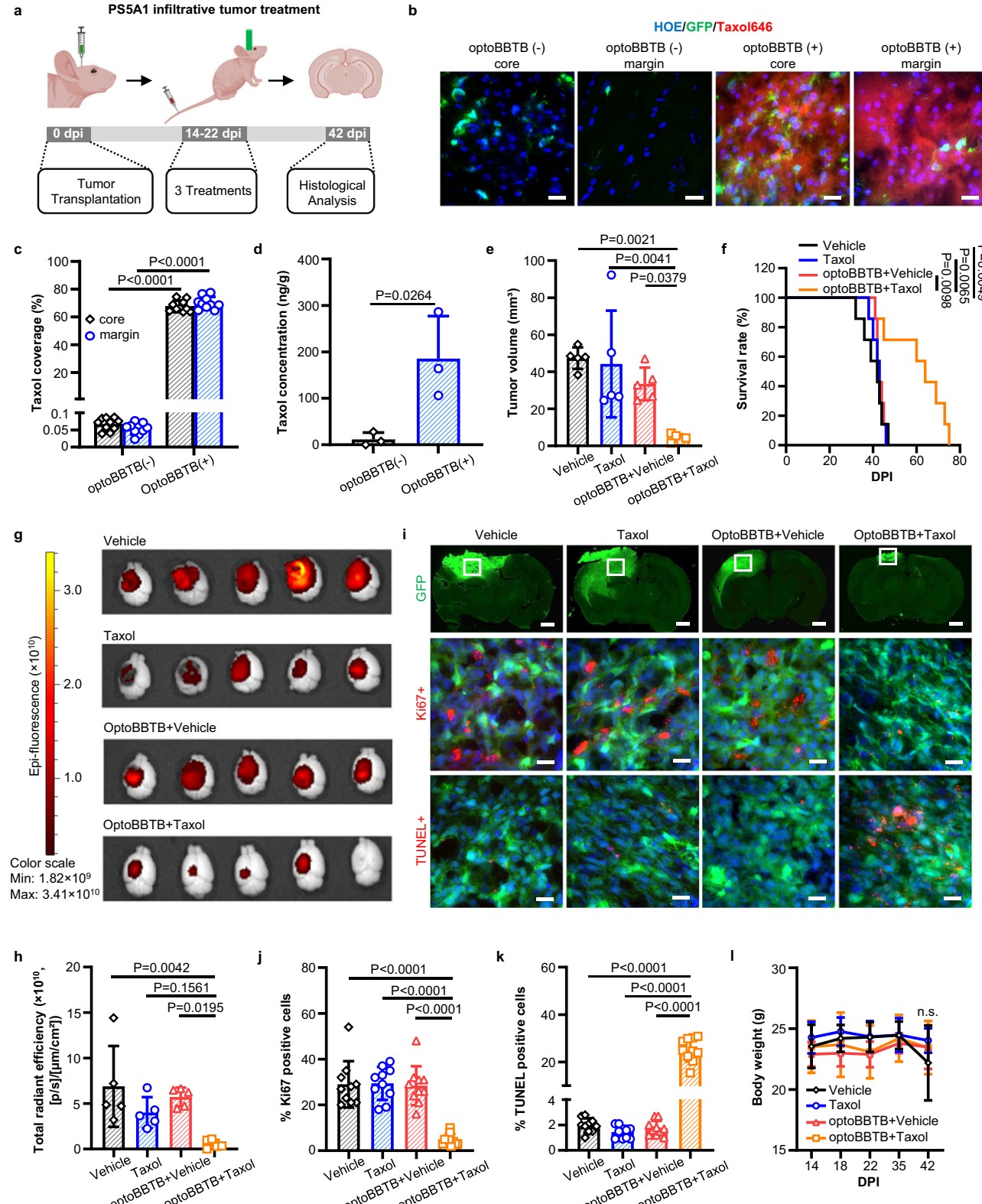

using IHC staining. The blood vessel density analysis showed that optoBBTB did not impact the vessel coverage percentages in the tumor core and margin (Supplementary Fig. 4a). Moreover, no significant difference in the immunofluorescence of junctional protein was observed before and after optoBBTB (Supplementary Fig. 5b). These results suggest that optoBBTB in 73 C GEMM did not influence the density or the junctional protein immunofluorescence of the angiogenic blood vessels. In our recent work[35], we demonstrated that

laser excitation of vascular-targeting AuNPs was associated with a transient elevation and propagation of $Ca^{2+}$, actin polymerization, and phosphorylation of ERK1/2 (extracellular signal-regulated protein kinase). They collectively activated the cytoskeleton resulting in increased paracellular permeability. We hypothesize that the increased barrier permeability after optoBBTB is due to the $Ca^{2+}$-mediated activation of the mechanobiological pathways and the re-arrangement of the cytoskeleton. Moreover, angiogenic blood vessels may respond

**Fig. 5 | OptoBBTB improves therapeutic outcomes in the infiltrative PS5A1 GEMM. a** Schematic illustration of the treatment timeline. The illustration figure was created with Biorender.com. **b**, **c** OptoBBTB facilitates the delivery of fluorescent Taxol646 to the tumor core and margin. The scale bar represents 20 μm. The quantification of Taxol delivery was performed by analyzing fluorescent area fraction. For each group, $N = 10$ images from 3 mice. Data are expressed as Mean ± SD. **d** The analysis of Taxol concentration in the tumor without or with optoBBTB at 14 days post injection (dpi). $N = 3$ mice. Data are expressed as Mean ± SD. **e** The analysis of tumor volume by GFP fluorescent signal at 42 dpi. $N = 5$ mice in each group. Data are expressed as Mean ± SD. **f** Kaplan-Meier survival analysis. $N = 7$ mice in each group. **g**, **h** Tumor size imaging by GFP fluorescent, and the quantification of GFP fluorescent using Living Image® Software for IVIS® Lumina

III In Vivo Imaging System. $N = 5$ mice in each group. Data are expressed as Mean ± SD. **i** Top: Tumor area indicated by GFP fluorescent at 42 dpi. The scale bar represents 1 mm. Middle: Ki67 staining shows cell proliferation. Bottom: TUNEL staining indicates cell apoptosis. The scale bars represent 20 μm. The ki67 and TUNEL images were taken from the boxes in the top lane. **j**, **k** Quantification of Ki67 staining and TUNEL staining after treatments. $N = 10$ images from 3 mice. Data are expressed as Mean ± SD. **l** The record of body weight change during 0-42 dpi in PS5A1 GBM treatment. $N = 5$ mice in each group. Data are expressed as Mean ± SD. Data in **c** and **d** were analyzed by unpaired Student's two-sided $t$-test, in **e**, **h**, **j**, **k**, and **l** were by One-way ANOVA followed by Tukey's multiple comparisons test, and in **f** were by logrank test. Source data are available as a Source Data file.

differently to optoBBTB than normal brain microvasculature. Further investigation may be focused on examining how the blood vessel phenotypes respond to optoBBTB and change the barrier permeability.

## OptoBBTB improves therapeutic outcomes for PS5A1 and 73 C GEMMs

With the optimized optoBBTB for Taxol delivery, we investigated the therapeutic outcomes in the PS5A1 GEMM. We began with evaluating the drug efficacy and mechanism of action in vitro. Taxol binds to the mitotic spindle apparatus and disrupts chromosomal segregation, which leads to mitotic catastrophe and cell death[21,36,37]. In vitro fluorescent imaging showed the internalization of Taxol646 in PS5A1 tumor cells after 1-h co-incubation (Supplementary Fig. 10a). These cells were sensitive to the Taxol with an $IC_{50}$ value of 6.3 nM after 72 h of incubation (Supplementary Fig. 10b). To assess the treatment efficacy of optoBBTB in PS5A1 GEMM, mice were randomly grouped at 14 dpi and treated intravenously with (1) vehicle, (2) free Taxol (12.5 mg/kg), (3) optoBBTB followed by vehicle delivery, and (4) optoBBTB followed by Taxol delivery (12.5 mg/kg). The treatment regimen included 3 cycles at 3-day intervals starting from 14 dpi to 22 dpi, and the treatment efficiency was evaluated at 42 dpi (Fig. 5a). The data show that optoBBTB greatly enhanced the delivery of fluorescent Taxol646 in the tumor core and margin compared with no optoBBTB treatment (Fig. 5b, c). The Taxol concentration in the tumor without or with optoBBTB was 12 ± 15 ng/g and 185 ± 92 ng/g, respectively, indicating a 16-fold concentration increase after optoBBTB (Fig. 5d). These tumors showed no T1-weighted contrast enhancement by MRI consistent with an intact BBTB and minimal T2-weighted hyperintensity (Supplementary Fig. 11a). This observation is consistent with the clinical scenario where GBM cells are undetectable to conventional MR sequences[38]. To verify the presence of tumor cells, we collected all the tumor-containing brain slices and analyzed the tumor volume by GFP fluorescent (Supplementary Fig. 11b). Remarkably, optoBBTB+Taxol yielded the smallest tumor volume (4 ± 2 mm³), a 5–7-fold reduction when compared with vehicle (47 ± 6 mm³), Taxol (44 ± 29 mm³), and optoBBTB+vehicle (33 ± 9 mm³) groups (Fig. 5e). Moreover, optoBBTB +Taxol delivery significantly increased the median overall survival by 50%, from 40 days to 60 days (Fig. 5f), due to a marked inhibition of tumor growth (Fig. 5g, h, i top). Ki67 staining and cell apoptosis analysis were performed by calculating the signal-positive (ki67+ or TUNEL + ) cell numbers over total cell numbers. The results show that optoBBTB+Taxol decreased cell proliferation and increased cellular apoptosis compared with the other groups (Fig. 5i middle-bottom, j, k). Body weight at 42 dpi was similar across treatment groups (Fig. 5l), indicating that the AuNP administration and the treatments did not induce significant additional systematic toxicity. These results demonstrate that optoBBTB allows the brain entry of Taxol, leading to treatment response and improved overall survival in infiltrative GEMM.

We subsequently analyzed the therapeutic effect of Taxol on the angiogenic 73 C GEMM. Co-incubation of 73 C tumor cells and Taxol646 (3 μM) showed that Taxol accumulated in the microtubules

enriched cytoplasm (Supplementary Fig. 12a), consistent with its known mechanisms of action to bind to and stabilize microtubules[39]. Furthermore, Taxol was highly potent against 73 C tumor cells with an $IC_{50}$ value of 10.52 nM (Supplementary Fig. 12b). To evaluate the optoBBTB in vivo on the angiogenic 73 C GEMM, we started with measuring the initial tumor volume by MRI at 3 dpi. The mice were randomly grouped and treated with (1) vehicle, (2) Taxol (12.5 mg/kg), (3) optoBBTB followed by vehicle, and (4) optoBBTB followed by Taxol (12.5 mg/kg). We performed three treatments at a 3-day interval (covering 4-12 dpi). At 15 dpi, we measured the tumor volume by MRI and harvested the brains for histological analysis (Fig. 6a). MRI T2-weighted scan was used to measure the tumor volume since T1-weighted gadolinium enhancement showed low signal intensity, probably due to the intact BBTB at the early tumor stage (e.g., 3 dpi, Supplementary Fig. 13a). Figure 6b, c show that a single dose of optoBBTB enhanced the delivery of Taxol to the tumor core and margin compared with no optoBBTB treatment. The Taxol concentration in the tumor without or with optoBBTB was 240 ± 168 ng/g and 1206 ± 1094 ng/g, indicating a 5-fold concentration increase after optoBBTB (Fig. 6d). The enhanced Taxol delivery produced a statistically significant difference in slowing the tumor progression and increasing survival (Fig. 6e–i). Tumor volume analysis by MRI showed that optoBBTB+Taxol delivery resulted in a 2.2 to 2.6-fold volume reduction in the tumor (36 ± 7 mm³) compared with vehicle (90 ± 20 mm³), Taxol (77 ± 8 mm³), and optoBBTB+vehicle delivery (90 ± 10 mm³) by MRI (Fig. 6e, Supplementary Fig. 13b). The smallest tumor size in the group of optoBBTB+Taxol was also confirmed by fluorescent imaging (Fig. 6g) and Hoechst staining of cell nuclei (Fig. 6i, top). Consistent with the smaller tumor volume seen at 15 dpi, the overall median survival of the mice was also increased by 33% from 18 days to 24 days after optoBBTB+Taxol treatment (Fig. 6f). Immunohistology analysis of the tumors in the optoBBTB+Taxol group showed a marked decrease in proliferation (Ki67 positive cells) as well as an increase in cell death (TUNEL positive cells) relative to the other cohorts (Fig. 6i middle, bottom, j, k). These data suggest that following BBTB disruption, Taxol can induce cell cycle arrest and cell death. No significant difference in body weight was observed by the end of three treatments (Fig. 6l). Taken together, our data show that optoBBTB can significantly enhance the delivery of Taxol to an angiogenic 73 C GEMM with a rapidly expanding tumor mass, which is sufficient to induce tumor cell death and cell-cycle arrest leading to increased overall survival.

## Discussion

GBM is considered surgically incurable due to its ability to diffusely infiltrate through the brain parenchyma, well beyond the regions outlined by T1-weighted contrast enhancement. There is increasing interest in extending the surgical margins to include the non-enhancing T2-weighted regions of abnormal signals. However, early clinical studies have shown only a marginal benefit, which must be weighed against the increased risk of neurological injury associated with removing the functional brain. Therefore, enhancing drug

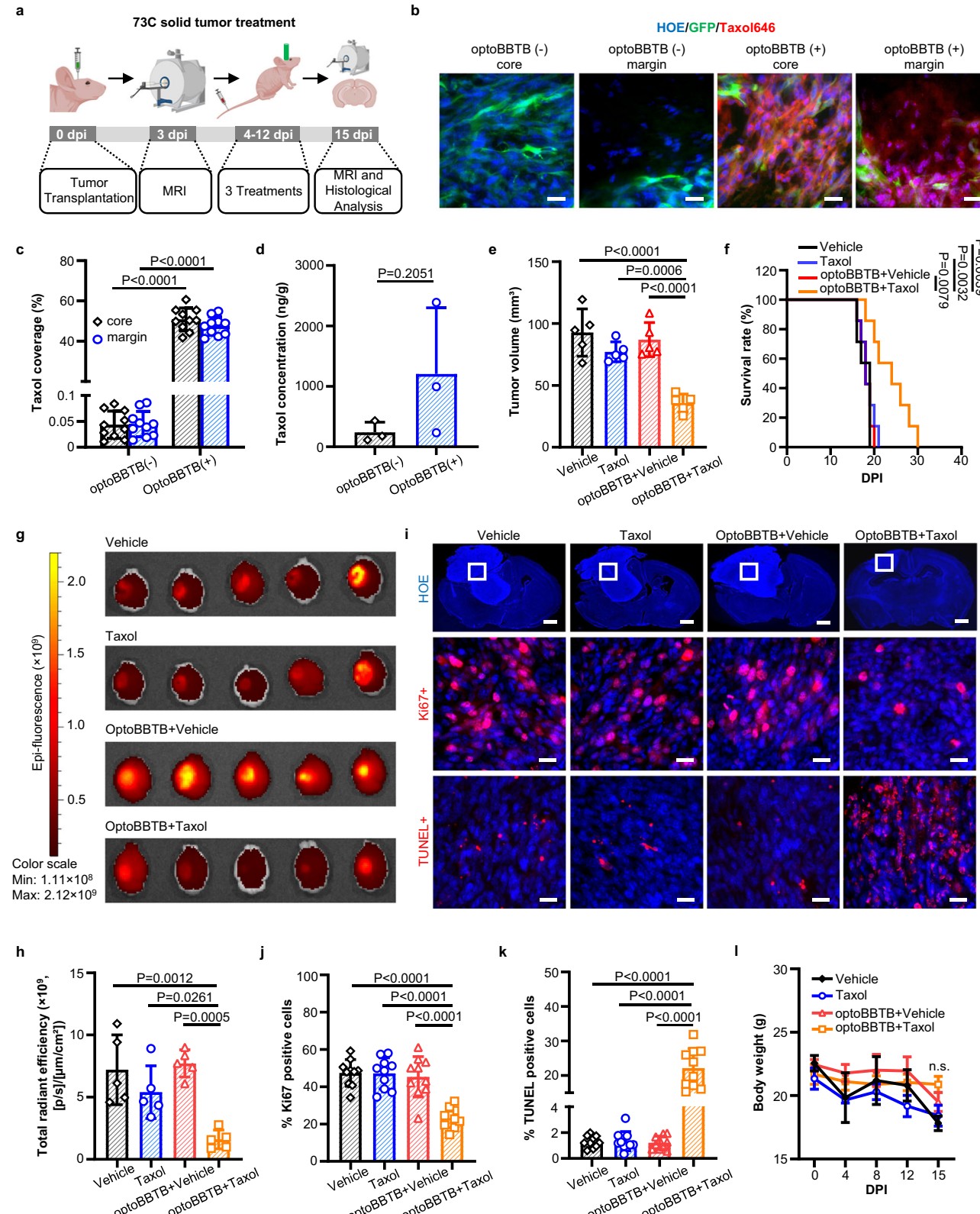

delivery to this area is critical to improving GBM treatment efficacy. The BBB has long been recognized as a significant impediment to developing more effective strategies to treat GBM. In this study, we showed that the optoBBTB reversibly increases BBTB permeability in two clinically relevant GEMMs that recapitulate GBM margin (infiltrative, PS5A1 GEMM) and core (angiogenic, 73 C GEMM) characteristics. Taxol is a common chemotherapeutic agent previously

abandoned following an early-phase clinical trial due to its limited brain penetration. Its efficacy in GBM treatment is currently under evaluation in combination with other BBTB penetration techniques but showed limited in vivo efficacy[40,41]. We demonstrated that optoBBTB increased Taxol delivery to both GEMMs at doses that significantly suppressed tumor growth by reducing tumor cell proliferation and inducing cell death, which further prolonged the survival of

**Fig. 6 | OptoBBTB improves therapeutic outcomes in the angiogenic 73 C GEMM. a** Schematic illustration of the treatment timeline. The illustration figure was created with Biorender.com. **b, c** OptoBBTB facilitates the delivery of fluorescent Taxol646 to tumors (7 days post injection, dpi). The scale bars represent 20 μm. The quantification analysis of Taxol delivery was performed by analyzing fluorescent area fraction. For each group, $N = 10$ images from 3 mice. Data are expressed as Mean ± SD. **d** The analysis of Taxol concentration in the tumor without or with optoBBTB at 7 dpi. $N = 3$ mice. Data are expressed as Mean ± SD. **e** The analysis of tumor volume was measured by MRI at 15 dpi. Each dot represents one animal. $N = 5$ mice in each group. Data are expressed as Mean ± SD. **f** Kaplan-Meier survival analysis, $N = 7$ mice in each group. **g, h** Tumor size imaging by GFP fluorescent, and the quantification of GFP fluorescent using Living Image® Software for IVIS® Lumina III In Vivo Imaging System. $N = 5$ mice in each group. Data are expressed as Mean ± SD. **i** Top: Tumor size indicated by fluorescent imaging at 15 dpi using Hoechst staining (HOE) of cell nuclei. The scale bars represent 1 mm. Middle: Ki67 staining showing cell proliferation. The scale bars represent 20 μm. Bottom: TUNEL staining indicates cell apoptosis. The scale bars represent 20 μm. The ki67 and TUNEL images were taken from the boxes in the top lane. **j, k** Quantification of Ki67 staining and TUNEL staining after treatment. $N = 10$ images from 3 mice. Data are expressed as Mean ± SD. **l** The record of body weight change during 0-15 dpi in 73 C GEMM treatment. Data are expressed as Mean ± SD. $N = 5$ mice in each group. Data in **c** and **d** were analyzed by unpaired Student's two-sided *t*-test, in **e**, **h**, **j**, **k**, and **l** were by One-way ANOVA followed by Tukey's multiple comparisons test, and in **f** were by logrank test. Source data are available as a Source Data file.

tumor-bearing mice without causing adverse effects. These results demonstrate that optoBBTB is effective for drug delivery and GBM treatment in two preclinical GEMMs. One limitation of our current study is that we tested the treatment using GEMMs in immunodeficient mice. It is important to further assess the efficacy of optoBBTB in immunocompetent models.

There is increasing recognition that one significant factor contributing to limited progress in identifying more effective therapies for GBM is the reliance on preclinical models, which fail to fully recapitulate GBM pathophysiology[18]. GBM is characterized by a high degree of spatiotemporal heterogeneity. Driven by high levels of angiogenic signals, GBM cells can induce microvascular proliferation in the tumor core regions, which are irregular structures with poor hemodynamics and limited function[42]. On the other hand, GBM cells at the tumor margin are characterized by diffuse single-cell infiltration through the brain parenchyma, including neuron-rich regions of gray matter neuropil and along white matter tracts[43,44]. Here, GBM cells co-opt the pre-existing dense brain microvasculature for metabolic support and nutrient exchange without disrupting the normal structure or functions of the microvessels[45–47]. To this end, we utilized two genetically engineered GBM cell lines (PS5A1 and 73 C) to establish GEMMs. PS5A1 model has an infiltrative growth pattern with vessel co-option development, and the 73 C model recapitulates GBM features such as an angiogenic tumor core and intratumoral heterogeneity in terms of BBTB function and TJ composition. To generate a mouse glioma, it typically requires the activation of an oncogene (e.g., EGFR mutation, PDGFRa, cMET, Braf^V600E) in combination with loss of one or more suppressors (PTEN^{f/f}, P53^{f/f}, and INK4a/b^{−/−} Arf ^{f/f}). Our GEMMs include loss of critical tumor suppressor genes (PTEN^{−/−} and P53^{−/−}; or PTEN^{−/−} and INK4ab/Arf^{−/−}) that are seen in virtually all human GBM. While the Braf^V600E activating mutation is only seen in 5–7% of adult gliomas and is one of the most common mutations in pediatric gliomas, it is known to be a powerful activator of the mitogen-activated protein kinase (MAPK)/ERK signaling pathways, which is almost ubiquitously activated by any number of mutations (e.g., NF1 loss, EGFR mutations). Thus, in combination with PTEN loss, leading to activation of the phosphatidylinositol 3-kinase (PI3K/AKT) pathway, our GEMMs are driven by a powerful co-activation of both the AKT and ERK signaling pathways which are ubiquitously activated in GBM[30]. In addition, one of the major advantages of our combination of mutations is that while there is significant diversity of oncogenes that are seen in GBM which has influenced the choice of mutations that are selected to generate GEM models[48–51], they share one common important feature that they all to some extent activate a common downstream RAS/RAF/MEK/ERK signaling pathway, which in turn regulates transcriptional networks to drive tumorigenesis. NF1 loss leads to RAS activation (product of NF1 is a negative regulator of RAS), while the Braf^V600E constitutively activates RAF which is directly downstream from RAS. As noted above, the Braf^V600E is one of the most common activating mutations in all cancers, implying that it is capable of activating a critical regulatory step in the process of malignant transformation. Taken together, we suggest that

our GEMMs represent a relevant in vivo model system for testing drug delivery following BBTB disruption.

OptoBBTB opens avenues of therapeutic interventions for GBM patients. Since the intercellular TJs represent a formidable barrier against paracellular drug delivery at the BBB[13,52], approaches have been developed to modulate the TJ to enhance the delivery across the BBB, including co-administration of siRNA against claudin-5 and occludin, as well as exploiting claudins or cadherin inhibitory peptides[53–56]. However, a lack of a robust delivery system in humans, poor targeting efficacy, or a lack of site-specificity impedes the successful translation of these approaches. Here we demonstrated that optoBBTB specifically targeted the JAM-A component of the TJ to modulate the BBTB locally and reversibly, and multiple openings could be achieved for anticancer drug delivery. Compared with the above-mentioned TJ modulation approaches, optoBBTB demonstrates advantages such as high targeting efficiency and site-specificity. Focused ultrasound (FUS) with circulating microbubbles (MB) is an emerging approach to modulate BBB permeability non-invasively and reversibly[16,57]. OptoBBTB is a complementary approach and offers the feasibility to tune the laser beam size, enabling BBB modulation in larger or smaller areas.

Although modulation of the BBB permeability using the laser with/without plasmonic nanoparticles has been reported elsewhere[58–60], our method has three critical differences compared to these strategies. First, our AuNP-BV11 nanoparticles bind to the vasculature, and the optoBBB modulation approach involves the increase of paracellular permeability without causing damage to the blood vessels and junctional proteins under the conditions investigated in this work. It is worth noticing that these changes were assessed 30 min after optoBBTB, and further examination should be conducted at longer time points to obtain a comprehensive assessment. Second, optoBBTB exploits nanoscale mechanical perturbation rather than the heating effect[61]. Since the duty cycle of the laser is low, the amount of heat dissipated from the nanoparticle into the surrounding medium is insufficient to raise tissue temperature. Last, since our control group (optoBBTB+Vehicle) did not show a reduction of tumor volume or an increased survival rate, and the local temperature was not enhanced after laser excitation, we believe that the reduced tumor volume is mainly mediated by the extravasation of Taxol rather than by other mechanisms such as the heating effect. Recently, we revealed the mechanisms of the optical BBB modulation using an in vitro BBB model established with human cerebral microvascular endothelial cells. We showed that the picosecond laser excitation of vascular-targeting AuNPs produced nanoscale mechanical perturbation, which triggers several mechanobiological responses that lead to an increased paracellular permeability[35]. While it may contribute to the increased drug accumulation in tumor after optoBBTB, further work is warranted to investigate the BBTB opening mechanisms in the current models.

We compared the optoBBTB in these two GEMMs regarding nanoparticle targeting and the BBTB opening efficiency. Our results show that after nanoparticle administration, there was a significantly

higher gold accumulation in the tumor core of 73 C GEMM compared with PS5A1 GEMM. Therefore, the different blood vessel phenotypes might influence the nanoparticle binding efficiency, probably due to the increased JAM-A expression in the angiogenic 73 C GEMM. However, our results show that optoBBTB in PS5A1 GEMM was more efficient than in 73 C GEMM, regardless of the nanoparticle targets, for example increasing Taxol delivery by 16-fold vs 5-fold in these two models, respectively. We hypothesize that the blood vessel phenotypes may also respond differently to the mechanobiological activation of the BBB[35]. Therefore, PS5A1 GEMM with normal microvasculature might demonstrate a higher optoBBTB efficiency than angiogenic 73 C GEMM. Additional research is necessary to explore the impact of the vascular phenotype on optoBBTB.

OptoBBTB presents opportunities for further investigations. The 532 nm light exploited in the current study enables light delivery to the mouse cortex, and therefore, our method can be useful as a drug development and screening platform (optoBBTB and GEMMs) for testing a class of potent anticancer drugs for superficial tumors. Several approaches can be exploited to further advance the technique, such as utilizing near-infrared laser and near-infrared light absorbing nanoparticles to improve the light penetration depth in the tissue, or using optical fiber in the tumor surgical cavity for light delivery into deeper brain regions.

In conclusion, we proposed a brain tumor treatment strategy exploiting optoBBTB for drug delivery in clinically relevant GEMMs. Our work on the GBM intratumoral heterogeneity analysis and the GEMMs development has broad implications, as there is significant interest in developing and translating technologies on BBB modulation for brain tumor treatment using various modalities. Furthermore, our investigation shows that BBTB modulation benefits brain tumor treatment and opens up avenues to test a class of potent anticancer drugs by combining with strategies to increase brain permeability. These lines of investigation will bring perspectives and build a solid preclinical foundation for future investigations in brain tumor treatment.

## Methods

### Approvals and authorizations
Animal protocols were approved by the Institutional Animal Care Use Committee (IACUC) of the University of Texas at Dallas. For human research, one female patient (64-year-old) was chosen to illustrate the MRI findings of human GBM recurrence as a point of reference for the study. The study protocols using human GBM samples were performed under STU-022-011, Retrospective Studies on Clinicopathological Correlation in Neuro-Oncology. The study was approved by the Institutional Review Board (IRB), University of Texas Southwestern Medical Center. Written consent was obtained from the patient to collect the blood and residual tissue samples in the case of surgical resection. The authors state that all human experiments were performed in strict accordance with the relevant laws and institutional guidelines.

### Materials
Gold (III) chloride trihydrate, hydroquinone, sodium citrate tribasic dihydrate, endotoxin-free ultrapure water, Evans blue, EZ-link biotin, Triton X-100, Paclitaxel (Taxol), Cremophor® EL, sucrose, Epidermal growth factor (EGF), Recombinant Human Fibroblast Growth Factor Basic (FGF2), and Progesterone were purchased from Sigma-Aldrich. OPSS-PEG-SVA (3400 Da) and mPEG-thiol (1000 Da) were purchased from Laysan Bio, Inc. Endotoxin-free ultrapure water, Donkey serum, phosphate-buffered saline (sterile), saline (sterile), Hoechst stain (33342), borate buffer, paraformaldehyde (4% in PBS), gold reference standard solution, lectin Dylight 594, 20 kDa molecular weight cut off (MWCO) dialysis membrane, Gibco™ DMEM (high glucose, Gluta-MAX™, DMEM, Dulbecco's Modified Eagle's Medium)/F12 50:50 Mix

(DMEM/F12), fetal bovine serum (FBS), StemPro™ Accutase™ Cell Dissociation Reagent, trypsin-EDTA solution (0.25%), penicillin-streptomycin, Tissue-Plus™ O.C.T. compound, Invitrogen Fluoromount-G™ mounting medium, B-27™ supplement, Insulin Transferrin solution, and Cayman Chemical WST-1 Cell Proliferation Assay Kit were purchased from Thermo Fisher Scientific. All chemicals were analytical grade unless specified. Taxol Janelia Fluor® 646 was purchased from Tocris Bioscience. ApopTag® Red In Situ Apoptosis Detection Kit (S7165) was purchased from Millipore Sigma. Buprenorphine SR-LAB 5 mL (1 mg/mL) was purchased from ZooPharm.

Anti-JAM-A antibodies BV11 and BV12 (for IHC staining) were provided by Dr. Monica Giannotta at FIRC Institute of Molecular Oncology Foundation. Rat anti-CD31 (550274) was purchased from BD Biosciences, goat anti-CD31 (AF3628), rabbit anti-Claudin-5 (34-1600), rabbit anti-ZO-1(40-2200), rabbit anti-VE-cadherin (36-1900), Rabbit anti-Occludin (71-1500), rabbit anti-ki67 (MA5-14520), donkey anti-goat IgG Alexa 488 (A-11055), donkey anti-rat IgG Alexa 594 (A-21209), donkey anti-rabbit IgG Alexa 594/647 (A-21207, A-31573) were purchased from Fisher Scientific. Rat anti-transferrin (8D3, NB100-64979) was purchased from Novus Biologicals. Rat anti-VEGFR2 (BE0060) was purchased from Bio X Cell,

### Animals
The immunodeficient nude mice Foxn1[nu] (Nu/J, stock number 002019, 7 weeks old, female, 20–25 g) were ordered from Jackson Laboratories. All animals were bred in pathogen-free conditions, in temperature (20–22 °C) and humidity (52–57%)-controlled housing, under a 12-h light/dark cycle, and with free access to food and water. Mouse sex was not considered in the study design.

### Human GBM sample
Fresh brain tumor samples were obtained from adult patients during their operative procedure after informed consent was obtained. Brain tumors were graded, and tumor core and tumor margin were also defined by T2-weighted fluid-attenuated inversion recovery (FLAIR) imaging of the patient's brain, at the Southwestern Medical Centre, University of Texas, by a neuropathologist according to World Health Organization guidelines. In this study, we aim to use the patient data to show the intratumoral BBTB heterogeneity and the tumor recurrence at the infiltrative and intact tumor margin, to emphasis the necessity for BBTB opening and drug delivery in this region. Since we did not choose the genes to modify based on this patient data, the patient's WHO classification, and underlying tumor genetics are not relevant information for this study design.

### Glioma cell line and cell culture
PS5A1 and 73 C glioma cells were generated from Dr. Robert Bachoo's laboratory[29,30]. PS5A1 is a highly invasive mouse glioma cell line derived from de novo glioma in the adult BL6 background conditional mouse (LSL.Braf[V600E f/+;] INK4a/b.Arf[f/f]; PTEN[f/f]) that was induced by intracranial injection of AAV5-GFAP-Cre-GFP. 73 C glioma cells were generated in primary astrocyte cultures from neonatal mice that carried conditional mutations for PTEN[f/f], P53[f/f], and LSL.Braf[V600E f/+]. These cells were derived from conditional multi-allele primary astrocytes and infected ex vivo with an adeno-Cre virus to generate a primary transformed cell line. The transient infection with the adenovirus infects nearly 100% of the cells (verified by fluorescent GFP), which ensures the expression of Cre-protein (CMV promoter-driven) and a complete excision of the LSL-stop codon to activate the mutant Braf[V600E] from its endogenous promoter. Since the AAV transduction provides a transient GFP expression, these cells were further infected with Lenti-GFP and selected by puromycin for stable green fluorescent protein expression.

PS5A1 cells were cultured as free-floating neurospheres in DMEM/F12 medium, 2% of B-27, 20 ng/mL of EGF, and 20 ng/mL of FGF2, 20 ng/mL progesterone, and 1% insulin transferrin solution. 73 C

glioma cells were cultured in DMEM containing 10% fetal bovine serum (FBS) and 1% penicillin-Streptomycin.

Both cell lines used in this study were free of mycoplasma contamination (Lonza's MycoAlert® Mycoplasma Detection Assays). We routinely test the cell cultures when revived from liquid nitrogen storage and again repeat testing if cells maintained continually for >30 days.

### In vitro dose responses

3000 cells in 100 μL of cell culture medium were seeded in each well of the 96-well plate. A day later, 100 μL of taxol solution (0-1000 nM) was added to each well, and the solution was gently mixed and incubated in a $CO_2$ incubator (37 °C) for 72 h. Then the cell viability was measured using WST-1 assay and reading the absorbance at 450 nm. The effect of the vehicle (Cremophor® EL and ethanol) was excluded by incubating the cells with the vehicle, and the absorbance at 450 nm was substituted from the reading.

### In vitro cellular uptake of fluorescent Taxol

Glioma cells were seeded in a glass-bottom culture dish (35 mm petri dish with 10 mm microwell) with a density of 70000 cells/cm². One day later, Taxol Janelia Fluor® 646 was added to the dish to achieve a final concentration of 3 μM, and the cells were incubated for 1 h. Then cells were washed and fixed with 4% PFA for 10 min. The cell nuclei were stained with Hoechst staining for 10 min. Finally, the cells were washed three times with PBS before imaging.

### Glioma cell transplantation

All glioma cells used for transplantation were passaged 2 days before transplantation to reach 70–80% confluence. The glioma cells were dissociated and resuspended in Hank's Balanced Salt Solution (without Ca²⁺, Mg²⁺, HBSS). Cells were counted, and a suspension was prepared with a density of $2 \times 10^5$/μL.

Buprenorphine (1 mg/kg) was given to the mice subcutaneously before surgery. 368 nL of PS5A1 glioma cell or 92 nL of 73 C glioma cell suspension in HBSS was constantly injected into the mouse cortex (−1 mm, −1 mm, 0.5 mm) using a nanoinjector (World Precision Instrument) equipped with a glass micropipette. The glass micropipette has a 50 μm tip and was generated by a Micropipette Puller (P-1000, Sutter Instrument Co.). After injection, a thin layer of glue was applied over the exposed skull and the surrounding skin, and a layer of body double (Body Double™ Standard Set, Smooth-On, Inc.) was applied to the skull for protection. The mice were housed for 3 days (73 C) to 2 weeks (PS5A1) to allow tumor growth before starting the experiments. The animal ethics guidelines of the Institutional Animal Care and Use Committee (IACUC) at the University of Texas at Dallas were strictly followed, ensuring that the mouse was euthanized upon reaching either when its brain tumor surpassed the predetermined maximum volume of 1 cm³ or when its weight loss exceeded 20%.

### Dye permeability experiment

EZ-link biotin (660 Da, 2 mg/mL) and Evans Blue (66 kDa/albumin-bound, 2% w/v) were intravenously injected into two tumor-bearing mice at (1) 14-, 28- and 42 dpi for PS5A1 GEMM, and (2) 7-, 14- and 21 dpi for 73 C GEMM. 30 min after the dye injection, the mice were sacrificed by cervical dislocation, and the brains were extracted and post-fixed in 4% (w/v) PFA at 4 °C for 24 h. The brains were then snap-frozen on dry ice and cut into 20 μm thick coronal slices using a cryostat. The slices were stained with cy3-streptavidin to detect biotin and Hoechst solution to label nuclei before imaging.

### Immunohistochemical (IHC) staining

To immunostain vascular biomarker (CD31) and junctional proteins (i.e., Claudin-5, ZO-1, VE-cadherin, Occludin, and JAM-A), the mice brains were snap-frozen on dry ice once quickly removed from the skull and cut to 20 μm thick coronal slices on a cryostat. To analyze the influence of optoBBTB on vessel density and the immunofluorescence of the junctional proteins, the brains were collected at 30 min after the optoBBTB, followed by cryosectioned to 20 μm thick coronal slices. The brain slices were fixed for 10 min using ice-cold methanol at −20 °C. Blocking solution (5% normal donkey serum, 0.1% Triton X-100 in PBS) was applied to the tissue for 1 h at room temperature. After washing, the sections were first incubated overnight at 4 °C with the following primary antibodies: (1) rat anti-CD31, rabbit anti-Claudin-5, rabbit anti-ZO-1, rabbit anti-VE-cadherin, rabbit anti-Occludin, or (2) goat anti-CD31 and rat anti-JAM-A (BV12). The antibody dilution ratio was 1:500 in PBS, and 1 mL of the antibody solution was applied to 10–12 brain slices.

To immunostain transferrin receptor and vascular endothelial growth factor receptor 2 (VEGFR2) or to detect cell proliferation after treatment, the mice were perfused with PBS and 4% PFA. The brains were extracted and post-fixed in 4% PFA overnight, then cut into 20 μm thick coronal slices on a cryostat. The slices were incubated in applied to the tissue for 1 h at room temperature and then incubated with primary antibodies (1) goat anti-CD31, rat anti-Transferrin (8D3), rat anti-VEGFR2, or (2) rabbit anti-ki67 with a dilution ratio of 1:500 in PBS. 1 mL of the antibody solution was applied to 10–12 brain slices.

In all cases, the sections were then incubated with secondary antibodies (the dilution ratio was 1:500 in PBS) for 2 h at room temperature, followed by Hoechst solution (the dilution ratio was 1:2000 in PBS). 1 mL of the antibody solution or Hoechst solution was applied to 10–12 brain slices. The slides were washed with PBS and mounted with Fluoromount-G mounting medium for imaging.

### The preparation of brain vascular-targeting gold nanoparticles and control gold nanoparticles

The method to prepare brain vascular-targeting gold nanoparticles (AuNPs-BV11) was adapted from our previously reported method[17]. The 50 nm AuNPs were synthesized via seed-mediated growth starting from 15 nm gold seeds. To synthesize the seeds, 98 mL of ultrapure water was added to a 250 mL glass flask that was cleaned with aqua regia and rinsed thoroughly with ultrapure water. 1 mL of gold chloride solution (25 mM) was added to the flask. The solution was boiled, and 1 mL of trisodium citrate solution (112.2 mM) was added under vigorous stirring. The nanoparticles were boiled and stirred for 10 min, then cooled to room temperature. To synthesize 50 nm AuNP, 94 mL of ultrapure water was added to a 250 mL glass flask, followed by the sequential addition of 963 μL of gold chloride solution (25 mM), 3.7 mL of seeds (2.2 nM), 963 μL of trisodium citrate (15 mM) and 963 μL of hydroquinone (25 mM) under vigorous stirring. The resulting nanoparticles were stirred at room temperature overnight, and concentrated for further use. To prepare brain vascular-targeted AuNPs, BV11 (anti-JAM-A antibody) was diluted to 0.5 mg/mL in PBS, and then diluted in 2 mM ice-cold borate buffer (pH 8.5) to 0.05 mg/mL. OPSS-PEG-SVA was dissolved in 2 mM borate buffer and mixed with the BV11 antibody at a 125:1 molar ratio. The solution was incubated for 3 h on ice, followed by dialysis at 4 °C overnight using a 20 kDa MWCO membrane. The thiolated BV11 antibodies were then collected and mixed with the concentrated AuNPs at a 200:1 molar ratio, and incubated for 1 h on ice. To stabilize AuNP-BV11, mPEG-thiol (1 kDa) was added at 6 PEG/nm² and incubated for 1 h on ice. Finally, the modified AuNPs were washed three times with ice-cold borate buffer. AuNP-TfR and AuNP-VEGFR2 were prepared using a similar approach by replacing BV11 with the anti-TfR antibody and anti-VEGFR2 antibody. The control nanoparticles were prepared by incubating 50 nm AuNP with mPEG-thiol (PEG: polyethylene glycol, 1 kDa) with a thiol:AuNP=300:1 molar ratio for 3 h on ice. The particles were washed three times in borate buffer. The concentration, size distribution, and morphology of the nanoparticles were analyzed by Ultraviolet-Visible-Near Infrared absorption spectra with Gen5 software for BioTek Synergy 2 plate

reader, Dynamic Light Scattering (DLS) with Malvern Zetasizer software, and Transmission Electron Microscopy (TEM) with Gatan DigitalMicrograph.

## OptoBBTB optimization in GEMMs

The tumor-bearing mouse was anesthetized by 2–3% isoflurane (in air) and intravenously administrated AuNP-BV11. 1 h later, the body double on the scalp was peeled off to expose the skull. One pulse of the picosecond (ps) laser was applied to the tumor region through the intact skull. To optimize the BBTB modulation to achieve the highest opening efficiency, different nanoparticle doses and laser fluence conditions were tested with PS5A1 and 73 C GEMMs, as shown in Supplementary Table 1, 3. Evans blue dye (2% in PBS, 100 µL) and EZ-link biotin (2 mg/mL, 100 µL) was injected to visualize the BBTB modulation. 30 min after the laser excitation, the mice were perfused with PBS and 4% PFA, and the brains were extracted and post-fixed with PFA overnight. The brains were frozen on dry ice and cut into 20 µm thick slices using a cryostat. The slices were stained with cy3-streptavidin to detect biotin before imaging.

## Inductively coupled plasma mass spectrometry (ICP-MS)

We used ICP-MS to measure the AuNP-BV11 accumulation in PS5A1 GEMM (14 dpi) after each optoBBTB (40 mJ/cm², 1 pulse, repeated three times at 14, 18, and 22 dpi, nanoparticle administration dose was 18.5 µg/g). We also studied the nanoparticle degradation profile using healthy Nu/J mice. The mice received three nanoparticle injections and laser treatments at 3 days intervals, and the gold concentration in each organ was measured 60 days after the third nanoparticle injection and laser excitation. ICP-MS was also used to determine the biodistribution of AuNP-BV11 (37 µg/g) after intravenous (i.v.) injection to the 73 C glioma-bearing mice (7 dpi). 1 h after the nanoparticle injection, the mice were perfused with ice-cold PBS, and the main organs were collected. The tissue was then digested in fresh aqua regia until the tissue was fully dissolved. Then the solution was centrifuged at 5000 g for 10 min, and the supernatant was collected and diluted with ultrapure water for ICP-MS analysis with Agilent 7900 and MassHunter Software.

## Analysis of the BBTB opening window after laser excitation of AuNP-BV11

AuNP-BV11 (18.5 µg/g or 37 µg/g) was i.v. injected to PS5A1 and 73 C GEMMs at 14 and 7 dpi, respectively ($N$ = 3 mice for each group, namely mouse 1-3). The mice received a picosecond laser pulse (40 mJ/cm²) after 1 h. Then EZ-link biotin was injected immediately after laser excitation (mouse 1) or after 1 (mouse 2) or 3 days (mouse 3). 30 min after the dye injection, the brains were extracted and frozen on dry ice and cut into 20 µm thick slices using a cryostat. The brain slices were incubated with Cy3-streptavidin to detect biotin and Hoechst solution to stain cell nuclei. The biotin leakage was imaged using Olympus VS120 virtual slide microscope with a 10x objective.

## Local temperature measurement before and after optoBBTB

We used a FLIR ONE Thermal Imaging Camera for smartphones to measure the temperature change in the laser-irradiated region on the mouse's skull. The temperature before and after laser excitation was recorded using Vernier Thermal Analysis™ Plus. Briefly, a region of interest (ROI) that covered the laser irradiation region was manually selected on the app. We recorded a temperature baseline (1 min), followed by applying the laser (40 mJ/cm², 1 pulse), and then recorded continuously for 4 min.

## Fluorescent Taxol delivery to the tumor after optoBBTB

AuNP-BV11 (18.5 µg/g and 37 µg/g for PS5A1 and 73 C GEMMs, respectively) was i.v. injected into a tumor-bearing mouse (14 and 7 dpi, respectively). 1 h later, the mouse received a single pulse of ps laser (40 mJ/cm²), followed by Taxol Janelia Fluor® 646 (12.5 mg/kg) i.v.

administration. 30 min after laser excitation, the mouse was perfused with PBS and 4% PFA, and the brain was extracted and then post-fixed in 4% PFA overnight. The brain was frozen on dry ice and cut into 20 µm thick slices using a cryostat. The slices were stained with Hoechst to label the nuclei.

## Glioma treatment after optoBBTB

The treatment for PS5A1 GEMM was started at 14 dpi. The mice were randomly divided into four groups: (1) vehicle control; (2) free Taxol control (12.5 mg/kg); (3) optoBBTB+vehicle; (4) optoBBTB+Taxol (12.5 mg/kg) with five animals in each group. Specifically, Taxol was dissolved in a mixer of Cremophor EL: absolute ethanol (1:1 v/v, as the vehicle) to 6 mg/mL and then diluted to 2 mg/mL with saline. 18.5 µg/g of the AuNP-BV11 was i.v. injected into the tumor-bearing mouse. 1 h later, the mice received picosecond-laser excitation (40 mJ/cm², 1 pulse, pulse duration was 28 ps and beam size was 6 mm) in the tumor area, followed by receiving either vehicle or Taxol via i.v. injection. The treatment was repeated every 4 days and repeated three times. The mice were sacrificed at 42 dpi to analyze the tumor size.

For 73 C GEMM, the treatment was started on 4 dpi. The same treatment groups were used. To modulate the BBTB, 37 µg/g of the AuNP-BV11 was i.v. injected into the tumor-bearing mice. 1 h later, the mice received picosecond laser excitation (40 mJ/cm², 1 pulse) followed by either vehicle or Taxol via i.v. injection. The treatment was repeated every 4 days. Three treatments were conducted between 4–12 dpi. On 3 and 15 dpi, the tumor size at the start point and endpoint was measured by MRI (T2-weighted scan).

Similar treatment groups were used for PS5A1 and 73 C GEMMs to obtain the survival rate, with 7 mice in each group. The tumor-bearing mice were euthanized if they developed weight loss (>20%), loss of grooming, seizures, and focal motor deficits according to the approved animal protocol.

## Magnetic Resonance Imaging (MRI)

73 C glioma was visualized using a preclinical BioSpec 3 T MRI (Bruker, Billerica, MA, USA) with a mouse head coil (Item RF Res 128 1H 064/023 QSN TF, Model 1 P T167055, Serial S0013) with Paravision 360. T2-weighted scans were performed using a T2 RARE sequence with an echo time of 60 ms, repetition time of 2725.738 ms, an echo spacing of 15 ms, a rare factor of 10, 8 averages, 1 repetition, a slice thickness of 0.5 mm, a field of view of 20 by 20 mm, and a matrix size of 192 by 192. The number of slices ranged from 12 to 20 to cover the entire brain of each mouse. Images were exported to Digital Imaging and Communications in Medicine (DICOMs) and used Fiji/Image-J and ITK-SNAP for further analysis.

## In situ cell apoptosis detection

The cell apoptosis analysis was performed at 42 dpi (20 days after the 3rd treatment) or 15 dpi (3 days after the 3rd treatment) for PS5A1 and 73 C GEMMs, respectively. The mice were perfused with PBS and PFA, and the mice's brains were harvested and post-fixed in PFA overnight. Then the brains were snap-freeze on dry ice. 30 µm thick brain slices were cut on a cryostat, and the cell apoptosis detection was performed with ApopTag® Red In Situ Apoptosis Detection Kit following the manufacturer's protocol. Briefly, the slides were rinsed in two changes of PBS (5 min each) and post-fixed in pre-cooled ethanol: acetic acid 2:1 for 5 min at −20 °C, followed by rinsing twice in PBS. Then equilibration buffer (75 µL/5 cm²) was applied to the specimen and incubated for 1 min at room temperature. Then the excess liquid was removed, and the TdT enzyme was applied to the samples (55 µL/5 cm²), followed by incubation at 37 °C for 1 h. Subsequently, stop/wash buffer was added to the samples and incubated for 10 min at room temperature. The samples were washed three times in PBS, and anti-digoxigenin conjugate rhodamine was applied to the specimen (65 µL/5 cm²) and incubated for 30 min at room temperature. The

samples were washed four times in PBS, stained the cell nuclei with Hoechst staining (1:2000), and mounted with Fluoromount-G™ for imaging.

### Fluorescent microscopy, electron microscopy, and imaging analysis

All the fluorescent images were taken with the IVIS® Lumina III In Vivo Imaging System, Olympus SD-OSR spinning disk super-resolution microscope with Metamorph software, and Olympus VS120 virtual slide microscope with Olympus VS-ASW. The transmission electron microscopy (TEM) images were taken using a JEOL JEM-2010 microscope.

To measure the tumor size after treatment, the brains were extracted and imaged the GFP fluorescent using IVIS® Lumina III In Vivo Imaging System. The radiant efficiency was measured with Living Image® Software. To image the dye or Taxol extravasation after optoBBTB, the samples were imaged with a 10x objective (Olympus VS120 virtual slide microscope) or 100x oil immersion objective (Olympus SD-OSR spinning disk super-resolution microscope). To study the changes in junctional proteins using IHC staining, the samples were imaged with a 100x oil immersion objective (Olympus SD-OSR spinning disk super-resolution microscope). Then the images were analyzed by Fiji/ImageJ. The area fraction of Claudin-5, ZO-1, VE-cadherin, Occludin, and JAMA-A was obtained and normalized by CD31 (indicating cerebral vessel). To study the vasculature density, the images were acquired using a 10x objective (Olympus SD-OSR spinning disk super-resolution microscope). The vasculature density was analyzed by area fraction of CD31 or lectin. The ki67 and TUNEL staining were imaged with a 100x oil immersion objective (Olympus SD-OSR spinning disk super-resolution microscope). The image acquisition settings were kept constant during the experiment.

To obtain the tumor volume after treatment in PS5A1 GEMM, coronal brain slices (30 μm) were imaged with Olympus VS120 virtual slide microscope with a 10x objective. The image acquisition settings were kept constant between the samples. A threshold was set to cover the tumor areas using Fiji/Image-J (exemplified in Supplementary Fig. 11b), consistent across all brain slices analyzed. Therefore, the total tumor area can be measured using Fiji/Image-J. And the tumor volume can thus be determined by the product of total area and slice thickness.

### Statistics and reproducibility

Statistical analyses were performed using GraphPad Prism software. No data were excluded from the analyses. The survival time was analyzed by logrank test, other data were analyzed by one-way analysis of variance (ANOVA) followed by Tukey's multiple comparisons test and unpaired Student's two-sided $t$-test for two comparisons.

For in vitro experiment, six replicates were performed. For in vivo experiment, at least two independent experiments were performed. We used G*power analysis to calculate the sample sizes for tumor size and survival analysis. The effect size was obtained from our preliminary study. With 85% power, and alpha set to 0.05, the sample size required was calculated as $N = 5$ per group. The mice were randomly assigned to different experimental groups with no bias for further treatment. The investigators were not blinded to allocation during experiments and outcome assessment since the treatment groups were obvious from the result. The $N$ values per group and details of statistical testing are provided in the figure caption.

### Reporting summary

Further information on research design is available in the Nature Portfolio Reporting Summary linked to this article.

## Data availability

All data associated with this study are in the paper or the Supplementary Materials. The fluorescent images generated in this study have been deposited in the Zenodo database with the identifier [https://doi.org/10.5281/zenodo.8132255][62]. Source data are provided with this paper. Anti-JAM-A antibodies (BV11 and BV12) and glioblastoma cell lines are readily from the authors. Source data are provided with this paper.

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

## Acknowledgements

We thank Dr. Noelle Williams and Eric Crossley from the Preclinical Pharmacology Core at UT Southwestern Medical Center for the drug concentration analysis. We thank Dr. Jonghae Youn for assistance with

TEM imaging, Dr. Ning Gao for cell line genotyping, and Dr. Haihang Ye for reviewing and revising the manuscript. This research was partially supported by Cancer Prevention and Research Institute of Texas (CPRIT) grants RP190278 and RP210236, Department of Defense grant W81XWH-21-1-0219 and American Heart Association grant 19CSLOI34770004, National Institute of Health grant RF1NS110499, National Science Foundation grant 2123971, and funds from a Eugene McDermott Professorship to Z.Q. Magnetic resonance imaging in this research was supported in part by award RP180670 from the Cancer Prevention and Research Institute of Texas (CPRIT) to K.H. to establish the Small Animal Imaging Facility at the University of Texas at Dallas. Illustration figures were created with Biorender.com.

## Author contributions

Q.C. and Z.Q. conceived the original idea. Q.C. performed the experiment and analyzed the data. X.L., H.X., H.F. and X.Ge. assisted in experiments and analyzed the data. E.M. provided human GBM samples. X.Gao. and V.V. performed the IHC staining and imaging on human GBM samples. R.M., J.L., and K.H. performed magnetic resonance imaging. M.G. prepared the BV11 and BV12 antibodies. R.B. and Z.Q. supervised the project, analyzed the data, and discussed the results. Q.C. and Z.Q. wrote the paper. All authors revised the manuscript and have approved the final version.

## Competing interests

All authors declare no competing interests.
