## [Peer Review File · Nature Communications]

REVIEWER COMMENTS

Reviewer #1 (Remarks to the Author): Expert in glioblastoma genetically-engineered mouse models and therapy

This manuscript by Qin and colleagues details work in which they developed two new GEMMs for GBM, an area of unmet clinical need. The GEMMs are based on the BRAFv600E mutation, and despite being an extension of previous work, are useful novel tools for the study of GBM. One model in particular is claimed to be reflective of diffusely infiltrative GBMs, while the second model is more reflective of angiogenic GBMs. Next, the authors developed a treatment strategy based on pulsed laser excitation of vascular-targeted gold nanoparticles to enhance the BBTB permeability and increase Taxol entry. This led to improvement in survival of the mice, linked to reduced tumour cell proliferation. The work provides new models and a treatment strategy based on improving BBTB permeability which has exciting clinical implications. However, I do have some concerns which would need to be addressed prior to publication:

1. The authors do not show clear evidence of diffuse single-cell infiltration of tumour cells in the first model. Histology images reflecting this should be shown. Similarly, it is unclear if the BRAF mutation is present in all tumour cells - evidence to this effect should be presented.
2. The abnormal vasculature of the second model is clear, however the effects of their treatment strategy on this abnormal vasculature is not presented. It would be important to show how these vessels are changed in response to this treatment, given the vasculature is proposed to play a role in the pathogenesis of these tumours and influence the BBTB.
3. The relations of their model to other published GEMMs of GBM are not described, and it would be important to do so given the other important GEMMs available. For example, PMID: 26461091 uses Nf1, Trp53 and Pten mutations to generate GBM, and PMID: 32727536 used EGFRvIII in combination with TSG loss via transposons to produce GBMs. How do the authors model differ from these models which also reflect key characteristics of human GBM pathology and genomics? This needs to be highlighted in Discussion.
4. Descriptions of how the GEMMs were established should be made clearer - for example, it appears the cell transplants were done in nude mice but this is not apparent in main body of the text.

Reviewer #2 (Remarks to the Author): Expert in glioblastoma, nanoparticles, mouse models, and blood-brain barrier permeability

The authors have developed a novel method, optoBBTB, to modulate the barrier's permeability and increase the delivery of paclitaxel, which under normal circumstances is not effective in controlling glioma growth. The authors established two glioma GEMM models, displaying an angiocentric core, and

a diffuse infiltrative margin. They then developed nanoparticles activated by pulsed laser excitation to modulate the BBB to enhance the delivery of paclitaxel. The treatment increased survival by 50% and 33%.

This is an interesting manuscript, yet, the poor quality of many images limits a thorough assessment of their data.

In Fig. 1, it is not indicated where C, and D, originate from.

Equally, Fig. 1E is too dark to identify any structures, which moreover are not indicated with arrows.

The same problem is evident in Fig. 2. The immunofluorescent panels are too dark, and the authors have not indicated any particular structures analyzed.

Fig. 3 suffers from the same problems. The panels are too dark, and structures are not highlighted by arrows. A control group of laser stimulation on its own is missing. This figure also indicates a major shortcoming of this manuscript. BBB is only increased very near the original tumor. The actual problem in humans is to be able to target those cells that are far away from the tumor. These authors do not achieve BBB opening at large distances from the tumor. The tumor at 14 dpi in C is very small, and so are the tumors at 18, 24 dpi. Why are these tumors so small if no treatment has yet been delivered.

In Fig. 4, the increased survival achieved is very good. However, the location of all fluorescent images shown in Fig. 4 are not indicated, and thus, most of the figure is hard to evaluate.

Fig. 5 is uninterpretable and needs to be imaged again. All panels are too dark to identify any particular structures.

Fig. 6 has the exact same problems as Fig. 5.

Fig. 7 shows an acceptable survival increase. Yet, the anatomical provenance of all fluorescent images is not indicated. And all fluorescent images are too dark to interpret. The TUNEL images, in I, do not show what the authors wish to interpret. It is unclear how quantification was performed.

In summary, the authors need to redo all fluorescent panels in their figures when resubmitting the manuscript. The authors should update their Figures, including clear indications of where each high power magnification figure originates from.

In the Discussion, the authors need to tone down their assessment of the impact of their approach to open the BBB. Opening of the BBB in these experiments was restricted to the main tumor area, and maybe, a little bit further out. However, in human GBM the cells can be centimeters away from the main tumor mass, much farther than any drug penetration shown in this manuscript.

Once Figures have been improved as suggested, the manuscript could be resubmitted for further evaluation.

Reviewer #3 (Remarks to the Author): Expert in nanoparticles, brain tumours, and blood-brain barrier permeability

See attached file.

Reviewer #4 (Remarks to the Author): Expert in light-inducible nanoparticles

The paper entitled “Optical Blood-Brain-Tumor Barrier Modulation Enhances Drug Penetration and Therapeutic Outcome in Clinically Relevant infiltrative and Angiogenic Glioblastoma Models” is an interesting paper which describes the modulation of the blood brain barrier using plasmonic carriers that generate heat after laser stimulation to facilitate the extravasation of anticancer drugs. The paper is well-written however the narrative is complex since the authors have used two genetically engineered mouse models that recapitulate different phenotypes (i.e. diffusively infiltrative tumor margin and angiogenic core) in glioblastoma. This means that the authors needed to introduce and characterize both animal models before evaluating the potential of their opto-activation approach. The authors show that the anticancer drug extravasates to brain parenchyma leading to a reduction in tumor volume and a prolongation in the survival of the animals. The modulation of the BBB permeability using plasmonic nanocarriers is not novel. The authors (Ref. 17) as well as other groups (doi: 10.1039/c3nr06770j;10.1016/j.jconrel.2018.06.013;https://doi.org/10.1073/pnas.1018790108) have shown that plasmonic nanocarriers, in some cases only the laser, can indeed induce a transient opening of the BBB. The pre-clinical evaluation of the anticancer drug using two glioblastoma animal models showing different glioma phenotypes, being the phenotypes supported by human pathophysiology data, is relevant; however, the authors are brief to support experimentally the mechanism of their approach which decreases significantly the enthusiasm of this reviewer.

1- Plasmonic nanocarriers with and without antibodies that target JAM-A. The authors have used JAM-A targeted nanoparticles but it is not clear the importance of JAM-A targeting for the overall effect reported by the authors. It will be important to show whether gold nanoparticles without JAM-A conjugation will have the same effect in the extravasation of taxol. In addition, it is not clear what is the temperature that the laser irradiated region reaches with the 40 mJ/cm². Moreover, the authors should also evaluate what is the accumulation of taxol in animals irradiated without the administration of the JAM-A targeted nanoparticles.

2- Accumulation of the nanoparticles in the brain and in other organs after multiple administration. In the PS5A1 GBM-bearing mice, it is not clear what is the accumulation of nanoparticles after each laser stimuli and how different is the accumulation profile to other regions in the brain. In addition, the authors do not present any data about the effect of the accumulation of the nanoparticles after multiple administrations, in particular in the liver and spleen, which according to the results presented in Fig. S1 are the organs showing higher accumulation of the nanomaterials. It is also not clear what is the degradation profile of the nanoparticles accumulated in those organs.

3- Accumulation of taxol in the brain and the mechanism underlying the reduction of tumor volume. Although the authors quantify the taxol that extravasates by fluorescence, it is not clear the

concentration of the drug that indeed crosses the BBB. Perhaps the authors can quantify the concentration by HPLC or other methodologies. The authors should also clarify whether the reduction of tumor volume is only mediated by the extravasated taxol or by other mechanisms (e.g. activation of the immune system by the heating effect).

4- BBB maturity and extravasation profile in the 73C GEMM mouse model. I wonder if the authors could clarify whether occludin expression is reduced in the tumor core as they observed in human biopsies (Fig. 1). The authors show that the blood vessels in the tumor core are immature likely due to alterations in the expression of ZO-1 at protein level. I wonder if the low staining is due to alterations in the structure of ZO-1 protein or potential artifacts. To further confirm this effect, the reviewer suggests the authors to confirm the decrease in ZO-1 expression by transcriptomic analyses. Moreover, if the blood vessels in this mouse model are immature, and thus leakier, I wonder what is the mechanism behind the opto-activation of the nanoparticles.

5- Leakage of taxol from tumor blood vessels in the absence of BBTB modulation (73C GEMM mouse model). The authors demonstrate in Fig. 5A and 5B that BBTB at day 14 dpi is leaky and allows the extravasation of EZ-link biotin (600 Da) and Evans blue (66 kDa) into the brain parenchyma. Thus, it is not clear why in Fig. S6C the authors do not observe dye (Biotin) leakage into the tumor without BBTB modulation. The authors should also quantify the leakage of taxol in conditions without BBTB modulation (i.e. without laser activation).

6- In the discussion section, the authors should clarify what are the advantages and limitations of this opto-activation strategy relatively to ultrasound and other approaches documented in the literature to open the BBB at specific sites.

Minor issues:

1- In page 4 (first paragraph), I have the impression that reference 13 should be replaced by reference 17.

2- I could not find reference to the wavelength of the laser used.

Headings and figure titles

Results

Residue tumor cells in human GBM show no contrast enhancement and have intact BBTB before developing marginal recurrence

Fig. 1. Human GBM shows marginal recurrence that associated with no initial contrast enhancement and intratumoral BBTB heterogeneity.

OptoBBTB improves drug penetration in diffusely infiltrative PS5A1 GEMM

Fig.2. PS5A1 GEMM is infiltrative and has intact BBTB during disease progression.

Fig. 3. OptoBBTB improves drug penetration to the brain in the infiltrative PS5A1 GEMM.

OptoBBTB improves therapeutic outcome for PS5A1 GEMM

OptoBBTB improves therapeutic outcome for PS5A1 GEMM

Fig. 4. OptoBBTB improves therapeutic outcomes in the infiltrative PS5A1 GEMM

73C GEMM shows robust angiogenesis and immature dysfunctional tumor-associated vessels during disease progression

Fig. 5. 73C GEMM shows heterogeneous loss of BBTB integrity and angiogenesis during disease progression.

OptoBBTB improves drug penetration and therapeutic outcomes in the 73C GEMM

Fig. 6. OptoBBTB improves drug penetration to the brain in the angiogenic 73C GEMM.

Fig. 7. OptoBBTB improves therapeutic outcomes in the angiogenic 73C GEMM.

Discussion

Key Results

The authors report the use of an optical method called 'optoBBTB' to noninvasively increase BBB permeability in a local region of the brain near the surface. The optoBBTB method was described in a previous publication and is comprised of transcranial pulsed laser excitation of gold nanoparticles. Using two mouse models of GBM, the authors show that using optoBBTB followed by paclitaxel leads to smaller tumor volume compared to paclitaxel alone or optoBBTB with vehicle treatment. Survival data was collected, but do not statistically support improved survival based on the analyses shown.

The first model (PS5A1) is shown to have infiltrative histology and maintain tight junctions, whereas the second model (73C) is shown to have an angiogenic histology. While the models behave differently in the absence of treatment (with the angiogenic being much more aggressive), the impact of optoBBTB appears similar in each model.

Overall, the authors show that using pulsed laser excitation of gold nanoparticles followed by paclitaxel treatment, paclitaxel accumulates locally and leads to slower tumor growth.

Significance

In the murine models studied, the noninvasive optoBBTB method is feasible and, with the specific experimental conditions used, increases delivery of Taxol for superficial tumors, leading to prolonged survival. The results build modestly on prior work from the same authors using this system in a non-tumor bearing model.

The specifics of the optoBBTB method does not appear to be a focus of this manuscript, and there is very little description of the method. The choice of gold nanoparticle characteristics (size, surface functionalization, choice of JAM-A target) appear to be pre-determined, and control nanoparticles without targeting were not utilized. While this may represent a modular technology platform wherein the parameters of optoBBTB can be changed to impact drug delivery, the data here only support the utility of a very specific system using a single drug, albeit in 2 histologically distinct models.

In patients, most glial tumors are not at the surface of the brain, and the human skull is significantly thicker than a mouse skull. The authors do not directly address these aspects in terms of potential future applications of this technology.

The significance of the two new GEM models described in this manuscript is unclear. See discussion below.

Data and methodology

While interesting, the patient case that comprises the first section of the results does not clearly relate to the rest of the paper. The patient age, WHO classification, and underlying tumor genetics are not described. The authors state that based on the patient characteristics of a heterogeneous BBTB with leaky and intact regions, they establish two GEMM models. Was this patient the reason for choosing the genes to modify for the tumor models later described? The connection of this part of the results to the rest of the paper is not clear, and the data is entirely observational utilizing only one patient, greatly limiting its generalizability.

Contextually, there have been several publications in the field noting intratumoral heterogeneity and in particular recognizing the phenomenon that the BBB/BBTB retains regions with vascular integrity especially in smaller tumors or at the tumor periphery. (see Sarkaria, *et al. Neuro Oncol* 2018; Nduom *et al. J Neurosurg* 2013; Dubois, *et al., Front Cell Neurosci* 2014; Nagaraja and Lee, *Microcirculation* 2020). The authors assert that 'there has been no independent ICH verification of the BBTB status in this region,' a statement that requires some clarification – do they mean that tissue at the tumor resection margin has never been studied by IHC, or that it has not been investigated? Perhaps consider reviewing work by Jin *et al. Nat Medicine* 2017 where this heterogeneity was probed in some detail. That said, there is value to reiterating the key clinical point that the BBTB is regionally heterogeneous, as it is important for drug delivery. Overall, the section titled '**Residue tumor cells in human GBM show no contrast enhancement and have intact BBTB before developing marginal recurrence**' could be significantly reduced and/or re-framed as introductory.

After this section, the authors describe two distinct tumor models, which are both characterized by BRAFV600E alteration plus Pten loss, and distinguished by having concomitant loss of either p53 or Ink4ab/Arf. The choice of these particular genetic models is confusing (see discussion below), but the authors make a compelling argument that the tumor vasculature is different between the models, with one having an infiltrative and the other having an angiogenic phenotype. However, based on the ordering of the sections/figures, it is difficult to directly compare between the models, which are characterized separately in figures 2 and 5.

Rather than comparing the performance of models as they relate to optoBBTB, the authors make the argument optoBBTB is relevant for each model on its own. Some of these claims are rather observational, making the claim that OptoBBTB improves drug penetration in the absence of a control group (e.g. Figure 3 and Figure 6). In their prior work describing this system (Li et al Nano Letters 2021), gold nanoparticles functionalized with PEG but without a JAM-A targeting antibody were used as controls. Given the different phenotypes between the two models, it would be very relevant to test a control gold nanoparticle that does not target tight junctions, in order to determine if this targeting group is necessary in both infiltrative and angiogenic models of glioma.

It's not clear why the data provided and analyzed in figure 5 and figure 6 differ. In figure 5, JAM-A expression is unchanged when comparing tumor core to margin to contralateral brain, but in figure 6 it is shown to be significantly increased.

Clarity and context

It is admirable and important that the authors examined two engineered murine models of GBM, increasing the impact of this work.

However, the authors report they are establishing these two GEMM models –PS5A1 and 73C— presumably for the first time in this manuscript, and to that end additional description and methodology is warranted. The methods section related to these GEMM models is quite sparse, and there is no validation of the genotype provided. Is there perhaps another paper describing these models from the laboratory of Robert Bachoo that should be cited? If this is indeed the first description of the models, additional information about the generation of PS5A1 cells and 73C cells would be needed in order for other labs to replicate this work. It seems that one was generated using intracranial injection of an AAV vector in a transgenic mouse, and the other through ex vivo infection of astrocytes derived from a transgenic mouse. The subsequent tumor models were all intracranial injection of these mouse cell lines (into a nude mouse as opposed to the native BL6 background). While this is a type of GEM model, some additional context in the body of the paper would be informative, as transplantation of mouse engineered cell lines is quite distinct from a sporadic or de novo tumor when studying the blood-brain barrier.

To this end, readers would also benefit from some discussion benchmarking features of their two new models (1) *Brafv600+/-, Ink4ab/Arf-/-, Pten-/-*, and (2) *Brafv600+/-, p53-/-, Pten-/-*) with other similar models of GBM. (de Vries *et al. Clin Cancer Res* 2010; Kim *et al. Cancer Res* 2012, and others using combinations of Pten, p53 and Ink4ab/Arf4). The main difference in the authors models appears to be the introduction of BRAFV600E mutation, which is seen commonly in pediatric low grade glioma but only rarely in adult high grade glioma, is very rarely seen clinically in conjunction with the other mutations described here. Have the authors shown that BRAFV600+ is contributing to the difference in infiltrative versus angiogenesis between their two models, or would these changes be seen in the absence of BRAFV600+? It would be helpful to provide context/clarity about why these gene combinations were chosen and

Suggested improvements

Major points

- Overall, the manuscript seems to have two major focuses – the GEM models, and the optoBBTB technology. The comparison of the technology in these two models could be powerful in understanding the role of the nanoparticle targeting ligand or whether tumor vascular phenotype impacts the parameters needed to impact therapy, but the current structure of the manuscript does not bring these two topics together in a cohesive way. Ultimately, if the method works equally well in each group, then the relevance of the models for predicting different clinical scenarios is unclear.
- It would increase the impact of this work significantly to test additional optoBBTB parameters. This represents an opportunity to make a link between the vascular phenotypes that the authors nicely describe and the nanoparticle targeting groups. This was attempted in S7, but the authors state that the other targeting groups did not improve efficiency compared to BV11. This data is not quantified or discussed further, but does not appear to have been tested in the setting of efficacy. An alternative direction to consider would be testing different tumor locations. While not directly addressed in this manuscript, the very superficial nature of the implanted tumors and the transcranial nature of optoBBTB appear linked, and one may draw the conclusion that this method is not relevant except for potentially very narrow clinical scenarios.
- Figures 3 and 6 lack a control group. While the authors state that the contralateral side can act as an internal control, the contralateral side does not have a tumor and is not appropriate to use as a control.
- Statistical analysis is sorely lacking, and powering of mouse studies is not described. For example, why were only 2 mice used for figure 7c, 5 mice for 7d, and 7 mice for 7e? There are no statistics performed on the mouse survival curves in Figures 4 and 7 but the text reports an increase in medial survival with percent improvement. Statistical analysis is needed to support this claim. There is no quantification of IVIS imaging in Figure 4C or 7C but the authors report 'marked decrease in tumor size'. Does this data correlate to the quantified tumor volume by histology in 4D and 7D?

- Additional information about animal studies, including euthanasia criteria is important. The authors should clarify why female immunodeficient mice were utilized, since the GEM glioma cell lines were generated in a BL6 background.
- Data analysis, especially with microscopy, is not robustly described. E.g. how were microscopy slides used to calculate area x thickness? What was the thickness of each slice? Was a 3-dimensional mask used? How was the tumor thresholded and was this kept constant for all samples analyzed? Were microscopy settings kept constant between conditions?
- Additional clarity around the choice of mouse models, how they were generated, and additional characterization is needed.
- The authors should describe their method of implanting genetically engineered mouse cells into the brain and the location of injection. The tumors are orthotopic, but also very superficial, to the point that the 73C tumors are primarily exophytic at the time points shown for efficacy studies. Were superficial tumors generated purposefully in order to use transcranial pulsed laser? Would this method be effective for a tumor in another location?
- In the discussion, the authors state that they showed that optoBBTB 'reversibly increases BBTB permeability' but the experiments supporting the reversible nature in these mouse models (Fig S2 and S6) are difficult to interpret. Clearly the same mouse cannot be shown in each image, and the images in S2B and S6B appear to have a much higher baseline signal of biotin. The authors should describe these experiments, including number of mice and analysis methodology in greater detail in order to support the claim of reversibility.
-

Minor points

- The authors state that taxol is 'among the most widely used oncology drugs' but do not provide a reference.
- The statement in the introduction that this investigation provides 'the first definitive evidence of BBTB modulation and therapeutic effect...' is quite broad, as there are many other interpretations of 'BBTB modulation' including technologies like focused ultrasound and osmotic BBB disruption.
- The authors should carefully review scale bars; Figure 1 C-D are both listed as having the same scale, however the nuclei sizes are visually quite different.
- The authors note data about 73C GEMM survival with TMZ treatment, but data is not shown or cited.
- The characterization of nanoparticles used with OptoBBTB (figure S1) does not make clear whether the radius or diameter is shown, and does not indicate whether z-average or number average is being used to estimate size with dynamic light scattering. In D, it is not clear whether data from AuNP or AuNP-BV11 is shown. There appears to be only one replicate of data in S1 A-D, but presumably multiple batches of NPs were generated for this study.
- The method of GFP transduction in the PS5A1 model is not clear, from the methods an AAV expressing GFP was used to generate the murine cell line, which was later infected with 'lenti-GFP'. Methods here should be expanded. The genotype of each tumor is not confirmed

- There are several minor grammatical errors throughout that can be corrected to improve readability
- The references should be very thoroughly checked by the authors – several references appear to be misnumbered, making it difficult to review this paper. Notably, reference 13 – which in the text refers to the authors prior work describing optoBBTB—is not correct and this reviewer presumes they are instead referring to reference 17 (Li et al Nano Letters 2021).

Response to reviewers

Reviewer #1 (Remarks to the Author): Expert in glioblastoma genetically-engineered mouse models and therapy

This manuscript by Qin and colleagues details work in which they developed two new GEMMs for GBM, an area of unmet clinical need. The GEMMs are based on the BRAFv600E mutation, and despite being an extension of previous work, are useful novel tools for the study of GBM. One model in particular is claimed to be reflective of diffusely infiltrative GBMs, while the second model is more reflective of angiogenic GBMs. Next, the authors developed a treatment strategy based on pulsed laser excitation of vascular-targeted gold nanoparticles to enhance the BBB permeability and increase Taxol entry. This led to improvement in survival of the mice, linked to reduced tumour cell proliferation. The work provides new models and a treatment strategy based on improving BBB permeability which has exciting clinical implications.

However, I do have some concerns which would need to be addressed prior to publication.

Response:

We thank the reviewer for the positive and constructive feedback. We have addressed all raised questions in detail.

1. The authors do not show clear evidence of diffuse single-cell infiltration of tumour cells in the first model. Histology images reflecting this should be shown. Similarly, it is unclear if the BRAF mutation is present in all tumour cells - evidence to this effect should be presented.

Response:

We thank the reviewer for this comment. In our revised manuscript, we provided images of the diffuse single-cell infiltration and the vessel co-option growth patterns in the PS5A1 GEMM. Briefly, we labeled the blood vessels in PS5A1 GEMM (14 dpi) with tomato Lectin 594. The mouse was perfused and fixed with PBS and 4% PFA, followed by cryosectioned to 20 μ m thick coronal slices. The brain slices were imaged with 20 or 40x objectives (Olympus SD-OSR spinning disk super-resolution microscope). We updated Fig. S2 to show the growth patterns in PS5A1 GEMM.

We also added details to clarify the generation of the genetically engineered mouse line. The primary glioma cell line was generated by first generating a multi-allele mouse by crossing the mouse carried conditional floxed tumor suppressor genes (*Pten^{fl/fl}*, *p53^{fl/fl}*, and *INK4a/b^{-/-}Arf^{fl/fl}*) along with the oncogene (*Braf^{V600+/-}*) driver mouse which carries a lox-stop-lox stop codon to generate the compound multi-allele mouse (*Isl.Braf^{V600+/-}*, *Pten^{fl/fl}* and *p53^{fl/fl}*). These mice were purchased from Jackson Labs and genotypes verified in juvenile mice before weaning. These mice have been extensively reported on in the cancer field in furtherance to understanding the fundamental mechanism of cancer. To generate our GEMM, we used primary astrocytes, since this is a presumptive cell of origin from glioma, from new born (postnatal day 1 or 2) pups that are genotype verified (from tail DNA) using the same primers recommended by the vendor of these transgenic mice (Jackson Labs). To activate the oncogene (*Isl.Braf^{V600+/-}*) and delete the tumor suppressor genes (*Pten^{fl/fl}*, *p53^{fl/fl}*, and *INK4a/b^{-/-}Arf^{fl/fl}*), the primary astrocyte cultures are infected with and Adeno-Cre-GFP virus (MOI of 10). The transient infection with the adeno virus infects nearly 100% of the cells (verified by fluorescent GFP), which ensures the expression of Cre-protein (CMV promoter driven) and a complete excision of the *Isl*-STOP codon to activate the mutant *Braf^{V600}* from its

endogenous promoter. Once the oncogene, *Braf*^{V600} has been activated and in combination with tumor suppressors deletion, primary astrocytes undergo a dramatic acceleration (>10x) in growth rate compared with any rare uninfected cells. Primary astrocytes are well known to rapidly senesce under culture conditions (maximum 1-2 passages), so in the event there is a rare un-infected astrocyte and therefore not carry the glioma associated mutations, those cells would be rapidly be senesced during passaging of cells in culture. Moreover, such cells cannot survive or contribute to an expanding tumor mass since non-transformed cells are unable to survive following engraftment. We updated the method to clarify this point.

Revision:

A. Fig. S2, updated.

Fig. S2. Characterization of PS5A1 GEMM. a Characterization of the diffuse single-cell infiltration growth pattern (arrowhead). **b** Characterization of the vessel co-option growth pattern (arrows). In (A-B), the blood vessels are labeled with Lectin594, and the tumor cells are indicated by GFP. Scale bars represent 50 μm .

B. Page 36, method, updated.

- (a) Original: PS5A1 and 73C glioma cells were generated from Dr. Robert Bachoo's laboratory (25, 26). PS5A1 is a highly invasive mouse glioma cell line derived from de novo glioma in the adult BL6 background conditional mouse (*Braf*^{V600E/f/+}; *Ink4ab/Arf*^{f/f}; *Pten*^{f/f}) that was induced by intracranial injection of AAV5-GFAP-Cre-GFP. 73C glioma cells were generated in primary astrocyte cultures from neonatal mice that carried conditional mutations for *Pten*^{f/f}, *p53*^{f/f}, and *LSL**Braf*^{V600E}. These cells were derived from conditional multi-allele primary astrocytes and

infected ex vivo with an adeno-Cre virus to generate a primary transformed cell line. PS5A1 cells were cultured as free-floating neurospheres in DMEM/F12 medium, 2% of B-27, 20 ng/mL of EGF, and 20 ng/mL of FGF2, 20 ng/mL progesterone and 1% insulin transferrin solution. 73C glioma cells were cultured in DMEM containing 10% fetal bovine serum (FBS) and 1% penicillin-Streptomycin.

- (b) Revised: PS5A1 and 73C glioma cells were generated from Dr. Robert Bachoo's laboratory (29, 30). PS5A1 is a highly invasive mouse glioma cell line derived from de novo glioma in the adult BL6 background conditional mouse (*Braf*^{N600Ej/+}; *Ink4ab/Arf*^{ff/ff}; *Pten*^{ff/ff}) that was induced by intracranial injection of AAV5-GFAP-Cre-GFP. 73C glioma cells were generated in primary astrocyte cultures from neonatal mice that carried conditional mutations for *Pten*^{ff/ff}, *p53*^{ff/ff}, and *LSLBraf*^{N600E}. These cells were derived from conditional multi-allele primary astrocytes and infected ex vivo with an adeno-Cre virus to generate a primary transformed cell line. The transient infection with the adeno virus infects nearly 100% of the cells (verified by fluorescent GFP), which ensures the expression of Cre-protein (CMV promoter driven) and a complete excision of the LSL-stop codon to activate the mutant *Braf*^{N600} from its endogenous promoter. Since the AAV transduction provides a transient GFP expression, these cells were further infected with Lenti-GFP and selected by puromycin for stable green fluorescent protein expression. PS5A1 cells were cultured as free-floating neurospheres in DMEM/F12 medium, 2% of B-27, 20 ng/mL of EGF, and 20 ng/mL of FGF2, 20 ng/mL progesterone and 1% insulin transferrin solution. 73C glioma cells were cultured in DMEM containing 10% fetal bovine serum (FBS) and 1% penicillin-Streptomycin.

2. The abnormal vasculature of the second model is clear, however the effects of their treatment strategy on this abnormal vasculature is not presented. It would be important to show how these vessels are changed in response to this treatment, given the vasculature is proposed to play a role in the pathogenesis of these tumours and influence the BBTB.

Response:

We thank the reviewer for this great question. In our revised manuscript, we analyzed the properties of these abnormal vessels in the 73C GEMM after optoBBTB.

We first studied the density changes of these abnormal vessels after the treatment using immunohistochemistry (IHC) staining. Briefly, AuNP-BV11 (37 µg/g) was intravenously delivered to the 73C tumor-bearing mice. 1 hour later, the mice received a single pulse of picosecond laser excitation (40 mJ/cm²). 30 min later, the brains were collected and cryosectioned to 20 µm thickness coronal slices. We used IHC staining to label the blood vessels by CD31, and the blood vessel coverage in the tumor area was visualized using a 10x objective. The results show that before and after optoBBTB, there is no significant difference in the blood vessel density in tumor core and margin (updated Fig. S3a).

We further investigated the influence of optoBBTB on other key junctional proteins such as Claudin-5, VE-Cadherin, Occludin, and JAM-A by IHC staining. As shown in updated Fig. S4b, there is no significant difference in the area fraction ratio of protein over blood vessels (i.e., CLDN5/CD31, VE-Cad/CD31, Occludin/CD31, and JAM-A/CD31) before and after treatment. Therefore, the treatment does not influence the immunofluorescent of the junctional proteins.

In terms of the possible mechanisms, in our recent work (*Nanoscale*, 2023,15, 3387-3397), we demonstrate that laser excitation of vascular-targeting AuNPs is associated with a transient elevation and propagation of Ca²⁺, actin polymerization, and phosphorylation of ERK1/2 (extracellular signal-regulated protein kinase). They collectively activate the cytoskeleton resulting in increased paracellular

permeability. We hypothesize that the increased barrier permeability after optoBBTB is due to the Ca²⁺-mediated activation of the mechanobiological pathways and the re-arrangement of the cytoskeleton. Future work is planned to investigate how the angiogenic vessels in 73C GEMM in response to the laser treatment to increase the BBTB permeability.

In summary, our results suggest that the optoBBTB does not influence the structure of the blood vessels in 73C GEMM, and the possible mechanism for optoBBTB might include Ca²⁺-mediated activation of mechanobiological pathways and cytoskeleton re-arrangement. We updated the figure and text to include the results.

Revision:

A. Page 38, method, updated.

- (a) Original: To immunostaining vascular biomarker (CD31) and junctional proteins (i.e., Claudin-5, ZO-1, VE-cadherin, and JAM-A), the mice brains were snap-frozen on dry ice once quickly removed from the skull and cut to 20 μ m thick coronal slices on a cryostat. The brain slices were fixed for 10 min using ice-cold methanol at -20 °C.
- (b) Revised: To immunostaining vascular biomarker (CD31) and junctional proteins (i.e., Claudin-5, ZO-1, VE-cadherin, **Occludin**, and JAM-A), the mice brains were snap-frozen on dry ice once quickly removed from the skull and cut to 20 μ m thick coronal slices on a cryostat. **To analyze the influence of optoBBTB on vessel density and the immunofluorescent of the junctional proteins, the brains were collected at 30 min after the optoBBTB, followed by cryosectioned to 20 μ m thick coronal slices.** The brain slices were fixed for 10 min using ice-cold methanol at -20 °C.

B. Page 9, result, updated the text.

- (a) Original: IHC staining of junctional proteins showed that the immunofluorescence of Claudin-5, VE-Cadherin, and JAM-A persisted during 7-21 dpi at both tumor core and margin (**Fig. S6D**, **Fig. 5C**). However, there was a significantly lower level of ZO-1 expression at the tumor core at 14 and 21 dpi (**Fig. 5D**), consistent with the observation in human GBM (**Fig. 1E**). Further quantification analysis of the area fraction ratio of protein over blood vessel (CD31) suggested that the relative protein coverage ratio for Claudin-5, VE-Cadherin, and JAM-A was comparable at the tumor core, margin, and contralateral side during 7-21 dpi.
- (b) Revised: IHC staining of junctional proteins showed that the immunofluorescence of Claudin-5, VE-Cadherin, **Occludin**, and JAM-A persisted during 7-21 dpi at both tumor core and margin (**Fig. S4a**, **Fig. 2c**). However, there was a significantly lower level of ZO-1 expression at the tumor core at 14 and 21 dpi (**Fig. 2d**). Further quantification analysis of the area fraction ratio of protein over blood vessel (CD31) suggested that the relative protein coverage ratio for Claudin-5, VE-Cadherin, **Occludin**, and JAM-A was comparable at the tumor core, margin, and contralateral side during 7-21 dpi.

C. Page 16, result, updated the text.

- (a) Original: We subsequently investigated the efficacy of optoBBTB in the 73C GEMM. The overexpression of JAM-A in the tumor area makes the nanoparticles attractive for enhanced GBM accumulation (**Fig. 6A-B**). Inductively coupled plasma mass spectrometry (ICP-MS) analysis shows >80% increase of AuNP-BV11 accumulation in the tumor compared to normal

brain tissue (0.32 ± 0.06 versus 0.20 ± 0.02 %ID/g in tumor and normal brain, respectively, **Fig. 6C**). Biodistribution of the injected particles in other organs is consistent as previously reported (**Fig. S1E**) (17). We further optimized the optoBBTB in the 73C GEMM to obtain the optimal opening efficiency after single-pulse laser stimulation (**Fig. S67A**, **Table S1**). The highest BBTB opening level was achieved by injecting $36\ \mu\text{g/g}$ AuNP-BV11, followed by $40\ \text{mJ}/\text{cm}^2$ laser excitation (1 pulse). The BBTB recovered within 1 day, and no dye leakage into the brain was observed afterward (**Fig. S7B**).

- (b) Revised: We subsequently investigated the efficacy of optoBBTB in the 73C GEMM. The overexpression of JAM-A in the tumor area due to the formation of angiogenic vessels makes the nanoparticles attractive for enhanced GBM accumulation (**Fig. 4a, b**). ICP-MS analysis showed a $>50\%$ increase of AuNP-BV11 accumulation in the tumor compared with normal brain tissue (0.32 ± 0.06 versus 0.20 ± 0.02 %ID/g in tumor and normal brain, respectively, **Fig. S7a**). The biodistribution of the injected particles in other organs was consistent with previously reported (**Fig. S7a**) (17). We further optimized the optoBBTB in the 73C GEMM to obtain the optimal opening efficiency after single-pulse laser stimulation (**Fig. S7b**, **Table S3**). The highest BBTB opening level was achieved by injecting $36\ \mu\text{g/g}$ AuNP-BV11 and $40\ \text{mJ}/\text{cm}^2$ laser excitation (1 pulse). The BBTB recovered within 1 day, and no dye leakage into the brain was observed afterward (**Fig. S7c**). Since the BBTB in 73C GEMM remained intact at 7 dpi, optoBBTB significantly improved the delivery of both small molecules (EZ-link biotin, 660 Da) and large molecules (Evans blue, 66 kDa) after i.v. injection (**Fig. 4c**), while BBTB modulation using AuNP-PEG did not increase the Taxol646 delivery into the tumor (**Fig. 4d**, **Fig. S7d**). The local temperature measurement showed that the average temperature before and after laser excitation was $32.1\pm 0.2\ ^\circ\text{C}$ and $31.9\pm 0.1\ ^\circ\text{C}$ (**Fig. S7e**), indicating no temperature increase after optoBBTB in 73C GEMM. Similarly, a three-cycle treatment regimen could be used for drug delivery in 73C GEMM (**Fig. 4e**).

We noted that the BBTB modulation displayed a higher efficiency in the PS5A1 GEMM than in the 73C GEMM (**Fig. 3, 4, S6, S7**), although there was a significantly higher AuNP-BV11 accumulation in the tumor core of 73C GEMM compared with PS5A1 GEMM (0.32 ± 0.06 %ID/g, and 0.18 ± 0.03 %ID/g, respectively, **Table S4**). To increase the BBTB opening efficiency in the 73C GEMM, we attempted to functionalize AuNPs with other vasculature targets, such as the anti-vascular endothelial growth factor 2 (VEGFR2) antibody and the anti-transferrin receptor (TfR) antibody, since VEGFR2 and TfR are overexpressed in 73C GEMM (**Fig. S8a, b**). However, these nanoparticles did not improve BBTB opening efficiency compared with AuNP-BV11 (**Fig. S8c, d**). To probe the mechanisms of the optoBBTB, we analyzed the changes in the irregular blood vessels in 73C GEMM after laser stimulation using IHC staining. The blood vessel density analysis showed that optoBBTB did not influence the vessel coverage percentages in the tumor core and margin (**Fig. S3a**). Moreover, no significant difference in the immunofluorescent of junctional protein was observed before and after optoBBTB (**Fig. S4b**). These results suggest that optoBBTB in 73C GEMM did not influence the density or the junctional protein immunofluorescent of the angiogenic blood vessels. In our recent work (35), we demonstrated that laser excitation of vascular-targeting AuNPs was associated with a transient elevation and propagation of Ca^{2+} , actin polymerization, and phosphorylation of ERK1/2 (extracellular signal-regulated protein kinase). They collectively activated the cytoskeleton resulting in increased paracellular permeability. We hypothesize that the increased barrier permeability after optoBBTB is due to the Ca^{2+} -mediated activation of the mechanobiological pathways and the re-arrangement of the cytoskeleton. Moreover, angiogenic blood vessels may respond differently to optoBBTB than normal brain microvasculature. Further investigation may be focused on examining how the blood vessel phenotypes respond to optoBBTB and change the barrier permeability.

D. Fig. S3, updated.

Fig. S3. Blood vessel labeling in 73C GEMM shows denser but not well-perfused vasculature. **a** IHC staining and quantification of blood vessels using CD31 at 7-21 dpi. The cell nuclei are labeled with Hoechst staining (HOE). The scale bars represent 50 μ m. Quantification of blood vessel coverage was performed by CD31 area fraction. N=15 images from 3 mice. **b** A comparison of blood vessels labeling with tomato lectin594 or CD31 at 7 dpi. The cell nuclei are labeled by Hoechst staining (HOE). The scale bars represent 100 μ m. The ratio of cell nuclei to blood vessels was quantified by area fraction. N=15 images from 3 mice. Data in the box and whisker plots are given from the minima to maxima, the bounds of the box represent the 25th percentile and 75th percentile, and the middle line of the box is the median. Quantification analysis was performed with One-way ANOVA followed by Tukey's multiple comparisons test or unpaired Student's two-sided *t* test. Source data are available as a Source Data file.

E. Fig. S4, updated.

Fig. S4. Junctional proteins labeling in 73C GEMM shows no significant changes in Claudin-5, VE-Cadherin, Occludin or JAM-A. **a** IHC staining of Claudin-5 (Cldn5), VE-Cadherin (VE-Cad), and Occludin at 7-21 dpi. The blood vessels are stained with CD31, and the cell nuclei are indicated by Hoechst staining (HOE). The arrow indicates blood vessels, and the arrowhead indicates junctional proteins. The scale bars represent 20 μ m. Quantification analysis of the expression of Claudin-5, VE-Cadherin, and Occludin over CD31 was performed by area fraction. N=15 images from 3 mice. **b** The quantification analysis of the expression of junctional proteins over CD31 before and after optoBBTB by area fraction. N=15 images from 3 mice. Data in the box and whisker plots are given from the minima to maxima, the bounds of the box represent the 25th percentile and 75th percentile, and the middle line of the box is the median. Quantification analysis was performed with One-way ANOVA followed by Tukey's multiple comparisons test (**a**) or unpaired Student's two-sided *t* test (**b**). Source data are available as a Source Data file.

3. The relations of their model to other published GEMMs of GBM are not described, and it would be important to do so given the other important GEMMs available. For example, PMID: 26461091 uses Nf1, Trp53 and Pten mutations to generate GBM, and PMID: 32727536 used EGFRvIII in combination with TSG loss via transposons to produce GBMs. How do the authors model differ from these models which also reflect key characteristics of human GBM pathology and genomics? This needs to be highlighted in Discussion.

Response:

We thank the reviewer for this suggestion. As the reviewer notes in part, p53 and Pten are common tumor suppressors mutations in GBM that are lost in 20% and 40% of clinical tumors, which we incorporate into our models. Other major tumor suppressor frequently lost (>50%) in the GBM and is CDKN2A/B, which in mice has been traditionally referred to as INK4a.*b*^{-/-} Arf^{-/-} (also referred to as p16, 15, p19^{-/-}), and we also use to generate our mouse models. NF1 tumor suppressor loss is seen in about 10-12% of GBM cases. It has been frequently been used in the mouse of GBM (especially the Parada lab, who originally describe this mouse model). The common use of this model is, in part, due to practical considerations, since the mouse NF1 gene sits in close proximity (in cis) to p53, which significantly reduces the complexity of animal husbandry and ensures that loss of p53 is highly likely to be associated with concurrent loss of NF1. It's important to keep in mind, that in order to generate a mouse tumor, it requires at a minimum, 2-genetic hits (Knudson hypothesis), an oncogene (e.g., *EGFR* mutation, *PDGFRA*, *cMET*, *Braf*^{V600E}) in combination with loss of one or more suppressors (*Pten*^{ff}, *p53*^{ff}, and *INK4a/b*^{-/-} *Arf*^{ff}). NF1 is technically a tumor suppressor, however it acts as an oncogene due to the loss of neurofibromin, which activates Ras signaling.

Braf^{V600E} is the most common missense mutation across all cancers (melanoma, lung, colon, thyroid, kidney, and brain), with a prevalence of 5-8% in adult gliomas and >50% in pediatric brain tumors, therefore it is a highly relevant GEM brain tumor model. There are several highly specific *Braf*^{V600E} inhibitors that have shown to be effective for treating melanoma (brain tumors trials are in progress). However, these drugs are only transiently effective (including melanoma brain metastasis) with tumors rapidly developing resistance to *Braf*^{V600E} inhibitors. Therefore, the consideration of Taxol delivery as an alternative agent to target our (*Braf*^{V600+/-}, *PTEN*^{-/-}, *p53*^{-/-}) model is highly relevant (as noted *Braf*^{V600+/-} inhibitor clinical trials for brain tumors are ongoing). In addition, one of the major advantages of our combination of mutations is that while there is significant diversity of oncogenes that are seen in GBM, they share one common important feature that they all to some extent activate a common downstream RAS/RAF/MEK/ERK signaling pathways, which in turn regulates transcriptional networks to drive tumorigenesis. NF1 loss leads to Ras activation (product of NF1 is a negative regulator of Ras), while the *Braf*^{V600E} constitutively activates Raf which is directly downstream from Ras. As noted above, the *Braf*^{V600E} is one of the most common activating mutations in all cancers, implying that it is capable of

activating a critical regulatory step in the process of malignant transformation. Therefore, together with our analysis on their BBTB permeability, our GEMMs reflect the key characteristics of human GBM genomics and represent a relevant in vivo model system.

We updated the discussion and added references to clarify this point.

Revision:

A. Page 5, introduction, updated the text.

- (a) Original: Taxol is among the most widely used oncology drug because of its proven efficacy in multiple cancer subtypes, but it was abandoned for GBM treatment following its failure in early-phase clinical trials due to poor brain penetration (21-24). This study reveals that repeated cycles of optoBBTB coupled with systemic administration of Taxol suppress tumor growth (6 and 2.4-fold) by reducing tumor cell proliferation and increasing cell death, resulting in significantly improved median survival (50% and 33% increase) in the infiltrative (PS5A1) and angiogenic (73C) models, respectively.
- (b) Revised: Taxol is among the most widely used oncology drug because of its proven efficacy in multiple cancer subtypes, but it was abandoned for GBM treatment following its failure in early-phase clinical trials due to poor brain penetration (21-24). Moreover, although several highly specific *Braf*^{V600E} inhibitors have shown to be effective for treating melanoma (25), these drugs are only transiently effective with tumors (including melanoma brain metastasis) rapidly developing resistance to *Braf*^{V600E} inhibitors (26). Therefore, the consideration of Taxol delivery to our *Braf*^{V600E} models is highly relevant, since *Braf*^{V600E} inhibitor clinical trials for brain tumors are ongoing (27, 28). This study reveals that repeated cycles of optoBBTB coupled with systemic administration of Taxol suppress tumor growth (6 and 2.4-fold) by reducing tumor cell proliferation and increasing cell death, resulting in significantly improved median survival (50% and 33% increase) in the infiltrative (PS5A1) and angiogenic (73C) models, respectively.

Reference:

25. Dummer, R., Queirolo, P., Abajo Guijarro, A. M., Hu, Y., Wang, D., de Azevedo, S. J., Robert, C., Ascierto, P. A., Chiarion-Sileni, V., Pronzato, P., Spagnolo, F., Mujika Eizmendi, K., Liskay, G., de la Cruz Merino, L., & Tawbi, H. (2022). Atezolizumab, vemurafenib, and cobimetinib in patients with melanoma with CNS metastases (TRICOTEL): a multicentre, open-label, single-arm, phase 2 study. *Lancet. Oncol.* **23**, 1145–1155 (2022).
26. Bonfill-Teixidor, E., Iurlaro, R., Handl, C., Wichmann, J., Arias, A., Cuartas, I., Emmenegger, J., Romagnani, A., Mangano, L., Lorber, T., Berrera, M., Godfried Sie, C., Köchl, F., Eckmann, J., Feddersen, R., Kornacker, M., Schnetzler, G., Cicuendez, M., Cordero, E., Topczewski, T. E., Ferres-Pijoan, A., González, J., Martínez-Ricarte, F., Muñoz-Couselo, E., Tabernero, J., Bischoff, J. R., Pettazzoni, P., & Seoane, J. Activity and resistance of a brain-permeable paradox breaker BRAF inhibitor in melanoma brain metastasis. *Cancer. Res.* **82**, 2552–2564 (2022).
27. Lim-Fat, M. J., Song, K. W., Iorgulescu, J. B., Andersen, B. M., Forst, D. A., Jordan, J. T., Gerstner, E. R., Reardon, D. A., Wen, P. Y., & Arrillaga-Romany, I. Arrillaga-Romany, Clinical, radiological and genomic features and targeted therapy in BRAF V600E mutant adult glioblastoma. *J. Neurooncol.* **152**, 515-522 (2021).
28. Wen, P. Y., Stein, A., van den Bent, M., De Greve, J., Wick, A., de Vos, F. Y. F. L., von Bubnoff, N., van Linde, M. E., Lai, A., Prager, G. W., Campone, M., Fasolo, A., Lopez-Martin, J. A., Kim, T. M., Mason, W. P., Hofheinz, R. D., Blay, J. Y., Cho, D. C., Gazzah, A., Pouessel, D., Yachnin, J., Boran, A., Burgess, P., Ilankumaran, P., Gasal, E., & Subbiah, V. Dabrafenib plus

trametinib in patients with BRAF^{V600E}-mutant low-grade and high-grade glioma (ROAR): a multicentre, open-label, single-arm, phase 2, basket trial. *Lancet. Oncol.* **23**, 53–64 (2022).

B. Page 30, discussion, updated the text and reference.

- (a) Original: These GEMMs include loss of critical tumor suppressor genes (*Pten*^{-/-} and *p53*^{-/-}; or *Pten*^{-/-} and *Ink4ab/Arf*^{-/-}) that are seen in virtually all human GBM. While the *Braf*^{V600E} activating mutation is only seen in 5-7% of adult gliomas and is one of the most common mutations in pediatric gliomas, it is known to be a powerful activator of the mitogen-activated protein kinase (MAPK)/ERK signaling pathways, which is almost ubiquitously activated by any number of mutations (e.g., NF1 loss, EGFR mutations). Thus, in combination with PTEN loss, leading to activation of the phosphatidylinositol 3-kinase (PI3K/AKT) pathway, our GEMMs are driven by a powerful co-activation of both the AKT and ERK signaling pathways which are ubiquitously activated in GBM (26) Taken together, we suggest that our GEMMs represent a relevant in vivo model system for testing drug delivery following BBTB disruption.
- (b) Revised: To generate a mouse glioma, it typically required the activation of an oncogene (e.g., *EGFR* mutation, *PDGFRA*, *cMET*, *Braf*^{V600E}) in combination with loss of one or more suppressors (*Pten*^{fl/fl}, *p53*^{fl/fl}, and *INK4a/b*^{-/-}*Arf*^{fl/fl}). Our GEMMs include loss of critical tumor suppressor genes (*Pten*^{-/-} and *p53*^{-/-}; or *PTEN*^{-/-} and *Ink4ab/Arf*^{-/-}) that are seen in virtually all human GBM. While the *Braf*^{V600E} activating mutation is only seen in 5-7% of adult gliomas and is one of the most common mutations in pediatric gliomas, it is known to be a powerful activator of the mitogen-activated protein kinase (MAPK)/ERK signaling pathways, which is almost ubiquitously activated by any number of mutations (e.g., NF1 loss, EGFR mutations). Thus, in combination with PTEN loss, leading to activation of the phosphatidylinositol 3-kinase (PI3K/AKT) pathway, our GEMMs are driven by a powerful co-activation of both the AKT and ERK signaling pathways which are ubiquitously activated in GBM (26). In addition, one of the major advantages of our combination of mutations is that while there is significant diversity of oncogenes that are seen in GBM which has influenced the choice to mutations that are selected to generate GEMM models (48-51), they share one common important feature that they all to some extent activate a common downstream RAS/RAF/MEK/ERK signaling pathway, which in turn regulates transcriptional networks to drive tumorigenesis. NF1 loss leads to Ras activation (product of NF1 is a negative regulator of RAS), while the *Braf*^{V600E} constitutively activates Raf which is directly downstream from Ras. As noted above, the *Braf*^{V600E} is one of the most common activating mutations in all cancers, implying that it is capable of activating a critical regulatory step in the process of malignant transformation. Taken together, we suggest that our GEMMs represent a relevant in vivo model system for testing drug delivery following BBTB disruption.

Reference:

48. Alcantara Llaguno, S. R., Wang, Z., Sun, D., Chen, J., Xu, J., Kim, E., Hatanpaa, K. J., Raisanen, J. M., Burns, D. K., Johnson, J. E., & Parada, L. F. Adult lineage-restricted CNS progenitors specify distinct glioblastoma subtypes. *Cancer cell.* **28**, 429–440 (2015).
49. Noorani, I., de la Rosa, J., Choi, Y. H., Strong, A., Ponstingl, H., Vijayabaskar, M. S., Lee, J., Lee, E., Richard-Londt, A., Friedrich, M., Furlanetto, F., Fuente, R., Banerjee, R., Yang, F., Law, F., Watts, C., Rad, R., Vassiliou, G., Kim, J. K., Santarius, T., Bradner, S., & Bradley, A. PiggyBac mutagenesis and exome sequencing identify genetic driver landscapes and potential therapeutic targets of EGFR-mutant gliomas. *Genome. Biol.* **21**, 181 (2020).
50. de Vries, N. A., Bruggeman, S. W., Hulsman, D., de Vries, H. I., Zevenhoven, J., Buckle, T., Hamans, B. C., Leenders, W. P., Beijnen, J. H., van Lohuizen, M., Berns, A. J., & van Tellingen,

O. Rapid and robust transgenic high-grade glioma mouse models for therapy intervention studies. *Clin. Cancer Res.* **16**, 3431–3441 (2010).

51. Kim, H. S., Woolard, K., Lai, C., Bauer, P. O., Maric, D., Song, H., Li, A., Kotliarova, S., Zhang, W., & Fine, H. A. Gliomagenesis arising from Pten- and Ink4a/Arf-deficient neural progenitor cells is mediated by the p53-Fbxw7/Cdc4 pathway, which controls c-Myc. *Cancer Res.* **72**, 6065–6075 (2012).

4. Descriptions of how the GEMMs were established should be made clearer - for example, it appears the cell transplants were done in nude mice but this is not apparent in main body of the text.

Response:

We thank the reviewer for this comment. According to other reviewer's comments, we re-framed the structure of the manuscript. We added a description of how the GEMMs were established in the first section in the revised manuscript.

Revision:

A. Page 6, result, updated the text.

Revised:

To recapitulate these features in preclinical models, we characterized two GEMMs in terms of their BBTB integrity, tumor progression patterns, and TJ properties. These GEMMs were generated using neural-stem-cell-derived PS5A1 (*Braf*^{V600+/+}, *Ink4ab/Arf*^{-/-}, *Pten*^{-/-}) and astrocyte-derived glioma cell line 73C (*Braf*^{V600E}, *Pten*^{-/-}, *p53*^{-/-}) (29, 30). These cell lines were engineered to express green fluorescent protein (GFP). We first established PS5A1 and 73C GEMMs in female nude mice (Nu/J, 002019, age 7 weeks, the Jackson Laboratory) and evaluated their BBTB permeability. Specifically, 368 nL of PS5A1 glioma cell suspension or 92 nL of 73C glioma cells ($2 \times 10^5/\mu\text{L}$) was constantly injected into the mouse cortex (-1 mm, -1 mm, 0.5 mm) using a nanoinjector equipped with a glass micropipette (50 μm tip, see method for details). Although the cell line was generated in a BL6 background, we used immunodeficient mice to make the results comparable to the existing literature since most glioblastoma treatment studies used immunodeficient mice.

Reviewer #2 (Remarks to the Author): Expert in glioblastoma, nanoparticles, mouse models, and blood-brain barrier permeability

The authors have developed a novel method, optoBBTB, to modulate the barrier's permeability and increase the delivery of paclitaxel, which under normal circumstances is not effective in controlling glioma growth. The authors established two glioma GEMM models, displaying an angiocentric core, and a diffuse infiltrative margin. They then developed nanoparticles activated by pulsed laser excitation to modulate the BBTB to enhance the delivery of paclitaxel. The treatment increased survival by 50% and 33%.

This is an interesting manuscript, yet, the poor quality of many images limits a thorough assessment of their data.

Response:

We thank the reviewer for the detailed and insightful comments. Addressing those helped us significantly improve the manuscript. Please note that according to the comments from other reviewers, we re-arranged the structure of the manuscript and the order of the figures in order to make a better comparison on the two models. Please see below detailed responses to each of the concerns raised by the reviewer.

1. In Fig. 1, it is not indicated where C, and D, originate from. Equally, Fig. 1E is too dark to identify any structures, which moreover are not indicated with arrows.

Response:

We thank the reviewer for this comment. In the updated Fig. S1, we used boxes to show that Fig.S1c was originated from the resected recurrent tumor in Fig. S1b. Fig. S1d was from the tumor core in Fig. S1c. We removed the IHC staining from the revised manuscript.

Revision:

Fig. S1. Human GBM shows marginal recurrence associated with no initial contrast enhancement and intratumoral BBTB heterogeneity. **a** MR image (T1-weighted contrast-enhanced imaging) of a patient with a high-grade glioma who completed the standard treatment, and for 4 years of serial imaging, there was no evidence of recurrence. The yellow arrows show the area of intact BBTB. **b** MR image with gadolinium (T1-weighted contrast-enhanced imaging) of this patient demonstrates the development of new contrast enhancement in the surgical margin. **c, d** Hematoxylin and Eosin (H&E) and MIB-1 staining from resected enhancing tumor in B demonstrate a highly proliferative tumor (MIB-1 80%) with microvascular proliferation. The scale bars represent 20 μm and 50 μm in c and d, respectively.

2. The same problem is evident in Fig. 2. The immunofluorescent panels are too dark, and the authors have not indicated any particular structures analyzed.

Response:

We thank the reviewer for this comment. We re-arranged the previous Fig. 2 and 5 in order to have a better comparison of the two GEMMs. In the revised Fig. 1, we aim to compare the BBTB permeability during disease progression in 73C and PS5A1 GEMMs. We used arrows to indicate the blood vessels, and asterisks to indicate the dye leakage. In the revised Fig. 2, we characterized the vessel properties of PS5A1 and 73C GEMMs including the leakage to small and large molecules, vascular density (perfused – lectin, all – CD31), and junctional proteins. We used arrows indicate blood vessels, and the arrowheads indicate tight junction proteins. We also updated the immunofluorescent panels for better visualization.

Revision:

A. Fig. 1, updated.

Fig. 1. PS5A1 GEMM has intact BBTB, and 73C GEMM shows heterogeneous loss of BBTB integrity during disease progression. **a** Characterization of the BBTB permeability in PS5A1 GEMM using EZ-link biotin (Biotin, red, 660 Da) and Evans blue (EB, yellow, 66 kDa when bound to albumin) at 14, 28, and 42 dpi. The tumor cells express GFP, and the cell nuclei are indicated by Hoechst staining (HOE, blue). The ROIs selected are 1. tumor core, 2. tumor margin, and 3. contralateral side with no tumor. The scale bars represent 1 mm in the top panel and 20 μ m in the bottom panels. **The blood vessels are indicated by arrows.** **b** Characterization of the BBTB permeability in 73C BBTB using EZ-link biotin (Biotin, red) and Evans blue (EB, yellow) at 7-21 dpi. The cell nuclei are indicated by Hoechst staining (HOE, blue). The ROIs selected are 1. tumor core, 2. tumor margin, and 3. contralateral side with no tumor. **The blood vessels are indicated by arrows, and the dye leakage is indicated by asterisks.** The scale bars represent 1 mm in the top panel and 20 μ m in the middle and bottom panels. **c, d** The quantification of biotin and Evans blue coverage in PS5A1 and 73C GEMMs by area fraction. Data are expressed as Mean \pm SD. N=15 images from 3 mice. **Data in the box and whisker plots are given from the minima to**

maxima, the bounds of the box represent the 25th percentile and 75th percentile, and the middle line of the box is the median. Quantification analysis was performed with One-way ANOVA followed by Tukey's multiple comparisons test. n.s. represents no significant difference. Source data are available as a Source Data file.

B. Fig. 2, updated.

Fig. 2. PS5A1 GEMM shows diffuse infiltration and vascular co-option for tumor cells while 73C GEMM shows vascular angiogenesis with loss of ZO-1 coverage. a Blood vessel labeling with tomato

Lectin594 (indicated by arrows) in PS5A1 GEMM at 14 dpi and the quantification of Lectin coverage by area fraction. The scale bars represent 20 μm . N=15 images from 3 mice. **b** Blood vessel labeling with CD31 (indicated by arrows) in PS5A1 GEMM at 14 dpi and the quantification of CD31 coverage by area fraction. The scale bars represent 20 μm . N=15 images from 3 mice. **c, d** IHC staining and quantification analysis of junctional protein JAM-A and ZO-1 in 73C GEMM at 7-21 dpi. The blood vessels are stained with CD31 (red), and the cell nuclei are indicated by Hoechst staining (HOE, blue). The arrows indicate blood vessels, and the arrowheads indicate tight junction proteins. The scale bars represent 20 μm . The quantification of JAM-A and ZO-1 coverage on the blood vessel by area fraction. N=15 images from 3 mice. In **a-d**, data in the box and whisker plots are given from the minima to maxima, the bounds of the box represent the 25th percentile and 75th percentile, and the middle line of the box is the median. The quantification analysis was performed with One-way ANOVA followed by Tukey's multiple comparisons test, n.s. represents no significant difference. Source data are available as a Source Data file.

3. Fig. 3 suffers from the same problems. The panels are too dark, and structures are not highlighted by arrows. A control group of laser stimulation on its own is missing. This figure also indicates a major shortcoming of this manuscript. BBTB is only increased very near the original tumor. The actual problem in humans is to be able to target those cells that are far away from the tumor. These authors do not achieve BBB opening at large distances from the tumor. The tumor at 14 dpi in C is very small, and so are the tumors at 18, 24 dpi. Why are these tumors so small if no treatment has yet been delivered.

Response:

We thank the reviewer for this great question.

In the revised Fig. 3, we updated the fluorescent images for better visualization. In the updated Fig. 3b-d, we used arrows to indicate the tumor area, and asterisks to indicate the dye leakage. Moreover, we added a control group in which the optoBBTB was performed with AuNP-PEG that cannot target the tight junction (Fig. 3c). Briefly, we synthesized 50 nm AuNP and functionalized the particles with mPEG (1kDa, see page 39, updated method). The nanoparticles were delivered to the mice by intravenous injection (18.5 $\mu\text{g/g}$). Then a single picosecond laser pulse was applied (40 mJ/cm^2), followed by the injection of fluorescent Taxol646 (12.5 mg/kg). The Taxol leakage was evaluated using both fluorescent imaging (Fig. 3c) and HPLC (Fig. S6d). The results show that no Taxol penetration is observed under this condition using laser stimulation of AuNP-PEG nanoparticles.

We agree that the actual problem in humans is to be able to target those cells that are far away from the tumor. Indeed, there are many challenges to treat brain tumors, and our methods show advantages in treating superficial brain tumors or areas that can be easily accessed with a light fiber such as in the surgical cavity (Fig. S13). We updated the discussion to add this point and the potential approaches to open the BBTB in the deeper brain.

The PS5A1 tumor shown in Fig. 3d was relatively small since it is an infiltrative tumor, tumor cells migrate along the blood vessels and may evade extensively into the surrounding brain tissue (as shown in updated Fig. S2). Therefore, they appear small in one coronal brain slice.

Revision:

A. Fig. 3, updated.

Fig. 3. OptoBBTB improves drug penetration to the brain in the infiltrative PS5A1 GEMM. a Schematic illustration of optoBBTB. EC: endothelial cell. TJ: tight junction. **b** Delivery of EZ-link biotin

(Biotin) and Evans blue (EB) after optoBBTB using ps laser and AuNP-BV11. The tumor is indicated by GFP fluorescent (arrows), and BBTB opening is characterized by Biotin or EB leakage (asterisks). The scale bar represents 1 mm. **c** OptoBBTB using ps laser and AuNP-PEG does not improve the delivery of fluorescent Taxol (Taxol646). The tumor is indicated by GFP fluorescent (arrow). The scale bar represents 1 mm. **d** Multiple BBTB modulations in the PS5A1 GEMM at 14, 18, and 22 dpi for fluorescent Taxol646 delivery. The tumor cells are indicated by GFP signal (arrows), and Taxol646 leakage is indicated by asterisks. The scale bar represents 1 mm.

B. Fig. S6d, updated.

Fig. S6d. The analysis of Taxol concentration in tumor in the PS5A1 GEMM at 14 dpi. N=3 mice. The tested groups are (1) Taxol administration into PS5A1 GEMM (Taxol only), and (2) optoBBTB with AuNP-PEG followed by Taxol administration. Data are expressed as Mean \pm SD. The quantification analysis was performed by unpaired Student's two-sided *t* test.

C. Page 13, updated the text.

- (a) Original: However, Taxol cannot pass through the BBB, which may partly account for the lack of clinical efficacy. To investigate the impact of multiple BBTB opening during tumor treatment, we co-delivered AuNP-BV11 and Taxol646 to PS5A1 GBM-bearing mice for 3 times with 4 days between treatments.
- (b) Revised: However, Taxol cannot pass through the BBB, which may partly account for the lack of clinical efficacy. To investigate the effectiveness of optoBBTB in Taxol delivery, we first demonstrated that optoBBTB using AuNP-PEG with no specific targeting to TJ protein did not improve the delivery of Taxol Janelia Fluor 646 (Taxol646) into the tumor region (Fig. 3c, Fig. S6d). Next, we performed optoBBTB with AuNP-BV11, followed by the administration of Taxol646 to PS5A1 GEMM for 3 times with 3 days between treatments, to investigate the impact of multiple BBTB openings during tumor treatment

D. P39, Method, updated the text.

- (a) Original: The method to prepare brain vascular-targeting gold nanoparticles (AuNPs-BV11) was adapted from our previously reported method (13). AuNP-TfR and AuNP-VEGFR2 were prepared using a similar approach by replacing BV11 with the anti-TfR antibody and anti-VEGFR2 antibody. The concentration, size distribution, and morphology of the nanoparticles were analyzed using Ultraviolet-Visible Spectroscopy (UV-Vis), Dynamic Light Scattering (DLS), and Transmission Electron Microscopy (TEM).

- (b) Revised: The method to prepare brain vascular-targeting gold nanoparticles (AuNPs-BV11) was adapted from our previously reported method (13). AuNP-TfR and AuNP-VEGFR2 were prepared using a similar approach by replacing BV11 with the anti-TfR antibody and anti-VEGFR2 antibody. The control nanoparticles were prepared by mixing 50 nm AuNP with mPEG-thiol (PEG: polyethylene glycol, 1 kDa) with a thiol: AuNP=300:1 molar ratio for 3 hour on ice. The particles were washed three times in pure water. The concentration, size distribution, and morphology of the nanoparticles were analyzed using Ultraviolet-Visible Spectroscopy (UV-Vis), Dynamic Light Scattering (DLS), and Transmission Electron Microscopy (TEM).

E. Page 33, discussion, updated the text.

- (a) Original: Second, while light can be delivered transcranially in the mouse brain, fiber delivery to the human brain is envisioned, especially after surgical removal of the primary tumor. Placing an optical fiber in the tumor surgical cavity would allow the delivery of side-emitting light to the tumor margin that is far from the tumor mass (Fig. S11).
- (b) Revised: Among the various methods to change the BBTB permeability, optoBBTB presents unique opportunities for further preclinical and clinical investigations. In preclinical settings, our method can be useful as a drug development and screening platform (optoBBTB and GEMMs) for testing a class of potent anticancer drugs. In terms of clinical investigations, there are several opportunities for further development. First, the 532 nm light exploited in the current study enables light delivery to the mouse cortex and therefore, the treatment of cortically-located tumors. However, to treat brain tumor in the deep brain region, near-infrared light-absorbing nanoparticles can be exploited since the light in this region exhibits deeper tissue penetration to cover the tumor margin in a larger animal model. Second, while light can be delivered transcranially in the mouse brain, fiber delivery to the human brain is envisioned, especially after the surgical removal of the primary tumor. Placing an optical fiber in the tumor surgical cavity would allow the delivery of side-emitting light to the tumor margin far from the tumor mass (Fig. S13). Moreover, since human skulls are significantly thicker than mice skulls, extracranial light delivery is within the realm of implementation with a transparent cranial window to replace a portion of the skull (63). Furthermore, our recent work investigated opening the blood-spinal cord which represent another important application of the technology and has advantages compared with the state-of-the-art methods (64). Further work is ongoing to investigate tumor treatment in this area.

Reference:

64. Gao, Z., David, E. T., Leong, T. W., Li, X., Cai, Q., Mwirigi, J., Giannotta, M., Dejana, E., Wiggins, J., Krishnagiri, S., Bachoo, R. M., Price, T. J., & Qin, Z. *bioRxiv* 2022.05.20.492752 (2022).

4. In Fig. 4, the increased survival achieved is very good. However, the location of all fluorescent images shown in Fig. 4 are not indicated, and thus, most of the figure is hard to evaluate.

Response:

We thank the reviewer for this comment. In the updated Fig. 5, we revised and included the location of these fluorescent images that were taken using boxes.

Revision:

Fig. 5. OptoBBTB improves therapeutic outcomes in the infiltrative PS5A1 GEMM. **a** Schematic illustration of the treatment timeline. **b** OptoBBTB facilitates the delivery of fluorescent Taxol646 to the tumor core and margin. The scale bar represents 20 μm . **c** Quantification of Taxol delivery by fluorescent

area fraction. For each group, N=10 images from 3 mice. Data are expressed as Mean \pm SD. **d** The analysis of Taxol concentration in the tumor without or with optoBBTB at 14 dpi. N=3 mice. Data are expressed as Mean \pm SD. **e** The analysis of tumor volume by GFP fluorescent signal at 42 dpi. N=5 mice in each group. Data are expressed as Mean \pm SD. **f** Kaplan-Meier survival analysis. N=7 mice in each group. **g, h** Tumor size imaging by GFP fluorescent, and the quantification of GFP fluorescent using Living Image® Software for IVIS® Lumina III In Vivo Imaging System. N=5 mice in each group. Data are expressed as Mean \pm SD. **i** Top: Tumor area indicated by GFP fluorescent at 42 dpi. The scale bar represents 1 mm. Middle: Ki67 staining shows cell proliferation. Bottom: TUNEL staining indicates cell apoptosis. The scale bars represent 20 μ m. The ki67 and TUNEL images were taken from the boxes in the top lane. **j, k** Quantification of Ki67 staining and TUNEL staining after treatments. N=10 images from 3 mice. Data are expressed as Mean \pm SD. **l** The record of body weight change during 0-42 dpi in PS5A1 GBM treatment. N=5 mice in each group. Data are expressed as Mean \pm SD. Quantification analysis in **c** and **d** was performed with unpaired Student's two-sided *t* test, in **e, h, j, k,** and **l** was performed with One-way ANOVA followed by Tukey's multiple comparisons test, and in **f** was with logrank test. Source data are available as a Source Data file.

5. Fig. 5 is uninterpretable and needs to be imaged again. All panels are too dark to identify any particular structures.

Response:

We thank the reviewer for this comment. In the revised manuscript, Fig. 5 has been re-organized into Fig. 1 and 2. We updated all the fluorescent panels for better visualization.

6. Fig. 6 has the exact same problems as Fig. 5.

Response:

We thank the reviewer for this comment. In the revised Fig. 4, we updated all the fluorescent images. We also used arrows to indicate the tumor area and asterisks to indicate the dye leakage.

Revision:

Fig. 4. OptoBBTB improves drug penetration to the brain in the angiogenic 73C GEMM. **a** IHC staining shows overexpression of JAM-A in the 73C GEMM at 7 dpi. The cell nuclei are indicated by Hoechst staining (HOE). The scale bars represent 1 mm. **b** Quantification of JAM-A expression in the normal brain (contralateral side) and the tumor by mean fluorescent intensity. $N=15$ images from 3 mice. Data in the box and whisker plots are given from the minima to maxima, the bounds of the box represent the 25th percentile and 75th percentile, and the middle line of the box is the median. The quantification analysis was performed with unpaired Student's two-sided t test. **c** optoBBTB with ps laser and AuNP-

BV11 allows the delivery of small molecule EZ-link biotin (660 Da) and large molecule Evans blue (66 kDa, albumin-bound) to the tumor. The tumor is indicated by Hoechst staining of the cell nuclei (HOE, arrow). The scale bars represent 1 mm. **d** OptoBBTB using ps laser and AuNP-PEG does not improve the delivery of fluorescent Taxol (Taxol646). The tumor is indicated by Hoechst staining of the cell nuclei (HOE, arrow). The scale bar represents 1 mm. **e** Multiple BBTB modulations in the 73C GEMM at 4, 8, and 12 dpi. AuNP-BV11 and Taxol646 were injected intravenously. The cell nuclei are indicated by Hoechst staining (HOE, arrow), and Taxol646 leakage is indicated by asterisks. The scale bars represent 1 mm. Source data are available as a Source Data file.

7. Fig. 7 shows an acceptable survival increase. Yet, the anatomical provenance of all fluorescent images is not indicated. And all fluorescent images are too dark to interpret. The TUNEL images, in I, do not show what the authors wish to interpret. It is unclear how quantification was performed.

Response:

We thank the reviewer for this comment. In the revised Fig. 6, we indicated the anatomical location of the fluorescent images using boxes and updated the images for better visualization.

The TUNEL in i was performed with a ApopTag® Red In Situ Apoptosis Detection Kit. It can specifically detect DNA cleavage and chromatin condensation associated with cell apoptosis. Positive ApopTag® results reveal in situ staining inside early apoptotic nuclei and apoptotic bodies.

The quantification analysis in j and k was performed by counting the signal positive (ki67+ or TUNEL+) cell numbers and the total cell numbers using Image-J, and calculating the percentage of the signal positive (ki67+ or TUNEL+) cells. This information has been added to the text.

Revision:

A. Page 21, results, updated.

- (a) Original: Ki67 staining and cell apoptosis analysis showed that optoBBTB+Taxol decreased cell proliferation and increased cellular apoptosis compared with the other groups (**Fig. 5I** middle-bottom, **J, K**).
- (b) Revised: Ki67 staining and cell apoptosis analysis was performed by calculating the signal positive (ki67+ or TUNEL+) cell numbers over total cell numbers. The results showed that optoBBTB+Taxol decreased cell proliferation and increased cellular apoptosis compared with the other groups (**Fig. 5i** middle-bottom, **j, k**).

B. Fig. 6, updated.

Fig. 6. OptoBBTB improves therapeutic outcomes in the angiogenic 73C GEMM. **a** schematic illustration of the treatment timeline. **b** OptoBBTB facilitates the delivery of fluorescent Taxol646 to tumors (7 dpi). The scale bars represent 20 μm . **c** Quantification of taxol delivery by fluorescent area

fraction. N=10 images from 3 mice. Data are expressed as Mean \pm SD. **d** The analysis of Taxol concentration in the tumor without or with optoBBTB at 7 dpi. N=3 mice. Data are expressed as Mean \pm SD. **e** The analysis of tumor volume was measured by MRI at 15 dpi. Each dot represents one animal. N=5 mice in each group. Data are expressed as Mean \pm SD. **f** Kaplan-Meier survival analysis, N=7 mice in each group. **g, h** Tumor size imaging by GFP fluorescent, and the quantification of GFP fluorescent using Living Image® Software for IVIS® Lumina III In Vivo Imaging System. N=5 mice in each group. Data are expressed as Mean \pm SD. **i** Top: Tumor size indicated by fluorescent imaging at 15 dpi using Hoechst staining (HOE) of cell nuclei. The scale bars represent 1 mm. Middle: Ki67 staining showing cell proliferation. The scale bars represent 20 μ m. Bottom: TUNEL staining indicates cell apoptosis. The scale bars represent 20 μ m. The ki67 and TUNEL images were taken from the boxes in the top lane. **j, k** Quantification of Ki67 staining and TUNEL staining after treatment. N=10 images from 3 mice. Data are expressed as Mean \pm SD. **l** The record of body weight change during 0-15 dpi in 73C GEMM treatment. Data are expressed as Mean \pm SD. N=5 mice in each group. Quantification analysis in **c** and **d** was performed with unpaired Student's two-sided *t* test, in **e, h, j, k,** and **l** was performed with One-way ANOVA followed by Tukey's multiple comparisons test, and in **f** was with logrank test. Source data are available as a Source Data file.

8. In summary, the authors need to redo all fluorescent panels in their figures when resubmitting the manuscript. The authors should update their Figures, including clear indications of where each high power magnification figure originates from.

Response:

We thank the reviewer for the careful evaluation. All the comments have been addressed in the revised manuscript.

9. In the Discussion, the authors need to tone down their assessment of the impact of their approach to open the BBB. Opening of the BBB in these experiments was restricted to the main tumor area, and maybe, a little bit further out. However, in human GBM the cells can be centimeters away from the main tumor mass, much farther than any drug penetration shown in this manuscript.

Response:

We agree with the reviewer that there are limitations to treat GBM cells that are far away from the brain tumor mass. Our method is suitable for treating superficial brain tumors and cells that are around the surgical cavity (Fig. S13), and can be useful as a drug development and screening platform (optoBBTB and GEMMs) for testing a class of potent anticancer drugs. In our revised manuscript, we updated the discussion to include this point.

Revision:

Page 33, discussion, updated the text.

- (a) Original: Among the various methods to change the BBTB permeability, optoBBTB presents unique opportunities for further development and clinical translation. First, near-infrared light-absorbing nanoparticles can be exploited since the light in this region exhibits deeper tissue penetration to cover the tumor margin in a larger animal model. Second, while light can be delivered transcranially in the mouse brain, fiber delivery to the human brain is envisioned, especially after surgical removal of the primary tumor. Placing an optical fiber in the tumor surgical cavity would allow the delivery of side-emitting light to the tumor margin (Fig. S10).

Moreover, extracranial light delivery is within the realm of implementation with a transparent cranial window to replace a portion of the skull (43). All these developments will facilitate the next-stage translation of optoBBTB for GBM treatment going forward.

- (b) Revised: Among the various methods to change the BBTB permeability, optoBBTB presents unique opportunities for further preclinical and clinical investigations. In preclinical settings, our method can be useful as a drug development and screening platform (optoBBTB and GEMMs) for testing a class of potent anticancer drugs. In terms of clinical investigations, there are several opportunities for further development. First, the 532 nm light exploited in the current study enables light delivery to the mouse cortex and therefore, the treatment of cortically-located tumors. However, to treat brain tumor in the deep brain region, near-infrared light-absorbing nanoparticles can be exploited since the light in this region exhibits deeper tissue penetration to cover the tumor margin in a larger animal model. Second, while light can be delivered transcranially in the mouse brain, fiber delivery to the human brain is envisioned, especially after the surgical removal of the primary tumor. Placing an optical fiber in the tumor surgical cavity would allow the delivery of side-emitting light to the tumor margin far from the tumor mass (Fig. S13). Moreover, since human skulls are significantly thicker than mice skulls, extracranial light delivery is within the realm of implementation with a transparent cranial window to replace a portion of the skull (63). Furthermore, our recent work investigated opening the blood-spinal cord which represent another important application of the technology and has advantages compared with the state-of-the-art methods (64). Further work is ongoing to investigate tumor treatment in this area.

Once Figures have been improved as suggested, the manuscript could be resubmitted for further evaluation.

Response:

We believe we were able to thoroughly address the reviewers' concerns and hope you find the revised manuscript suitable for further evaluation.

Reviewer #3 (Remarks to the Author): Expert in nanoparticles, brain tumours, and blood-brain barrier permeability

Key Results

The authors report the use of an optical method called ‘optoBBTB’ to noninvasively increase BBB permeability in a local region of the brain near the surface. The optoBBTB method was described in a previous publication and is comprised of transcranial pulsed laser excitation of gold nanoparticles. Using two mouse models of GBM, the authors show that using optoBBTB followed by paclitaxel leads to smaller tumor volume compared to paclitaxel alone or optoBBTB with vehicle treatment. Survival data was collected, but do not statistically support improved survival based on the analyses shown.

Response:

We thank the reviewer for the careful evaluation of our manuscript and the valuable comments. In our revised manuscript, we updated Fig. 5f (PS5A1 GEMM) and 6f (73C GEMM) to include the statistical analysis by log-rank test on the survival data.

Revision:

Fig. R1. Kaplan-Meier survival analysis in updated Fig. 5 and 6. N=7 mice in each group. The statistical analysis was performed with logrank test.

The first model (PS5A1) is shown to have infiltrative histology and maintain tight junctions, whereas the second model (73C) is shown to have an angiogenic histology. While the models behave differently in the absence of treatment (with the angiogenic being much more aggressive), the impact of optoBBTB appears similar in each model.

Overall, the authors show that using pulsed laser excitation of gold nanoparticles followed by paclitaxel treatment, paclitaxel accumulates locally and leads to slower tumor growth.

Significance

In the murine models studied, the noninvasive optoBBTB method is feasible and, with the specific experimental conditions used, increases delivery of Taxol for superficial tumors, potentially leading to prolonged survival. The results build modestly on prior work from the same authors using this system in a non-tumor bearing model.

The specifics of the optoBBTB method does not appear to be a focus of this manuscript, and there is very little description of the method. The choice of gold nanoparticle characteristics (size, surface functionalization, choice of JAM-A target) appear to be pre-determined, and control nanoparticles without

targeting were not utilized. While this may represent a modular technology platform wherein the parameters of optoBBTB can be changed to impact drug delivery, the data here only support the utility of a very specific system using a single drug, albeit in 2 histologically distinct models.

Response:

We thank the reviewer for the comments. In our manuscript we aim to use the pre-determined AuNP-BV11 as an example to show the possibility of modulating the tight junction protein to open the BBTB for drug delivery into the tumor, which has not been explored before. Compared with other reported tight junction modulating approaches such as co-administration of siRNA against claudin-5 and occludin, and exploiting claudins or cadherin inhibitory peptides, our methods can modulate the BBTB locally and reversibly for the delivery of a variety of molecules ranging from 660 Da to 66 kDa, and multiple openings could be achieved. Moreover, we raised the point that developing and validating new treatment strategies with clinically relevant GBM models is a key step to bridge the gap between preclinical efficacy and successful clinical translation. Therefore, we tested the optoBBTB on two clinically relevant GEMMs. Taken together, our platform (optoBBTB and GEMMs) has the potential to be used for testing a class of potent anticancer drugs.

In our revised manuscript, we included BBTB modulation using a control nanoparticle AuNP-PEG in revised Fig. 3c and S6d (for PS5A1 GEMM) and Fig. 4d and S7d (for 73C GEMM). The AuNP-PEG does not target the tight junctions. The result shows that optoBBTB using AuNP-PEG did not enhance the Taxol delivery into the tumor evaluated by both fluorescent imaging and HPLC-MS analysis. We also expanded our method to include more details about the optoBBTB technique.

Revision:

A. Fig. 3c and 4d, updated.

Fig. R2. OptoBBTB using AuNP-PEG does not improve the delivery of fluorescent Taxol (Taxol646) in PS5A1 (Fig. 3c) and 73C GEMM (Fig. 4d). The tumor is indicated by GFP fluorescent in Fig. 3c and Hoechst staining in Fig.4d (arrow). The scale bar represents 1 mm.

B. Fig. S6d and S7d, updated.

Fig. R3. The analysis of Taxol concentration in the plasma and tumor in PS5A1 (S6d) and 73C (S7d). The results show that after optoBBTB with AuNP-PEG followed by Taxol administration, there was no significant difference of the Taxol concentration in the tumor compared to the mice without optoBBTB treatment (Taxol only). N=3 mice. Data are expressed as Mean \pm SD. The quantification analysis was performed by unpaired Student's two-sided *t* test (S6d) or One-way ANOVA followed by Tukey's multiple comparisons test (S7d), respectively.

C. Page 39, method, updated.

- Original: To optimize the BBTB modulation to achieve the highest opening efficiency, different nanoparticle doses and laser fluence conditions were tested with PS5A1 and 73C glioma-bearing mice, as shown in **Table S1** and **S3**.
- Revised: The tumor bearing mice was anesthetized by 2-3% isoflurane (in air) and intravenously administrated AuNP-BV11. 1 hour later, the body double on the scalp was peeled off to expose the skull. One pulse of the picosecond (ps) laser was applied to the tumor region through intact skull. To optimize the BBTB modulation to achieve the highest opening efficiency, different nanoparticle doses and laser fluence conditions were tested with PS5A1 and 73C glioma-bearing mice, as shown in **Table S1** and **S3**.

In patients, most glial tumors are not at the surface of the brain, and the human skull is significantly thicker than a mouse skull. The authors do not directly address these aspects in terms of potential future applications of this technology.

Response:

We thank the reviewer for the valuable comments. We agree that tissue penetration depth will be limited for translational study due to the strong absorption from endogenous chromophores and the scattering from tissue components. The thick human skull is also a significant obstacle for light delivery to the deeper brain. Indeed, superficial tumors is a significant challenge that our method has the potential address. We acknowledge that our method may not reach deep tumors by shining light from the surface, and proposed several approaches to reach the tumors in the deep brain region. We showed that the BBB in the deeper mouse brain region can be modulated using an optical fiber (*Nano Lett.* **2021**, 21, 22, 9805–9815, Figure S9b). Meanwhile, we are exploiting near-infrared (NIR) laser and NIR light-absorbing

nanoparticles to open the BBB since NIR light has deeper penetration into the brain. Moreover, the cranial window is another possible solution to reduce the influence of the thick skull, since it has been widely used for longitudinal imaging of the brain in animals. Lastly, our recent work showed that the spinal cord is a potential area that allow fiberoptic light delivery, while it is currently very challenging for other methods. We updated the discussion to address this aspect.

Revision:

Page 33, Discussion, updated the text.

- (a) Original: Among the various methods to change the BBB permeability, optoBBTB presents unique opportunities for further development and clinical translation. First, near-infrared light-absorbing nanoparticles can be exploited since the light in this region exhibits deeper tissue penetration to cover the tumor margin in a larger animal model. Second, while light can be delivered transcranially in the mouse brain, fiber delivery to the human brain is envisioned, especially after surgical removal of the primary tumor. Placing an optical fiber in the tumor surgical cavity would allow the delivery of side-emitting light to the tumor margin (Fig. S10). Moreover, extracranial light delivery is within the realm of implementation with a transparent cranial window to replace a portion of the skull (43). All these developments will facilitate the next-stage translation of optoBBTB for GBM treatment going forward.
- (b) Revised: Revised: Among the various methods to change the BBB permeability, optoBBTB presents unique opportunities for further preclinical and clinical investigations. In preclinical settings, our method can be useful as a drug development and screening platform (optoBBTB and GEMMs) for testing a class of potent anticancer drugs. In terms of clinical investigations, there are several opportunities for further development. First, the 532 nm light exploited in the current study enables light delivery to the mouse cortex and therefore, the treatment of cortically-located tumors. However, to treat brain tumor in the deep brain region, near-infrared light-absorbing nanoparticles can be exploited since the light in this region exhibits deeper tissue penetration to cover the tumor margin in a larger animal model. Second, while light can be delivered transcranially in the mouse brain, fiber delivery to the human brain is envisioned, especially after the surgical removal of the primary tumor. Placing an optical fiber in the tumor surgical cavity would allow the delivery of side-emitting light to the tumor margin far from the tumor mass (Fig. S13). Moreover, since human skulls are significantly thicker than mice skulls, extracranial light delivery is within the realm of implementation with a transparent cranial window to replace a portion of the skull (63). Furthermore, our recent work investigated opening the blood-spinal cord which represent another important application of the technology and has advantages compared with the state-of-the-art methods (64). Further work is ongoing to investigate tumor treatment in this area.

Reference:

64. Gao, Z., David, E. T., Leong, T. W., Li, X., Cai, Q., Mwirigi, J., Giannotta, M., Dejana, E., Wiggins, J., Krishnagiri, S., Bachoo, R. M., Price, T. J., & Qin, Z. *bioRxiv* 2022.05.20.492752 (2022).

The significance of the two new GEM models described in this manuscript is unclear. See discussion below.

Response:

We thank the reviewer for raising this point. The questions have been addressed in the revised manuscript.

Data and methodology

While interesting, the patient case that comprises the first section of the results does not clearly relate to the rest of the paper. The patient age, WHO classification, and underlying tumor genetics are not described. The authors state that based on the patient characteristics of a heterogeneous BBTB with leaky and intact regions, they establish two GEMM models. Was this patient the reason for choosing the genes to modify for the tumor models later described? The connection of this part of the results to the rest of the paper is not clear, and the data is entirely observational utilizing only one patient, greatly limiting its generalizability.

Response:

We thank the reviewer for this comment. In this study, we aim to use the patient data to show the intratumoral BBTB heterogeneity and the tumor recurrence at the infiltrative and intact tumor margin (Fig. S1), in order to emphasize the necessity for BBTB opening and drug delivery in this region. Since we did not choose the genes to modify based on this patient data, the patient age, WHO classification and underlying tumor genetics are not relevant information for this study design. In the updated manuscript, we re-organized this section to clarify.

To recapitulate human GBM features such as the intratumoral BBTB heterogeneity in preclinical models, we characterized two GEMMs in terms of their BBTB integrity, tumor progression patterns, and TJ properties. The tumor cell lines were generated in Dr. Bachoo's laboratory (Cell Rep. 2017, 18, 961–976; Cell Rep. 2020, 30, 2489–2500). We select the genes to modify since p53 and Pten are common tumor suppressors mutations in GBM that are lost in 20% and 40%, respectively of clinical tumors. Other major tumor suppressor frequently lost (>50%) in the GBM and is CDKN2A/B, which in mice has been traditionally referred to as INK4a.b^{-/-}Arf^{-/-} (also referred to as p16, 15, p19^{-/-}), which we also used to generate the GEMM.

We re-framed the introduction and results section 1 to better illustrate the rationale.

Revision:

- A. Fig. 1 has been moved to Fig. S1.
- B. Page 3-6, Introduction and result section 1, re-framed. The revisions are provided below the following questions.

Contextually, there have been several publications in the field noting intratumoral heterogeneity and in particular recognizing the phenomenon that the BBB/BBTB retains regions with vascular integrity especially in smaller tumors or at the tumor periphery. (see Sarkaria, et al. Neuro Oncol 2018; Nduom et al. J Neurosurg 2013; Dubois, et al., Front Cell Neurosci 2014; Nagaraja and Lee, Microcirculation 2020). The authors assert that 'there has been no independent ICH verification of the BBTB status in this region,' a statement that requires some clarification – do they mean that tissue at the tumor resection margin has never been studied by IHC, or that it has not been investigated? Perhaps consider reviewing work by Jin et al. Nat Medicine 2017 where this heterogeneity was probed in some detail. That said, there is value to reiterating the key clinical point that the BBTB is regionally heterogeneous, as it is important for drug delivery. Overall, the section titled 'Residue tumor cells in human GBM show no contrast enhancement and have intact BBTB before developing marginal recurrence' could be significantly reduced and/or re-framed as introductory.

Response:

We agree with the reviewer. In the revised manuscript, we reduced this section and re-framed into introduction. Relevant references were also provided.

Revision:

A. Page 3, updated the reference.

- (a) Revised: Although GBM can disrupt the integrity of the BBB in the hypoxic and angiogenic core, that the magnitude of this local disruption is nonuniform or insufficient to allow drug penetration in meaningful quantities (6-10). Moreover, evidence suggests that GBM has tumor cells infiltrating into the neighboring tissue without disrupting the BBB, which subsequently drives the inevitable fatal recurrence (11).

Reference:

6. van Tellingen, O., Yetkin-Arik, B., de Gooijer, M. C., Wesseling, P., Wurdinger, T., & de Vries, H. E. Overcoming the blood–brain tumor barrier for effective glioblastoma treatment. *Drug Resist. Updat.* **19**, 1-12 (2015).
7. Sarkaria, J. N., Hu, L. S., Parney, I. F., Pafundi, D. H., Brinkmann, D. H., Laack, N. N., Giannini, C., Burns, T. C., Kizilbash, S. H., Laramy, J. K., Swanson, K. R., Kaufmann, T. J., Brown, P. D., Agar, N. Y. R., Galanis, E., Buckner, J. C., & Elmquist, W. F. Is the blood-brain barrier really disrupted in all glioblastomas? A critical assessment of existing clinical data. *Neuro. Oncol.* **20**, 184–191 (2018).
8. Nduom, E. K., Yang, C., Merrill, M. J., Zhuang, Z., & Lonser, R. R. Characterization of the blood-brain barrier of metastatic and primary malignant neoplasms: Laboratory investigation. *J. Neurosurg.* **119**, 427-433 (2013).
9. Dubois, L. G., Campanati, L., Righy, C., D'Andrea-Meira, I., Spohr, T. C., Porto-Carreiro, I., Pereira, C. M., Balça-Silva, J., Kahn, S. A., DosSantos, M. F., Oliveira, M.deA., Ximenes-da-Silva, A., Lopes, M. C., Faveret, E., Gasparetto, E. L., & Moura-Neto, V. Gliomas and the vascular fragility of the blood brain barrier. *Front. Cell. Neurosci.* **8**, 418. (2014).
10. Nagaraja, T. N., & Lee, I. Y. Cerebral microcirculation in glioblastoma: A major determinant of diagnosis, resection, and drug delivery. *Microcirculation* **28**, e12679 (2021).
11. Belykh, E., Shaffer, K. V., Lin, C., Byvaltsev, V. A., Preul, M. C., & Chen, L. Blood-brain barrier, blood-brain tumor barrier, and fluorescence-guided neurosurgical oncology: delivering optical labels to brain tumors. *Front. Oncol.* **10**, 739 (2020).

B. Page 4, updated the text.

- (a) Original: Here, we report a GBM treatment approach by BBTB modulation followed by chemotherapy in clinically relevant infiltrative and angiogenic GBM models. We first evaluated a human GBM that shows heterogeneous microvascular and BBTB phenotypes, including both leaky and intact BBTB regions. Based on these characteristics, we established primary conditional astrocyte mouse cell lines that carried mutations seen in both adult and pediatric high-grade gliomas (namely, (1) *Braf*^{V600+/-}, *Ink4ab/Arf*^{+/-}, *Pten*^{-/-}, and (2) *Braf*^{V600+/-}, *p53*^{-/-}, *Pten*^{-/-}). In particular, the two genetically engineered mouse models (GEMMs) show diffuse single-cell infiltration through the brain parenchyma (former, PS5A1) or and a rapidly expanding angiogenic mass with limited single-cell infiltration (the latter, 73C), respectively. Together these two GEMMs represent a reasonable facsimile of the GBM tumor-stromal phenotype seen in the clinical setting.

- (b) Revised: Here, we report a GBM treatment approach by BBTB modulation followed by chemotherapy in clinically relevant infiltrative and angiogenic GBM models. We first provided evidence that human GBM shows intratumoral heterogeneous BBTB permeability, including both leaky and intact BBTB regions. To capture these features in preclinical mouse models, we characterized two genetically engineered mouse models (GEMMs) that show diffuse single-cell infiltration through the brain parenchyma (former, PS5A1) or a rapidly expanding angiogenic mass with limited single-cell infiltration (the latter, 73C), respectively. These primary conditional mouse cell lines carried mutations seen in both adult and pediatric high-grade gliomas (namely, (1) *Braf*^{V600+/-}, *Ink4ab/Arf*^{-/-}, *Pten*^{-/-}, for PS5A1 GEMM, and (2) *Braf*^{V600+/-}, *p53*^{-/-}, *Pten*^{-/-}, for 73C GEMM). Together these two GEMMs represent a reasonable facsimile of the GBM tumor-stromal phenotype seen in the clinical setting.

C. Page 5, removed the original section 1, and updated the text.

Revised: We examined a human GBM that shows intratumoral BBTB heterogeneity and recurrence. The patient was treated with standard of care for GBM, including surgical resection and concurrent radiation (60 Gy) and TMZ, followed by 12 monthly cycles of TMZ. At the end of treatment and for 4 years of serial MR imaging, there was no evidence of recurrence (Fig. S1a). However, within 4 months after an unchanged MR scan, the patient developed focal seizures, and a repeat MRI showed a new enhancing mass at the medial tumor margin (Fig. S1b). Biopsy of the mass revealed a classic GBM phenotype with microvascular proliferation and tumor proliferation rate (MIB-1) of 80% (Fig. S1c, d). These results suggest that human GBM shows infiltrative characteristics and marginal recurrence with no initial contrast enhancement (therefore intact BBTB), indicating the clinical need to establish pre-clinical GEMMs that capture these hallmarks to assess the drug efficacy and novel therapeutic strategies accurately.

To recapitulate these features in preclinical models, we characterized two GEMMs in terms of their BBTB integrity, tumor progression patterns, and TJ properties. These GEMMs were generated using neural-stem-cell-derived PS5A1 (*Braf*^{V600+/-}, *Ink4ab/Arf*^{-/-}, *Pten*^{-/-}) and astrocyte-derived glioma cell line 73C (*Braf*^{V600E}, *Pten*^{-/-}, *p53*^{-/-}) (29, 30). These cell lines were engineered to express green fluorescent protein (GFP). We first established PS5A1 and 73C GEMMs in female nude mice (Nu/J, 002019, age 7 weeks, the Jackson Laboratory) and evaluated their BBTB permeability. Specifically, 368 nL of PS5A1 glioma cell suspension or 92 nL of 73C glioma cells ($2 \times 10^5/\mu\text{L}$) was constantly injected into the mouse cortex (-1 mm, -1 mm, 0.5 mm) using a nanoinjector equipped with a glass micropipette (50 μm tip, see method for details). Although the cell line was generated in a BL6 background, we used immunodeficient mice to make the results comparable to the existing literature since most glioblastoma treatment studies used immunodeficient mice. The BBTB integrity of the mice during GBM progression was analyzed using i.v. injection of EZ-link biotin (660 Da) and Evans blue (66 kDa, albumin-bound).

After this section, the authors describe two distinct tumor models, which are both characterized by BRAFV600E alteration plus Pten loss, and distinguished by having concomitant loss of either p53 or Ink4ab/Arf. The choice of these particular genetic models is confusing (see discussion below), but the authors make a compelling argument that the tumor vasculature is different between the models, with one having an infiltrative and the other having an angiogenic phenotype. However, based on the ordering of the sections/figures, it is difficult to directly compare between the models, which are characterized separately in figures 2 and 5.

Response:

We thank the reviewer for this comment. In the revised manuscript, we discussed the rationale of our GEMMs (see response to the above comment). We also reorganized the structure to make a better comparison of the two models. In the revised Fig. 1, we compared the BBTB permeability during disease progression in PS5A1 and 73C GEMM. In the revised Fig. 2, we compared the histopathological characterizations of the two GEMMs.

Revision:

A. Fig. 1, updated.

Fig. 1. PS5A1 GEMM has intact BBTB, and 73C GEMM shows heterogeneous loss of BBTB integrity during disease progression. **a** Characterization of the BBTB permeability in PS5A1 GEMM using EZ-link biotin (Biotin, red, 660 Da) and Evans blue (EB, yellow, 66 kDa when bound to albumin) at 14, 28, and 42 dpi. The tumor cells express GFP, and the cell nuclei are indicated by Hoechst staining (HOE, blue). The ROIs selected are 1. tumor core, 2. tumor margin, and 3. contralateral side with no tumor. The scale bars represent 1 mm in the top panel and 20 μ m in the bottom panels. **The blood vessels**

are indicated by arrows. **b** Characterization of the BBTB permeability in 73C BBTB using EZ-link biotin (Biotin, red) and Evans blue (EB, yellow) at 7-21 dpi. The cell nuclei are indicated by Hoechst staining (HOE, blue). The ROIs selected are 1. tumor core, 2. tumor margin, and 3. contralateral side with no tumor. The blood vessels are indicated by arrows, and the dye leakage is indicated by asterisks. The scale bars represent 1 mm in the top panel and 20 μ m in the middle and bottom panels. **c, d** The quantification of biotin and Evans blue coverage in PS5A1 and 73C GEMMs by area fraction. Data are expressed as Mean \pm SD. N=15 images from 3 mice. Data in the box and whisker plots are given from the minima to maxima, the bounds of the box represent the 25th percentile and 75th percentile, and the middle line of the box is the median. Quantification analysis was performed with One-way ANOVA followed by Tukey's multiple comparisons test. n.s. represents no significant difference. Source data are available as a Source Data file.

Fig. 2, updated.

Fig. 2. PS5A1 GEMM shows diffuse infiltration and vascular co-option for tumor cells while 73C GEMM shows vascular angiogenesis with loss of ZO-1 coverage. **a** Blood vessel labeling with tomato Lectin594 (indicated by arrows) in PS5A1 GEMM at 14 dpi and the quantification of Lectin coverage by area fraction. The scale bars represent 20 μ m. N=15 images from 3 mice. **b** Blood vessel labeling with CD31 (indicated by arrows) in PS5A1 GEMM at 14 dpi and the quantification of CD31 coverage by area fraction. The scale bars represent 20 μ m. N=15 images from 3 mice. **c, d** IHC staining and quantification analysis of junctional protein JAM-A and ZO-1 in 73C GEMM at 7-21 dpi. The blood vessels are stained with CD31 (red), and the cell nuclei are indicated by Hoechst staining (HOE, blue). **The arrows indicate blood vessels, and the arrowheads indicate tight junction proteins.** The scale bars represent 20 μ m. The

quantification of JAM-A and ZO-1 coverage on the blood vessel by area fraction. N=15 images from 3 mice. In a-d, data in the box and whisker plots are given from the minima to maxima, the bounds of the box represent the 25th percentile and 75th percentile, and the middle line of the box is the median. The quantification analysis was performed with One-way ANOVA followed by Tukey's multiple comparisons test, n.s. represents no significant difference. Source data are available as a Source Data file.

Rather than comparing the performance of models as they relate to optoBBTB, the authors make the argument optoBBTB is relevant for each model on its own. Some of these claims are rather observational, making the claim that OptoBBTB improves drug penetration in the absence of a control group (e.g. Figure 3 and Figure 6). In their prior work describing this system (Li et al Nano Letters 2021), gold nanoparticles functionalized with PEG but without a JAM-A targeting antibody were used as controls. Given the different phenotypes between the two models, it would be very relevant to test a control gold nanoparticle that does not target tight junctions, in order to determine if this targeting group is necessary in both infiltrative and angiogenic models of glioma.

Response:

We thank the reviewer for this comment. In the revised manuscript, we synthesized AuNP-PEG that does not target the TJs (page 39, method). We provided data to show optoBBTB with nanoparticle AuNP-PEG did not improve the Taxol delivery into the PS5A1 GEMM (14 dpi) and 73C GEMM (7 dpi), by both fluorescent imaging (Fig. 3c, 4d) and HPLC-MS analysis (Fig. S6d, S7d). The nanoparticle injection dose was 37 $\mu\text{g/g}$ and 18.5 $\mu\text{g/g}$ for PS5A1 and 73C GEMM, respectively. The laser fluence was 40 mJ/cm^2 , 1 pulse. We also re-arranged the manuscript to make the comparison of the two models easier.

Revision:

A. Fig. 3c and 4d, updated.

Fig. R2. OptoBBTB using AuNP-PEG does not improve the delivery of fluorescent Taxol (Taxol646) in PS5A1 (Fig. 3c) and 73C GEMM (Fig. 4d). The tumor is indicated by GFP fluorescent in Fig. 3c and Hoechst staining in Fig.4d (arrow). The scale bar represents 1 mm.

B. Fig. S6d and S7d, updated.

Fig. R3. The analysis of Taxol concentration in the plasma and tumor in PS5A1 (S6d) and 73C (S7d). The results show that after optoBBTB with AuNP-PEG followed by Taxol administration, there was no significant difference of the Taxol concentration in the tumor compared with the mice without optoBBTB treatment (Taxol only, $P=0.7176$ and 0.8257 in PS5A1 and 73C GEMM, respectively). $N=3$ mice. Data are expressed as Mean \pm SD. The quantification analysis was performed by unpaired Student's two-sided t test (S6d) or One-way ANOVA followed by Tukey's multiple comparisons test (S7d), respectively.

It's not clear why the data provided and analyzed in figure 5 and figure 6 differ. In figure 5, JAM-A expression is unchanged when comparing tumor core to margin to contralateral brain, but in figure 6 it is shown to be significantly increased.

Response:

We thank the reviewer for this question. In our updated manuscript, they are now Fig. 2c and 4b, respectively. In Fig. 2c, we used the ratio of JAM-A/CD31 fluorescent, and in Fig. 4b it was JAM-A mean fluorescent intensity. Since the CD31-labeled microvascular density was significantly increased in the tumor core compared with the tumor margin and contralateral brain region (Fig. S3a), therefore the mean JAM-A immunofluorescent (Fig. 4b) in the tumor is increased.

Clarity and context

It is admirable and important that the authors examined two engineered murine models of GBM, increasing the impact of this work.

However, the authors report they are establishing these two GEMM models –PS5A1 and 73C—presumably for the first time in this manuscript, and to that end additional description and methodology is warranted. The methods section related to these GEMM models is quite sparse, and there is no validation of the genotype provided. Is there perhaps another paper describing these models from the laboratory of Robert Bachoo that should be cited? If this is indeed the first description of the models, additional information about the generation of PS5A1 cells and 73C cells would be needed in order for other labs to replicate this work. It seems that one was generated using intracranial injection of an AAV vector in a transgenic mouse, and the other through ex vivo infection of astrocytes derived from a transgenic mouse. The subsequent tumor models were all intracranial injection of these mouse cell lines (into a nude mouse as opposed to the native BL6 background). While this is a type of GEM model, some additional context in the body of the paper would be informative, as transplantation of mouse engineered cell lines is quite distinct from a sporadic or de novo tumor when studying the blood-brain barrier.

Response:

We thank the reviewer for this point. The PS5A1 and 73C cell lines have been reported in other publications by Dr. Robert Bachoo, therefore this is not the first description of the models (Cell Rep. 2017, 18, 961–976; Cell Rep. 2020, 30, 2489–2500). However, in this work, we took a novel direction and investigated the BBTB permeability in these models, which is distinct from the previous work. We have added additional information about these two cells in the revised manuscript.

Revision:

A. Page 6, updated the text.

revised: To recapitulate these features in preclinical models, we characterized two GEMMs in terms of their BBTB integrity, tumor progression patterns, and TJ properties. These GEMMs were generated using neural-stem-cell-derived PS5A1 (*Braf*^{V600+/-}, *Ink4ab/Arf*^{-/-}, *Pten*^{-/-}) and astrocyte-derived glioma cell line 73C (*Braf*^{V600E}, *Pten*^{-/-}, *p53*^{-/-}) (29, 30). These cell lines were engineered to express green fluorescent protein (GFP). We first established PS5A1 and 73C GEMMs in female nude mice (Nu/J, 002019, age 7 weeks, the Jackson Laboratory) and evaluated their BBTB permeability. Specifically, 368 nL of PS5A1 glioma cell suspension or 92 nL of 73C glioma cells ($2 \times 10^5/\mu\text{L}$) was constantly injected into the mouse cortex (-1 mm, -1 mm, 0.5 mm) using a nanoinjector equipped with a glass micropipette (50 μm tip, see method for details). Although the cell line was generated in a BL6 background, we used immunodeficient mice to make the results comparable to the existing literature, since most of the glioblastoma treatment studies used immunodeficient mice. The BBTB integrity of the mice during GBM progression was analyzed using i.v. injection of EZ-link biotin (660 Da) and Evans blue (66 kDa, albumin-bound).

To this end, readers would also benefit from some discussion benchmarking features of their two new models (1) *Braf*^{V600+/-}, *Ink4ab/Arf*^{-/-}, *Pten*^{-/-}, and (2) *Braf*^{V600+/-}, *p53*^{-/-}, *Pten*^{-/-} with other similar models of GBM. (de Vries et al. Clin Cancer Res 2010; Kim et al. Cancer Res 2012, and others using combinations of *Pten*, *p53* and *Ink4ab/Arf4*). The main difference in the authors models appears to be the introduction of BRAFV600E mutation, which is seen commonly in pediatric low grade glioma but only rarely in adult high grade glioma, is very rarely seen clinically in conjunction with the other mutations described here. Have the authors shown that BRAFV600+ is contributing to the difference in infiltrative versus angiogenesis between their two models, or would these changes be seen in the absence of BRAFV600+? It would be helpful to provide context/clarity about why these gene combinations were chosen.

Response:

We thank the reviewer for this comment. We selected these genes to modify since *p53* and *Pten* are common tumor suppressors mutations in GBM that are lost in 20% and 40%, respectively, of clinical tumors, which we incorporate into our models. Other major tumor suppressor frequently lost (>50%) in the GBM and is *CDKN2A/B*, which in mice has been traditionally referred to as *INK4a.b*^{-/-} *Arf*^{-/-} (also referred to as *p16*, *p15*, *p19*^{-/-}) which we also use to generate our mouse models. *NF1* tumor suppressor loss is seen in about 10-12% of GBM cases. It has frequently been used in the mouse of GBM, in part due to practical considerations, since the mouse *NF1* gene sits in close proximity (in cis) to *p53*, which significantly reduces the complexity of animal husbandry and ensures that loss of *p53* is highly likely to be associated with concurrent loss of *NF1*. It's important to keep in mind, that in order to generate a mouse tumor, it requires at a minimum, 2-genetic hits (Knudson hypothesis), an oncogene (e.g., *EGFR* mutation, *PDGFRA*, *cMET*, *Braf*^{V600E}) in combination with loss of one or more suppressors (*PTEN*^{fl/fl},

p53^{ff}, and *INK4a/b^{-/-}Arf^{ff}*). *NF1* is technically a tumor suppressor, however it acts as an oncogene due to the loss of neurofibromin, which activates Ras signaling.

Braf^{V600E} is the most common missense mutation across all cancers (melanoma, lung, colon, thyroid, kidney, brain), with a prevalence of 5-8% in adult gliomas and >50% in pediatric brain tumors, therefore it is a highly relevant GEM brain tumor model. There are several highly specific *Braf^{V600E}* inhibitors that have shown to be effective for treating melanoma brain tumors trials are in progress). However, these drugs are only transiently effective (including melanoma brain metastasis) with tumors rapidly developing resistance to *Braf^{V600E}* inhibitors. Therefore, the consideration of Taxol delivery as an alternative agent to target our (*Braf^{V600E}*, *PTEN^{-/-}*, *p53^{-/-}*) model is highly relevant (as noted *Braf^{V600E}* inhibitor clinical trials for brain tumors are ongoing). In addition, one of the major advantages of our combination of mutations is that while there is significant diversity of oncogenes that are seen in GBM, they share one common important feature that they all to some extent activate a common downstream RAS/RAF/MEK/ERK signaling pathway, which in turn regulates transcriptional networks to drive tumorigenesis. *NF1* loss leads to Ras activation (product of *NF1* is a negative regulator of Ras), while the *Braf^{V600E}* constitutively activates Raf which is directly downstream from Ras. As noted above, the *Braf^{V600E}* is one of the most common activating mutations in all cancers, implying that it is capable of activating a critical regulatory step in the process of malignant transformation.

Due to the "two-hit" (Knudson's) hypothesis which state that normal cells need to have a minimum of 2 mutations, a loss of a tumor suppressor and an activating oncogene, to drive tumor transformation, the primary cells do not transform if there is an absence of *Braf^{V600E}*. We updated the discussion to clarify the selection of the gene combination.

Revision:

Page 30, discussion, updated.

- (a) Original: These GEMMs include loss of critical tumor suppressor genes (*Pten^{-/-}* and *p53^{-/-}*; or *Pten^{-/-}* and *Ink4ab/Arf^{-/-}*) that are seen in virtually all human GBM. While the *Braf^{V600E}* activating mutation is only seen in 5-7% of adult gliomas and is one of the most common mutations in pediatric gliomas, it is known to be a powerful activator of the mitogen-activated protein kinase (MAPK)/ERK signaling pathways, which is almost ubiquitously activated by any number of mutations (e.g., *NF1* loss, EGFR mutations). Thus, in combination with *PTEN* loss, leading to activation of the phosphatidylinositol 3-kinase (PI3K/AKT) pathway, our GEMMs are driven by a powerful co-activation of both the AKT and ERK signaling pathways which are ubiquitously activated in GBM (26) Taken together, we suggest that our GEMMs represent a relevant in vivo model system for testing drug delivery following BBTB disruption.
- (b) Revised: To generate a mouse glioma, it typically required the activation of an oncogene (e.g., *EGFR* mutation, *PDGFRA*, *cMET*, *Braf^{V600E}*) in combination with loss of one or more suppressors (*Pten^{ff}*, *p53^{ff}*, and *INK4a/b^{-/-}Arf^{ff}*). Our GEMMs include loss of critical tumor suppressor genes (*Pten^{-/-}* and *p53^{-/-}*; or *Pten^{-/-}* and *Ink4ab/Arf^{-/-}*) that are seen in virtually all human GBM. While the *Braf^{V600E}* activating mutation is only seen in 5-7% of adult gliomas and is one of the most common mutations in pediatric gliomas, it is known to be a powerful activator of the mitogen-activated protein kinase (MAPK)/ERK signaling pathways, which is almost ubiquitously activated by any number of mutations (e.g., *NF1* loss, EGFR mutations). Thus, in combination with *PTEN* loss, leading to activation of the phosphatidylinositol 3-kinase (PI3K/AKT) pathway, our GEMMs are driven by a powerful co-activation of both the AKT and ERK signaling pathways which are ubiquitously activated in GBM (26). In addition, one of the major advantages of our combination of mutations is that while there is significant diversity of oncogenes that are

seen in GBM which has influenced the choice to mutations that are selected to generate GEM models (48-51), they share one common important feature that they all to some extent activate a common downstream RAS/RAF/MEK/ERK signaling pathway, which in turn regulates transcriptional networks to drive tumorigenesis. NF1 loss leads to RAS activation (product of NF1 is a negative regulator of RAS), while the *Braf*^{V600E} constitutively activates RAF which is directly downstream from RAS. As noted above, the *Braf*^{V600E} is one of the most common activating mutations in all cancers, implying that it is capable of activating a critical regulatory step in the process of malignant transformation. Taken together, we suggest that our GEMMs represent a relevant in vivo model system for testing drug delivery following BBTB disruption.

Reference:

48. S. R. Alcantara Llaguno, Z. Wang, D. Sun, J. Chen, J. Xu, E. Kim, K. J. Hatanpaa, J. M. Raisanen, D. K. Burns, J. E. Johnson, L. F. Parada, Adult lineage-restricted CNS progenitors specify distinct glioblastoma subtypes. *Cancer cell*. **28**, 429–440 (2015).
49. I. Noorani, J. de la Rosa, Y. H. Choi, A. Strong, H. Ponstingl, M. S. Vijayabaskar, J. Lee, E. Lee, A. Richard-Londt, M. Friedrich, F. Furlanetto, R. Fuente, R. Banerjee, F. Yang, F. Law, C. Watts, R. Rad, G. Vassiliou, J. K. Kim, T. Santarius, S. Brandner, A. Bradley, PiggyBac mutagenesis and exome sequencing identify genetic driver landscapes and potential therapeutic targets of EGFR-mutant gliomas. *Genome. Biol.* **21**, 181 (2020).
50. N. A. de Vries, S. W. Bruggeman, D. Hulsman, H. I. de Vries, J. Zevenhoven, T. Buckle, B. C. Hamans, W. P. Leenders, J. H. Beijnen, M. van Lohuizen, A. J. Berns, O. van Tellingen, Rapid and robust transgenic high-grade glioma mouse models for therapy intervention studies. *Clin. Cancer. Res.* **16**, 3431–3441 (2010).
51. H. S. Kim, K. Woolard, C. Lai, P. O. Bauer, D. Maric, H. Song, A. Li, S. Kotliarova, W. Zhang, H. A. Fine, Gliomagenesis arising from Pten- and Ink4a/Arf-deficient neural progenitor cells is mediated by the p53-Fbxw7/Cdc4 pathway, which controls c-Myc. *Cancer. Res.* **72**, 6065–6075 (2012).

Suggested improvements

Major points

1. Overall, the manuscript seems to have two major focuses – the GEM models, and the optoBBTB technology. The comparison of the technology in these two models could be powerful in understanding the role of the nanoparticle targeting ligand or whether tumor vascular phenotype impacts the parameters needed to impact therapy, but the current structure of the manuscript does not bring these two topics together in a cohesive way. Ultimately, if the method works equally well in each group, then the relevance of the models for predicting different clinical scenarios is unclear.

Response:

We thank the reviewer for this great question.

In our revised manuscript, we reorganized the structure to make a better comparison of the two models. As discussed previously, in the revised Fig. 1, we compared the BBTB permeability during disease progression in PS5A1 and 73C GEMM. In the revised Fig. 2, we compared the vasculature and junctional protein characterizations of the two GEMMs. Moreover, in updated Fig. 3-4, we compared the optoBBTB in PS5A1 and 73C GEMMs, respectively. In updated Fig. 5-6, we evaluated the treatment outcomes in the two GEMMs. We believe that the current structure has a better comparison between the two models.

We also updated the manuscript to clarify the relevance of our models for predicting different clinical scenarios. Since the GBM is characterized by a high degree of spatiotemporal heterogeneity, we aim to use PS5A1 and 73C GEMMs to recapitulate GBM margin (infiltrative) and core (angiogenic) characteristics. Driven by high levels of angiogenic signals, GBM cells can induce microvascular proliferation in the tumor core regions, which are irregular structures with poor hemodynamics and limited function (73C GEMM). On the other hand, GBM cells at the tumor margin are characterized by diffuse single-cell infiltration through the brain parenchyma, including neuron-rich regions of grey matter neuropil and along white matter tracts. Here, GBM cells co-opt the pre-existing dense brain microvasculature for metabolic support and nutrient exchange without disrupting the normal structure or functions of the microvessels (PS5A1 GEMM). Therefore, these two clinically relevant GEMMs can recapitulate GBM margin (infiltrative) and core (angiogenic) characteristics.

Our results show that optoBBTB in PS5A1 GEMM had a higher efficiency than in 73C GEMM and we optimized the optoBBTB parameters to achieve the maximum BBTB opening efficiency in these two models (Fig. 3-4, Fig. S6-7). With these optimized optoBBTB parameters, we showed that the Taxol accumulation in the tumor after optoBBTB increased 16 and 5-fold in PS5A1 GEMM and 73C GEMM, respectively. The treatment reduced the tumor volume by 6 and 2.4-fold and prolonged the survival by 50% and 33%, in PS5A1 and 73C GEMM, respectively. We hypothesize that the vessel phenotypes may respond differently to the optoBBTB, resulting in different opening efficiency, since our recent study suggests Ca^{2+} -mediated activation of mechanobiological pathways and cytoskeleton protein re-organization are key parameters regulating BBB permeability (*Nanoscale*, 2023,15, 3387-3397). Further investigation may examine how the normal and angiogenesis blood vessels respond to laser stimulation and change the permeability.

In our revised manuscript, we updated the manuscript to include these above comparisons.

Revision:

A. Page 40, method, updated the text.

- (a) Original: ICP-MS was used to determine the biodistribution of AuNP-BV11 (37 $\mu\text{g/g}$) after intravenous (i.v.) injection to the 73C glioma-bearing mice (7 dpi). 1 hour after nanoparticle injection, the mice were perfused with ice-cold PBS, and the main organs were collected. The tissue was then digested in fresh aqua regia until the tissue was fully dissolved. Then the solution was centrifuged at 5000 g for 10 min, and the supernatant was collected and diluted with ultrapure water for ICP-MS analysis (Agilent 7900).
- (b) Revised: We used ICP-MS to measure the AuNP-BV11 (18.5 $\mu\text{g/g}$) accumulation in PS5A1 GEMM (14 dpi) after each optoBBTB (40 mJ/cm^2 , 1 pulse, repeated 3 times at 14, 18, and 22 dpi). We also studied the nanoparticle degradation profile using healthy Nu/J mice. The mice received 3 nanoparticle injections and laser treatments with 3 days interval, and the gold concentration in each organ were measured at 60 days after the third nanoparticle injection and laser excitation. ICP-MS was also used to determine the biodistribution of AuNP-BV11 (37 $\mu\text{g/g}$) after intravenous (i.v.) injection to the 73C glioma-bearing mice (7 dpi). 1 hour after nanoparticle injection, the mice were perfused with ice-cold PBS, and the main organs were collected. The tissue was then digested in fresh aqua regia until the tissue was fully dissolved. Then the solution was centrifuged at 5000 g for 10 min, and the supernatant was collected and diluted with ultrapure water for ICP-MS analysis (Agilent 7900).

B. Page 17, result, updated the text.

- (a) Original: We noted that the BBTB modulation displayed a higher efficiency in the PS5A1 GEMM than in the 73C GEMM (Fig. 3, 4, S6, S7). To increase the BBTB opening efficiency in the 73C GEMM, we attempted to functionalize AuNPs with other vasculature targets such as the anti-vascular endothelial growth factor 2 (VEGFR2) antibody and the anti-transferrin receptor (TfR) antibody, since VEGFR2 and TfR were overexpressed in 73C GEMM. However, these nanoparticles did not improve BBTB opening efficiency compared to AuNP-BV11 (Fig. S8).
- (b) Revised: We noted that the BBTB modulation displayed a higher efficiency in the PS5A1 GEMM than in the 73C GEMM (Fig. 3, 4, S6, S7), although there was a significantly higher AuNP-BV11 accumulation in the tumor core of 73C GEMM compared to PS5A1 GEMM (0.32 ± 0.06 %ID/g, and 0.18 ± 0.03 %ID/g, respectively, Table S4). To increase the BBTB opening efficiency in the 73C GEMM, we attempted to functionalize AuNPs with other vasculature targets, such as the anti-vascular endothelial growth factor 2 (VEGFR2) antibody and the anti-transferrin receptor (TfR) antibody, since VEGFR2 and TfR are overexpressed in 73C GEMM (Fig. S8a, b). However, these nanoparticles did not improve BBTB opening efficiency compared to AuNP-BV11 (Fig. S8c, d). To probe the mechanisms of the optoBBTB, we analyzed the changes in the irregular blood vessels in 73C GEMM after laser stimulation using IHC staining. The blood vessel density analysis show that optoBBTB did not influence the vessel coverage percentages in the tumor core and margin (Fig. S3a). Moreover, no significant difference in the immunofluorescent of junctional protein was observed before and after optoBBTB (Fig. S4b). These results suggest that optoBBTB in 73C GEMM did not influence the density or the junctional protein immunofluorescent of the angiogenic blood vessels. In our recent work (31), we demonstrated that laser excitation of vascular-targeting AuNPs was associated with a transient elevation and propagation of Ca^{2+} , actin polymerization, and phosphorylation of ERK1/2 (extracellular signal-regulated protein kinase). They collectively activated the cytoskeleton resulting in increased paracellular permeability. We hypothesize that the increased barrier permeability after optoBBTB is due to the Ca^{2+} -mediated activation of the mechanobiological pathways and the re-arrangement of the cytoskeleton. Moreover, angiogenic blood vessels may respond differently to optoBBTB than normal brain microvasculature. Further investigation may be focused on examining how the blood vessel phenotypes respond to optoBBTB and change the barrier permeability.

C. Page 33, discussion, added text.

Revised: We compared the optoBBTB in these two GEMMs regarding nanoparticle targeting and the BBTB opening efficiency. Our results show that after nanoparticle administration, there was a significantly higher gold accumulation (%ID/g) in the tumor core of 73C GEMM compared with PS5A1 GEMM. Therefore, the different blood vessel phenotypes might influence the nanoparticle binding efficiency, probably due to the increased JAM-A expression in the angiogenic 73C GEMM. However, our results show that optoBBTB in PS5A1 GEMM was more efficient than in 73C GEMM, regardless of the nanoparticle targets, for example increasing Taxol delivery by 16-fold vs 5-fold in these two models, respectively. We hypothesize that the blood vessel phenotypes may also respond differently to the mechanobiological activation of the BBB (35). Therefore, PS5A1 GEMM with normal microvasculature might demonstrate a higher optoBBTB efficiency than angiogenic 73C GEMM. Further work is warranted to investigate the effect of vascular phenotype in optoBBTB.

D. Table S4, updated.

Table S4. A comparison of gold concentration in PS5A1 and 73C GEMM. Unit: %ID/g.

	Brain	Tumor	Kidney	Heart	Lung	Spleen	Liver
PS5A1	0.2±0.1	0.18± 0.03	0.4±0.2	1.4±0.7	3.8±0.9	55±8	81±10
73C	0.20± 0.02	0.32± 0.06	0.4±0.1	0.4±0.2	2.2±0.8	33±4	69±9

2. It would increase the impact of this work significantly to test additional optoBBTB parameters. This represents an opportunity to make a link between the vascular phenotypes that the authors nicely describe and the nanoparticle targeting groups. This was attempted in S7, but the authors state that the other targeting groups did not improve efficiency compared to BV11. This data is not quantified or discussed further, but does not appear to have been tested in the setting of efficacy. An alternative direction to consider would be testing different tumor locations. While not directly addressed in this manuscript, the very superficial nature of the implanted tumors and the transcranial nature of optoBBTB appear linked, and one may draw the conclusion that this method is not relevant except for potentially very narrow clinical scenarios.

Response: We thank the reviewer for this great point.

In updated Fig. S8d, we included the statistical analysis to quantify the biotin leakage by mean intensity in the tumor core and the contralateral side (healthy brain). The result shows that in 73C GEMM, the fluorescent of the dye in the tumor core was significantly lower than in the contralateral side, no matter of the targets. As discussed above, our hypothesis is that blood vessels phenotypes may respond differently to the optoBBTB, such as the activation of mechanobiological pathways including the Ca²⁺ generation and propagation, and the re-organization of the cellular skeleton (*Nanoscale*, 2023,15, 3387-3397). Since anti-VEGFR2 antibody and the anti-TfR antibody functionalized nanoparticles did not improve BBTB opening efficiency compared with AuNP-BV11, we did not evaluate their efficacy in drug delivery and tumor treatment.

We agree with the reviewer that testing different brain locations will be an alternative way. In this study, the tumor cells were injected at about 0.5 mm depth in the cortex. We selected this depth because of the light penetration in the mouse brain is largely confined to the cortex. Therefore, the current method shows great advantage in treating superficial tumors. To reach tumors below the cortex, we showed the feasibility of using an optical fiber to open the BBB in the deeper brain region (*Nano Lett.* 2021, 21, 22, 9805–9815). Moreover, to deliver light to the deeper brain region, near-infrared light-absorbing nanoparticles can be exploited since the light in this region exhibits deeper tissue penetration. Therefore, we believe that this method is suitable for superficial tumors but can be optimized to target tumors in different brain locations. Furthermore, our recent work investigated opening the blood-spinal cord which represent another important application of the technology and has advantages compared with the state-of-the-art methods (bioRxiv 2022.05.20.492752). Further work is ongoing to investigate tumor treatment in this area. Therefore, the optical method can find several conditions to address limitations of current approaches. We updated the discussion to address this point.

Revision:

A. Fig. S8, updated.

Fig. S8. BBTB modulation using different targets in the 73C GEMM. **a** IHC staining shows the overexpression of vascular endothelial growth factor receptor 2 (VEGFR2) at 7 dpi. **b** IHC staining shows the overexpression of transferrin receptor (TfR) at 7 dpi. The blood vessels are stained by CD31. **c** A comparison of BBTB modulation efficacy using AuNP-VEGFR2 and AuNP-TfR. The cell nuclei are indicated by Hoechst staining (HOE). The nanoparticle dose is 37 $\mu\text{g/g}$, and the laser fluence is 40 mJ/cm^2 , 1 pulse. A laser pulse was applied to both sides of the brain. The tumor was injected into the left side of the brain, and the right side served as an internal control. The scale bars represent 1 mm in the slide scanner images in (a-c) and 20 μm in zoom-in images in (a, b). **d** The quantification of the biotin mean fluorescent intensity in the tumor core and the contralateral side after optoBBTB using different AuNPs. Data are expressed as Mean \pm SD, N=10 images from 3 mice. **The quantification analysis was performed by unpaired Student's two-sided *t* test. Source data are available as a Source Data file.**

B. Page 17, result, updated the text.

- (a) Original: We noted that the BBTB modulation displayed a higher efficiency in the PS5A1 GEMM than in the 73C GEMM. To increase the BBTB opening efficiency in the 73C GEMM, we attempted to functionalize AuNPs with other vasculature targets such as the anti-vascular endothelial growth factor 2 (VEGFR2) antibody and the anti-transferrin receptor (TfR) antibody,

since VEGFR2 and TfR were overexpressed in 73C GEMM. However, these nanoparticles did not improve BBTB opening efficiency compared to AuNP-BV11 (Fig. S7). One hypothesis is that normal and angiogenesis blood vessels may respond differently to the optoBBTB.

- (b) Revised: We noted that the BBTB modulation displayed a higher efficiency in the PS5A1 GEMM than in the 73C GEMM (Fig. 3, 4, S6, S7), although there was a significantly higher AuNP-BV11 accumulation in the tumor core of 73C GEMM compared with PS5A1 GEMM (0.32 ± 0.06 %ID/g, and 0.18 ± 0.03 %ID/g, respectively, Table S4). To increase the BBTB opening efficiency in the 73C GEMM, we attempted to functionalize AuNPs with other vasculature targets, such as the anti-vascular endothelial growth factor 2 (VEGFR2) antibody and the anti-transferrin receptor (TfR) antibody, since VEGFR2 and TfR are overexpressed in 73C GEMM (Fig. S8a, b). However, these nanoparticles did not improve BBTB opening efficiency compared with AuNP-BV11 (Fig. S8c, d). To probe the mechanisms of the optoBBTB, we analyzed the changes in the irregular blood vessels in 73C GEMM after laser stimulation using IHC staining. The blood vessel density analysis show that optoBBTB did not influence the vessel coverage percentages in the tumor core and margin (Fig. S3a). Moreover, no significant difference in the immunofluorescent of junctional protein was observed before and after optoBBTB (Fig. S4b). These results suggest that optoBBTB in 73C GEMM did not influence the density or the junctional protein immunofluorescent of the angiogenic blood vessels. In our recent work (31), we demonstrated that laser excitation of vascular-targeting AuNPs was associated with a transient elevation and propagation of Ca^{2+} , actin polymerization, and phosphorylation of ERK1/2 (extracellular signal-regulated protein kinase). They collectively activated the cytoskeleton resulting in increased paracellular permeability. We hypothesize that the increased barrier permeability after optoBBTB is due to the Ca^{2+} -mediated activation of the mechanobiological pathways and the re-arrangement of the cytoskeleton. Moreover, angiogenic blood vessels may respond differently to optoBBTB than normal brain microvasculature. Further investigation may be focused on examining how the blood vessel phenotypes respond to optoBBTB and change the barrier permeability.

C. Page 33, discussion, updated the text.

- (a) Original: Among the various methods to change the BBTB permeability, optoBBTB presents unique opportunities for further development and clinical translation. First, near-infrared light-absorbing nanoparticles can be exploited since the light in this region exhibits deeper tissue penetration to cover the tumor margin in a larger animal model. Second, while light can be delivered transcranially in the mouse brain, fiber delivery to the human brain is envisioned, especially after surgical removal of the primary tumor. Placing an optical fiber in the tumor surgical cavity would allow the delivery of side-emitting light to the tumor margin (Fig. S10). Moreover, extracranial light delivery is within the realm of implementation with a transparent cranial window to replace a portion of the skull (47). All these developments will facilitate the next-stage translation of optoBBTB for GBM treatment going forward.
- (b) Revised: Among the various methods to change the BBTB permeability, optoBBTB presents unique opportunities for further preclinical and clinical investigations. In preclinical settings, our method can be useful as a drug development and screening platform (optoBBTB and GEMMs) for testing a class of potent anticancer drugs. In terms of clinical investigations, there are several opportunities for further development. First, the 532 nm light exploited in the current study enables light delivery to the mouse cortex and therefore, the treatment of cortically-located tumors. However, to treat brain tumor in the deep brain region, near-infrared light-absorbing nanoparticles can be exploited since the light in this region exhibits deeper tissue penetration to cover the tumor margin in a larger animal model. Second, while light can be delivered

transcranially in the mouse brain, fiber delivery to the human brain is envisioned, especially after the surgical removal of the primary tumor. Placing an optical fiber in the tumor surgical cavity would allow the delivery of side-emitting light to the tumor margin far from the tumor mass (Fig. S13). Moreover, since human skulls are significantly thicker than mice skulls, extracranial light delivery is within the realm of implementation with a transparent cranial window to replace a portion of the skull (63). Furthermore, our recent work investigated opening the blood-spinal cord which represent another important application of the technology and has advantages compared with the state-of-the-art methods (64). Further work is ongoing to investigate tumor treatment in this area.

3. Figures 3 and 6 lack a control group. While the authors state that the contralateral side can act as an internal control, the contralateral side does not have a tumor and is not appropriate to use as a control.

Response:

We thank the reviewer for this comment. In our revised manuscript, we provided a control group by opening the BBTB using a control nanoparticle AuNP-PEG, which does not target the tight junction. Briefly, we synthesized 50 nm AuNP and functionalized the particles with mPEG (1 kDa). The particles were intravenously injected into the PS5A1 (18.5 $\mu\text{g/g}$) or 73C (37 $\mu\text{g/g}$) tumor bearing mice. 1 hour later, the mice received laser excitation in the tumor area (40 mJ/cm^2 , 1 pulse) followed by fluorescent Taxol646 (12.5 mg/kg). 30 min later, the mice were perfused with PBS and 4% PFA. The brains were extracted and cryosectioned to 20 μm thick coronal slices for fluorescent imaging. No fluorescent Taxol leakage was observed in the laser area (updated Fig. 3c and 4d). In another set of experiment, we analyzed the Taxol concentration in the tumor using HPLC-MS. The results show that there was no significant increase in the Taxol delivery after optoBBTB with AuNP-PEG compared with Taxol delivery to the GEMMs without optoBBTB (Fig. S6d, S7d). In summary, these results suggest that non-tight junction targeting AuNP-PEG in combination with laser excitation could not modulate the BBTB for Taxol delivery to the tumor.

Revision:

A. Page 39, Method, updated the text.

(a) Original: The preparation of brain vascular-targeting gold nanoparticles

The method to prepare brain vascular-targeting gold nanoparticles (AuNPs-BV11) was adapted from our previously reported method (13). AuNP-TfR and AuNP-VEGFR2 were prepared using a similar approach by replacing BV11 with the anti-TfR antibody and anti-VEGFR2 antibody. The concentration, size distribution, and morphology of the nanoparticles were analyzed using Ultraviolet-Visible Spectroscopy (UV-Vis), Dynamic Light Scattering (DLS), and Transmission Electron Microscopy (TEM).

(b) Revised: The preparation of brain vascular-targeting gold nanoparticles and control gold nanoparticles

The method to prepare brain vascular-targeting gold nanoparticles (AuNPs-BV11) was adapted from our previously reported method (17). AuNP-TfR and AuNP-VEGFR2 were prepared using a similar approach by replacing BV11 with the anti-TfR antibody and anti-VEGFR2 antibody. The control nanoparticles were prepared by mixing 50 nm AuNP with mPEG-thiol (PEG: polyethylene glycol, 1 kDa) with a thiol: AuNP=300:1 molar ratio, for 3 hour on ice. The particles were washed three times in pure water. The concentration, size distribution, and

morphology of the nanoparticles were analyzed using Ultraviolet-Visible Spectroscopy (UV-Vis), Dynamic Light Scattering (DLS), and Transmission Electron Microscopy (TEM).

B. Fig. 3c and 4d, updated.

Fig. R2. OptoBBTB using AuNP-PEG does not improve the delivery of fluorescent Taxol (Taxol646) in PS5A1 (Fig. 3c) and 73C GEMM (Fig. 4d). The tumor is indicated by GFP fluorescent in Fig. 3c and Hoechst staining in Fig.4d (white arrow). The scale bar represents 1 mm.

C. Fig. S6d and S7d, updated.

Fig. R3. The analysis of Taxol concentration in the plasma and tumor in PS5A1 (S6d) and 73C (S7d). The results showed that after optoBBTB with AuNP-PEG followed by Taxol administration, there was no significant difference of the Taxol concentration in the tumor compared with the mice that had no optoBBTB treatment (Taxol only). N=3 mice. Data are expressed as Mean \pm SD. The quantification analysis was performed by unpaired Student's two-sided *t* test (S6d) or One-way ANOVA followed by Tukey's multiple comparisons test (S7d), respectively.

4. Statistical analysis is sorely lacking, and powering of mouse studies is not described. For example, why were only 2 mice used for figure 7c, 5 mice for 7d, and 7 mice for 7e? There are no statistics performed on the mouse survival curves in Figures 4 and 7 but the text reports an increase in medial survival with

percent improvement. Statistical analysis is needed to support this claim. There is no quantification of IVIS imaging in Figure 4C or 7C but the authors report ‘marked decrease in tumor size’. Does this data correlate to the quantified tumor volume by histology in 4D and 7D?

Response:

We thank the reviewer for this suggestion. In our revised manuscript, all the in vivo results were repeated in at least 3 mice. We used G*power analysis to calculate the sample sizes for tumor size and survival analysis. The effect size (0.9) was obtained from our preliminary study. With 85% power, and alpha set to 0.05, the sample size required was calculated as $n = 5$ per group. The sample size of animal experiments matches well with similar studies in the field (*Nat Commun* **11**, 5687 (2020), *Nat Cancer* **2**, 932–949 (2021)), and we determined to use at least 5 mice per group. We updated the method to include this information.

In addition, we provided the mouse survival curves with statistical analysis to support the findings (updated Fig. 5f, 6f). We also included the quantification of the IVIS images using Living Image® Software for IVIS® Lumina III In Vivo Imaging System (updated Fig. 5h, 6h).

Revision:

A. Page 45, Method, updated the text.

(a) Original: **Statistical analysis**

Statistical analyses were performed using GraphPad Prism 9 software. The indication of each data dot, n values per group, and details of statistical testing are provided in the figure caption.

(b) Revised: **Statistical analysis**

Statistical analyses were performed using GraphPad Prism 9 software. The logrank test was performed for statistical analysis of survival time, and one-way analysis of variance (ANOVA) followed by Tukey’s multiple comparisons test and unpaired Student’s two-sided t test for two comparisons were performed for other statistical analyses.

For in vitro experiment, six replicates were performed. For in vivo experiment, at least three independent experiments were performed. We used G*power analysis to calculate the sample sizes for tumor size and survival analysis. The effect size was obtained from our preliminary study. With 85% power, and alpha set to 0.05, the sample size required was calculated as $n = 5$ per group.

The n values per group and details of statistical testing are provided in the figure caption.

B. Fig. 5f and 6f, updated.

Fig. R1. Kaplan-Meier survival analysis in updated Fig. 5 and 6. N=7 mice in each group. The statistical analysis was performed with logrank test.

C. Fig. 5h and 6h, updated.

Fig. R4. The quantification of GFP fluorescent in (5h) PS5A1 GEMM and (6h) 73C GEMM after treatment using Living Image® Software for IVIS® Lumina III In Vivo Imaging System. N=5 mice in each group. Data are expressed as Mean ± SD. Quantification analysis was performed with One-way ANOVA followed by Tukey's multiple comparisons test.

D. Page 21, updated the text.

- Original: Moreover, optoBBTB+Taxol delivery significantly increased the median overall survival by 50%, from 40 days to 60 days (**Fig. 5F**), due to a marked inhibition of tumor growth (**Fig. 5G, H, I** top). Ki67 staining and cell apoptosis analysis showed that optoBBTB+Taxol decreased cell proliferation and increased cellular apoptosis compared to the other groups (**Fig. 5I** middle-bottom, **J, K**).
- Revised: Moreover, optoBBTB+Taxol delivery significantly increased the median overall survival by 50%, from 40 days to 60 days (**Fig. 5f**), due to a marked inhibition of tumor growth (**Fig. 5g, h, i** top). Ki67 staining and cell apoptosis analysis was performed by calculating the signal positive (ki67+ or TUNEL+) cell numbers over total cell numbers. The results show that optoBBTB+Taxol decreased cell proliferation and increased cellular apoptosis compared with the other groups (**Fig. 5i** middle-bottom, **j, k**).

5. Additional information about animal studies, including euthanasia criteria is important. The authors should clarify why female immunodeficient mice were utilized, since the GEM glioma cell lines were generated in a BL6 background.

Response:

We thank the reviewer for this comment. In the revised manuscript, we provided the euthanasia criteria. According to the approved animal protocol, animals were deemed for euthanasia when they developed weight loss (>20%), loss of grooming, seizures and focal motor deficits.

Although the cell lines were generated in a BL6 background, we used immunodeficient mice in order to make the results comparable to the existing literature, since most of the glioblastoma treatment studies used immunodeficient mice. Moreover, we used female mice to ease the accommodation of female mice in the same cage since brain tumor bearing male mice invariably incur severe injuries and often death. In our revised manuscript, we included the sex in the abstract according to the guidance on Sex and Gender Reporting.

Revision:

A. Page 35, method, updated the text.

- (a) Original: The immunodeficient nude mice Foxn1nu (Nu/J, stock number 002019, 7 weeks old, female, 20-25 g) were ordered from Jackson Laboratories. Animal protocols were approved by the Institutional Animal Care Use Committee (IACUC) of the University of Texas at Dallas.
- (b) Revised: The immunodeficient nude mice Foxn1nu (Nu/J, stock number 002019, 7 weeks old, female, 20-25 g) were ordered from Jackson Laboratories. All animals were bred in pathogen-free conditions, in temperature and humidity-controlled housing, under a 12-h light/dark cycle, and with free access to food and water. Animal protocols were approved by the Institutional Animal Care Use Committee (IACUC) of the University of Texas at Dallas.

B. Page 42, updated the text.

- (a) Original: Similar treatment groups were used for PS5A1 and 73C glioma-bearing mice to obtain the survival rate, with 7 mice in each group.
- (b) Revised: Similar treatment groups were used for PS5A1 and 73C glioma-bearing mice to obtain the survival rate, with 7 mice in each group. The tumor bearing mice were euthanized if they developed weight loss (>20%), loss of grooming, seizures and focal motor deficits according to the approved animal protocol.

C. Page 6, updated the text.

- (a) Original: We next evaluated the growth and BBTB permeability for the PS5A1 model (*Braf*^{V600+/-}, *Ink4ab/Arf*^{-/-}, *Pten*^{-/-}) with diffuse single-cell infiltration through the brain parenchyma. These cells were engineered to express GFP. The BBTB integrity of the mice during PS5A1 GBM progression was analyzed using i.v. injection of EZ-link biotin (660 Da) and Evans blue (66 kDa, albumin-bound).
- (b) Revised: To recapitulate these features in preclinical models, we characterized two GEMMs in terms of their BBTB integrity, tumor progression patterns, and TJ properties. These GEMMs were generated using neural-stem-cell-derived PS5A1 (*Braf*^{V600+/-}, *Ink4ab/Arf*^{-/-}, *Pten*^{-/-}) and astrocyte-derived glioma cell line 73C (*Braf*^{V600E}, *Pten*^{-/-}, *p53*^{-/-}) (29, 30). These cell lines were engineered to express green fluorescent protein (GFP). We first established PS5A1 and 73C GEMMs in female nude mice (Nu/J, 002019, age 7 weeks, the Jackson Laboratory) and evaluated their BBTB permeability. Specifically, 368 nL of PS5A1 glioma cell suspension or 92 nL of 73C

glioma cells ($2 \times 10^5/\mu\text{L}$) was constantly injected into the mouse cortex (-1 mm, -1 mm, 0.5 mm) using a nanoinjector equipped with a glass micropipette (50 μm tip, see method for details). Although the cell line was generated in a BL6 background, we used immunodeficient mice to make the results comparable to the existing literature since most glioblastoma treatment studies used immunodeficient mice. The BBTB integrity of the mice during GBM progression was analyzed using i.v. injection of EZ-link biotin (660 Da) and Evans blue (66 kDa, albumin-bound).

D. Page 2, updated the text.

- (a) Original: Here we analyzed the intratumoral BBTB heterogeneity in human GBM and established two genetically engineered mouse models (GEMMs) that recapitulate two important glioma phenotypes, including the diffusely infiltrative tumor margin and angiogenic core.
- (b) Revised: Here we analyzed the intratumoral BBTB heterogeneity in human GBM and characterized two genetically engineered mouse models (GEMMs, in female mice) that recapitulate two important glioma phenotypes, including the diffusely infiltrative tumor margin and angiogenic core.

6. Data analysis, especially with microscopy, is not robustly described. E.g. how were microscopy slides used to calculate area x thickness? What was the thickness of each slice? Was a 3-dimensional mask used? How was the tumor thresholded and was this kept constant for all samples analyzed? Were microscopy settings kept constant between conditions?

Response:

We thank the reviewer for this question. To obtain the tumor volume after treatment in PS5A1 glioma-bearing mice, coronal brain slices (30 μm) were imaged with Olympus VS120 virtual slide microscope with a 10x objective. The image acquisition settings were kept constant between the samples. A threshold was set to cover the tumor areas using Fiji/Image-J (exemplified in **Fig. S10b**), consistent across all brain slices analyzed. Therefore, the total tumor area can be measured using Fiji/Image-J. And the tumor volume can thus be determined by the product of total area and slice thickness.

This information has been added to the revised manuscript.

Revision:

A. Page 44, updated the text.

- (a) Original: All the fluorescent images were taken with the Olympus SD-OSR spinning disk super-resolution microscope and Olympus VS120 virtual slide microscope. The transmission electron microscopy (TEM) images were taken using a JEOL JEM-2010 microscope. For the IHC staining, the images were analyzed by Fiji/ImageJ. To study the changes in junctional proteins, the area fraction of Claudin-5, ZO-1, VE-cadherin, and JAMA-A was obtained and normalized by CD3 (indicating cerebral vessel). Vasculature density was analyzed by area fraction of CD31. To obtain the tumor volume after treatment in PS5A1 glioma-bearing mice, brain slices were imaged with Olympus VS120 virtual slide microscope. The total area with tumor GFP fluorescent was determined by selecting the optimal threshold and analyzed with Fiji/Image-J. The tumor volume was calculated by area \times thickness.
- (b) Revised: All the fluorescent images were taken with the **IVIS® Lumina III In Vivo Imaging System**, Olympus SD-OSR spinning disk super-resolution microscope and Olympus VS120

virtual slide microscope. The transmission electron microscopy (TEM) images were taken using a JEOL JEM-2010 microscope.

To measure the tumor size after treatment, the brains were extracted and imaged the GFP fluorescent using IVIS® Lumina III In Vivo Imaging System. The radiant efficiency was measured with Living Image® Software. To image the dye or Taxol extravasation after optoBBTB, the samples were imaged with a 10x objective (Olympus VS120 virtual slide microscope) or 100x oil immersion objective (Olympus SD-OSR spinning disk super-resolution microscope). To study the changes in junctional proteins using IHC staining, the samples were imaged with a 100x oil immersion objective (Olympus SD-OSR spinning disk super-resolution microscope). Then the images were analyzed by Fiji/ImageJ. The area fraction of Claudin-5, ZO-1, VE-cadherin, Occludin, and JAMA-A was obtained and normalized by CD31 (indicating cerebral vessel). To study the vasculature density, the images were acquired using a 10x objective (Olympus SD-OSR spinning disk super-resolution microscope). The vasculature density was analyzed by area fraction of CD31 or lectin. The ki67 and TUNEL staining were imaged with a 100x oil immersion objective (Olympus SD-OSR spinning disk super-resolution microscope). The image acquisition settings were kept constant during the experiment.

To obtain the tumor volume after treatment in PS5A1 glioma-bearing mice, coronal brain slices (30 μm) were imaged with Olympus VS120 virtual slide microscope with a 10x objective. The image acquisition settings were kept constant between the samples. A threshold was set to cover the tumor areas using Fiji/Image-J (exemplified in **Fig. S10b**), consistent across all brain slices analyzed. Therefore, the total tumor area can be measured using Fiji/Image-J. And the tumor volume can thus be determined by the product of total area and slice thickness.

B. Fig. S10b, updated.

Fig. S10. Measurement of PS5A1 tumor size. **a** Magnetic Resonance Imaging of the tumor at 42 dpi shows no T1-weighted contrast enhancement or T2-weighted hyperintensity, while fluorescent imaging of GFP confirmed the presence of the tumor. The scale bar represents 1 mm. **b** Tumor volume analysis using fluorescent images. Left: an example of the original fluorescent image. The tumors are indicated by GFP fluorescent and arrows. Right: the image after processing with the threshold function in Fiji/Image-J.

7. Additional clarity around the choice of mouse models, how they were generated, and additional characterization is needed.

Response:

We thank the reviewer for this suggestion. As discussed above, in the revised manuscript (Discussion, page 30), we introduced the rationale of the selection of these tumor cell lines. We also included in the main text how the GEMMs are generated (Result, page 6).

8. The authors should describe their method of implanting genetically engineered mouse cells into the brain and the location of injection. The tumors are orthotopic, but also very superficial, to the point that the 73C tumors are primarily exophytic at the time points shown for efficacy studies. Were superficial

tumors generated purposefully in order to use transcranial pulsed laser? Would this method be effective for a tumor in another location?

Response:

We thank the reviewer for this question. We had the tumor implantation method in the method section Glioma cell transplantation. To make it clear, in our revised manuscript, we added this information in the main text.

The tumor cells were injected at about 0.5 mm below in the cortex. We selected this depth because of the light penetration in the mouse brain is largely confined to the cortex. However, in our previous publication, we showed that light delivery to the deeper brain region could be achieved by using an optical fiber (*Nano Lett.* 2021, 21, 22, 9805–9815). Moreover, to deliver light to the deeper brain region, near-infrared light-absorbing nanoparticles can be exploited since the light in this region exhibits deeper tissue penetration. Therefore, we believe that this method is suitable to treat superficial brain tumors, but can be optimized for a tumor in a deeper brain location.

Revision:

A. Page 6, update the text.

Revised: To recapitulate these features in preclinical models, we characterized two GEMMs in terms of their BBTB integrity, tumor progression patterns, and TJ properties. These GEMMs were generated using neural-stem-cell-derived PS5A1 (*Braf*^{N600+/-}, *Ink4ab/Arf*^{-/-}, *Pten*^{-/-}) and astrocyte-derived glioma cell line 73C (*Braf*^{N600E}, *Pten*^{-/-}, *p53*^{-/-}) (29, 30). These cell lines were engineered to express green fluorescent protein (GFP). We first established PS5A1 and 73C GEMMs in female nude mice (Nu/J, 002019, age 7 weeks, the Jackson Laboratory) and evaluated their BBTB permeability. Specifically, 368 nL of PS5A1 glioma cell suspension or 92 nL of 73C glioma cells ($2 \times 10^5/\mu\text{L}$) was constantly injected into the mouse cortex (-1 mm, -1 mm, 0.5 mm) using a nanoinjector equipped with a glass micropipette (50 μm tip, see method for details). Although the cell line was generated in a BL6 background, we used immunodeficient mice to make the results comparable to the existing literature since most glioblastoma treatment studies used immunodeficient mice. The BBTB integrity of the mice during GBM progression was analyzed using i.v. injection of EZ-link biotin (660 Da) and Evans blue (66 kDa, albumin-bound).

B. Page 33, Discussion, updated the text.

- (a) Original: Among the various methods to change the BBTB permeability, optoBBTB presents unique opportunities for further development and clinical translation. First, near-infrared light-absorbing nanoparticles can be exploited since the light in this region exhibits deeper tissue penetration to cover the tumor margin in a larger animal model. Second, while light can be delivered transcranially in the mouse brain, fiber delivery to the human brain is envisioned, especially after surgical removal of the primary tumor. Placing an optical fiber in the tumor surgical cavity would allow the delivery of side-emitting light to the tumor margin (Fig. S10).
- (b) Revised: Among the various methods to change the BBTB permeability, optoBBTB presents unique opportunities for further preclinical and clinical investigations. In preclinical settings, our method can be useful as a drug development and screening platform (optoBBTB and GEMMs) for testing a class of potent anticancer drugs. In terms of clinical investigations, there are several opportunities for further development. First, the 532 nm light exploited in the current study enables light delivery to the mouse cortex and therefore, the treatment of cortically-located

tumors. However, to treat brain tumor in the deep brain region, near-infrared light-absorbing nanoparticles can be exploited since the light in this region exhibits deeper tissue penetration to cover the tumor margin in a larger animal model. Second, while light can be delivered transcranially in the mouse brain, fiber delivery to the human brain is envisioned, especially after the surgical removal of the primary tumor. Placing an optical fiber in the tumor surgical cavity would allow the delivery of side-emitting light to the tumor margin far from the tumor mass (Fig. S13). Moreover, since human skulls are significantly thicker than mice skulls, extracranial light delivery is within the realm of implementation with a transparent cranial window to replace a portion of the skull (63). Furthermore, our recent work investigated opening the blood-spinal cord which represent another important application of the technology and has advantages compared with the state-of-the-art methods (64). Further work is ongoing to investigate tumor treatment in this area.

9. In the discussion, the authors state that they showed that optoBBTB ‘reversibly increases BBTB permeability’ but the experiments supporting the reversible nature in these mouse models (Fig S2 and S6) are difficult to interpret. Clearly the same mouse cannot be shown in each image, and the images in S2B and S6B appear to have a much higher baseline signal of biotin. The authors should describe these experiments, including number of mice and analysis methodology in greater detail in order to support the claim of reversibility.

Response:

We thank the reviewer for this comment. In this experiment, we used 3 mice to test the BBTB recovery at the 3 different time points. Briefly, we i.v. injected AuNP-BV11 (18.5 $\mu\text{g/g}$ or 37 $\mu\text{g/g}$) into PS5A1 or 73C tumor-bearing mice (mouse 1-3). 1 hour later, we applied ps laser stimulation followed by EZ-link biotin (2 mg/ml) injection into mouse 1. Mouse 1 was then perfused with PBS and 4% PFA after 30 min. The brain was extracted and cryosectioned into 20 μm thick coronal sections and stained with cy3-streptavidin to detect the presence of biotin. Next, we injected EZ-link biotin into mouse 2 and 3 on day 1 and day 3, respectively. The brains were processed using the same protocol. The biotin leakage was imaged using Olympus VS120 virtual slide microscope with a 10x objective. The image acquisition settings were kept constant between the samples in the same experiment. Although only representative images were shown in the figure, the experiments were repeated independently for 3 times to confirm the results.

Revision:

A. Page 41, updated the text.

- (a) Original: AuNP-BV11 (18.5 $\mu\text{g/g}$ or 37 $\mu\text{g/g}$) was i.v. injected to PS5A1 and 73C and tumor-bearing mice at 14 and 7 dpi, respectively (N=3 mice for each group). The mice received a picosecond laser pulse (40 mJ/cm^2) after 1 hour. Then EZ-link biotin was injected immediately after laser excitation or after 1 or 3 days. 30 min after the dye injection, the brains were extracted and frozen on dry ice and cut into 20 μm thick slices using a cryostat. The brain slices were incubated with Cy3-streptavidin to detect biotin and Hoechst solution to stain cell nuclei.
- (b) Revised: AuNP-BV11 (18.5 $\mu\text{g/g}$ or 37 $\mu\text{g/g}$) was i.v. injected to PS5A1 and 73C and tumor-bearing mice at 14 and 7 dpi, respectively (N=3 mice for each group, namely mouse 1-3). The mice received a picosecond laser pulse (40 mJ/cm^2) after 1 hour. Then EZ-link biotin was injected immediately after laser excitation (mouse 1) or after 1 (mouse 2) or 3 days (mouse 3). 30 min after the dye injection, the brains were extracted and frozen on dry ice and cut into 20 μm thick slices using a cryostat. The brain slices were incubated with Cy3-streptavidin to detect

biotin and Hoechst solution to stain cell nuclei. The biotin leakage was imaged using Olympus VS120 virtual slide microscope with a 10x objective.

B. Fig. S6b and 7c, updated.

Fig. R4. Recovery of BBB permeability after optoBBTB at 30 min, 1 day, and 3 days in PS5A1 GEMM (S6b) and 73C GEMM (S7c). In S6b, the tumor is indicated by GFP, and the dye leakage is indicated by red fluorescent and the arrows. In S7c, the tumor is indicated by Hoechst staining (HOE) of the cell nuclei, and the dye leakage is indicated by red fluorescent and the arrows. The scale bar represents 1 mm.

Minor points

10. The authors state that taxol is ‘among the most widely used oncology drugs’ but do not provide a reference.

Response:

We thank the reviewer for this comment. In the revised manuscript, the references have been added.

Revision:

Page 5, add references

- (a) Original: Taxol is among the most widely used oncology drug because of its proven efficacy in multiple cancer subtypes but was abandoned for GBM treatment following its failure in early phase clinical trials due to poor brain penetration.
- (b) Revised: Taxol is among the most widely used oncology drug because of its proven efficacy in multiple cancer subtypes, but it was abandoned for GBM treatment following its failure in early-phase clinical trials due to poor brain penetration (21-24).

Reference:

21. Weaver, B. A. How Taxol/paclitaxel kills cancer cells. *Mol. Biol. Cell.* **25**, 2677-2681 (2014).
22. Abu Samaan, T. M., Samec, M., Liskova, A., Kubatka, P., & Büsselberg, D. Paclitaxel's mechanistic and clinical effects on breast cancer. *Biomolecules.* **9**, 789 (2019).
23. Kampan, N. C., Madondo, M. T., McNally, O. M., Quinn, M., & Plebanski, M. Paclitaxel and its evolving role in the management of ovarian cancer. *Biomed. Res. Int.* **413076** (2015).
24. Blair, H. A., & Deeks, E. D. Albumin-bound paclitaxel: a review in non-small cell lung cancer. *Drugs.* **75**, 2017-2024 (2015).

11. The statement in the introduction that this investigation provides ‘the first definitive evidence of BBTB modulation and therapeutic effect...’ is quite broad, as there are many other interpretations of ‘BBTB modulation’ including technologies like focused ultrasound and osmotic BBB disruption.

Response:

We thank the reviewer for this comment. In our original manuscript (introduction, page 3-4), we have described several approaches to modulate the BBTB permeability, such as using hyperosmotic agents (mannitol), opening the TJ with a TJ modulator, enhancing drug penetration through inhibiting drug efflux transporters or via receptor-mediated transport, and using focused ultrasound and microbubbles. In our revised manuscript, we updated the text to specific our goal and result in this work.

Revision:

Page 5, updated the text.

- (a) Original: Our investigations provide the first definitive evidence of BBTB modulation and therapeutic benefit using clinically relevant models and establish the foundation for future clinical translation for GBM patients.
- (b) Revised: Our investigations provide **definitive evidence** of BBTB modulation and therapeutic benefit using **optoBBTB in** clinically relevant models.

12. The authors should carefully review scale bars; Figure 1 C-D are both listed as having the same scale, however the nuclei sizes are visually quite different.

Response:

We thank the reviewer for the comment. However, we showed in the legend that the scale bars in Fig. 1 C-D (now Fig. S1) represent 20 μm and 50 μm , respectively.

13. The authors note data about 73C GEMM survival with TMZ treatment, but data is not shown or cited.

Response:

We have removed this statement from the revised version.

14. The characterization of nanoparticles used with OptoBBTB (figure S1) does not make clear whether the radius or diameter is shown, and does not indicate whether z-average or number average is being used to estimate size with dynamic light scattering. In D, it is not clear whether data from AuNP or AuNP-

BV11 is shown. There appears to be only one replicate of data in S1 A-D, but presumably multiple batches of NPs were generated for this study.

Response:

We thank the reviewer for this question. In the revised Fig. S5d, we showed that we measured the hydrodynamic diameter. Since it was a monodispersed sample, we used Z-average (intensity weighted mean hydrodynamic diameter) to interpret the size of the nanoparticles. We also clarified that in Fig. S5b, the image shows the AuNP before antibody conjugation (therefore the AuNP core). In addition, In Fig. S5 b-d, we aim to show the representative characterization of the nanoparticles. We indeed synthesized multiple batches of the NPs but only the ones with comparable sizes and TEM morphologies were selected for the experiment.

Revision:

A. Fig. S5, updated.

Fig. S5. Characterization of AuNP-BV11 and its biodistribution in tumor-bearing mice. **a** The surface functionalization of AuNP-BV11. **b** The morphology and size of the AuNP core are characterized by Transmission Electron Microscopy. The size of the nanoparticles (50 ± 4 nm) was measured with Image-J by manually counting 100 particles. **c** The localized surface plasmon resonance peak of the nanoparticles is characterized by UV-Vis-NIR spectroscopy. **d** The nanoparticle hydrodynamic diameter distribution by relative intensity is characterized by Dynamic Light Scattering. The Z-average for AuNP and AuNP-BV11 was 49 nm and 69 nm, respectively. Source data are available as a Source Data file.

B. Page 12, updated the text.

- (a) Original: First, TJ component JAM-A targeted nanoparticles (AuNP-BV11, 50 nm) were prepared, and their physicochemical properties were characterized (Fig. S1A-D).
- (b) Revised: First, TJ component JAM-A targeted nanoparticles (AuNP-BV11, 50 nm, Fig. S5a) were prepared, and their physicochemical properties were characterized. A representative set of characterizations is shown in Fig. S5b-d. Only the batches with comparable physicochemical properties in terms of hydrodynamic diameter and morphology were selected for the experiments.

15. The method of GFP transduction in the PS5A1 model is not clear, from the methods an AAV expressing GFP was used to generate the murine cell line, which was later infected with 'lenti-GFP'. Methods here should be expanded. The genotype of each tumor is not confirmed.

Response:

We thank the reviewer for this comment. We used AAV5-GFAP-Cre-GFP to induce Cre-dependent mutations (BrafV600E^{f/+}; Ink4ab/Arf^{f/f}; Pten^{f/f}) and generate de novo glioma, where the GFP expression was transient as an indicator of AAV transduction within a shorter time period and GFP signal got weaker after multiple cycles of cell proliferation. For long term cell tracing, GFP Lenti virus was used to label cells permanently with GFP expression. The genotype of each tumor cell line was confirmed by PCR and DNA sequencing, which shows mutations in cell lines were consistent with the host mouse genotype. Since these GEMMs have been published, we did not include the genotyping in the manuscript.

Revision:

Page 36, updated.

- (a) Original: These cells were derived from conditional multi-allele primary astrocytes and infected ex vivo with an adeno-Cre virus to generate a primary transformed cell line. These cells were then infected with a Lenti-GFP and selected by puromycin with stable green fluorescent protein expression.
- (b) Revised: These cells were derived from conditional multi-allele primary astrocytes and infected ex vivo with an adeno-Cre virus to generate a primary transformed cell line. Since the AAV transduction provides a transient GFP expression, these cells were further infected with Lenti-GFP and selected by puromycin for stable green fluorescent protein expression.

16. There are several minor grammatical errors throughout that can be corrected to improve readability

Response:

We thank the reviewer for the comment. The typos and grammatical errors in the manuscript have been corrected.

17. The references should be very thoroughly checked by the authors – several references appear to be misnumbered, making it difficult to review this paper. Notably, reference 13 – which in the text refers to the authors prior work describing optoBBTB—is not correct and this reviewer presumes they are instead referring to reference 17 (Li et al Nano Letters 2021).

Response:

We thank the reviewer for the suggestion. The reference list has been updated in the revised manuscript.

Reviewer #4 (Remarks to the Author): Expert in light-inducible nanoparticles

The paper entitled “Optical Blood-Brain-Tumor Barrier Modulation Enhances Drug Penetration and Therapeutic Outcome in Clinically Relevant infiltrative and Angiogenic Glioblastoma Models” is an interesting paper which describes the modulation of the blood brain barrier using plasmonic carriers that generate heat after laser stimulation to facilitate the extravasation of anticancer drugs. The paper is well-written however the narrative is complex since the authors have used two genetically engineered mouse models that recapitulate different phenotypes (i.e. diffusively infiltrative tumor margin and angiogenic core) in glioblastoma. This means that the authors needed to introduce and characterize both animal models before evaluating the potential of their opto-activation approach. The authors show that the anticancer drug extravasates to brain parenchyma leading to a reduction in tumor volume and a prolongation in the survival of the animals. The modulation of the BBB permeability using plasmonic nanocarriers is not novel. The authors (Ref. 17) as well as other groups (doi: 10.1039/c3nr06770j; 10.1016/j.jconrel.2018.06.013; <https://doi.org/10.1073/pnas.1018790108>) have shown that plasmonic nanocarriers, in some cases only the laser, can indeed induce a transient opening of the BBB. The pre-clinical evaluation of the anticancer drug using two glioblastoma animal models showing different glioma phenotypes, being the phenotypes supported by human pathophysiology data, is relevant; however, the authors are brief to support experimentally the mechanism of their approach which decreases significantly the enthusiasm of this reviewer.

Response:

We thank the reviewer for the careful evaluation of our work and the constructive advice.

In our updated manuscript, we reorganized the structure to make a better comparison of the two models. We first compared the BBTB permeability during disease progression in PS5A1 and 73C GEMM (Fig. 1). We next compared the histopathological characterizations of the two GEMMs (Fig. 2). Then we showed the efficacy of the optoBBTB in PS5A1 and 73C GEMMs, respectively (Fig. 3-4), and evaluated the treatment outcomes in the two GEMMs (Fig. 5-6). Moreover, we added a discussion on the two models in terms of their characteristics and responses to the treatment, to better illustrate the role of the nanoparticle targeting or whether tumor vascular phenotype the therapy. We believe that the current structure has a better comparison between the two models before evaluating the optoBBTB efficacy.

We agree with the reviewer that the modulation of the BBB permeability using plasmonic nanoparticles has been reported. However, our method has three critical differences compared with these strategies. . First, our AuNP-BV11 nanoparticles binds to the vasculature and optoBBB modulation approach involves the increase of paracellular permeability without causing cell damage. Second, optoBBTB exploits nanoscale mechanical perturbation rather than the heating effect. Since the duty cycle of the laser is low, the amount of heat dissipated from the nanoparticle into the surrounding medium is insufficient to raise tissue temperature (*J. Phys. Chem. C, Nanomater. Interfaces.* 125, 26718–26730 (2021)). Last, since our control groups (optoBBTB+Vehicle) do not show a reduction of tumor volume or an increased survival rate, and the local temperature was not enhanced after laser excitation, we believe that the reduced tumor volume is mainly mediated by the extravasation of Taxol rather than by other mechanisms such as the activation of the immune system by the heating effect.

Moreover, in our recent publication (*Nanoscale*, 2023,15, 3387-3397), we demonstrated the mechanism of the optical BBB modulation using an in vitro BBB model, which was established using human cerebral microvascular endothelial cells. We showed that the picosecond laser excitation of vascular-targeting AuNPs produced nanoscale mechanical perturbation. This perturbation triggers several mechanobiological responses, including (1) actin polymerization; (2) Ca²⁺-influx including from mechanosensitive ion channels (such as TRPV4 and Piezo1); (3) The activation of Inositol Trisphosphate

(IP3) pathway, in which IP3 activates the receptors on the endoplasmic reticulum and leads to the Ca²⁺ release. The elevation of Ca²⁺ from the extracellular influx and intracellular IP3 pathway activates ERK1/2 phosphorylation. The phosphorylation of ERK1/2, together with the actin network, leads to an activation of the cytoskeleton resulting in an increase in paracellular permeability. While these mechanisms might contribute to the increased BBTB permeability after optoBBTB in the current study, we hypothesize that the blood vessels phenotypes may respond differently to the optoBBTB, resulting in different opening efficiency in the two GEMMs. Further investigation may examine how the normal and angiogenesis blood vessels respond to laser stimulation and change the permeability.

We included the above information in the revised discussion.

Revision:

A. Page 32, Discussion, added text and references.

Revised: Although modulation of the BBB permeability using the laser with/without plasmonic nanoparticles has been reported elsewhere (59-61), our method has three critical differences compared with these strategies. First, our AuNP-BV11 nanoparticles binds to the vasculature and optoBBB modulation approach involves the increase of paracellular permeability without causing cell damage. Second, optoBBTB exploits nanoscale mechanical perturbation rather than the heating effect (62). Since the duty cycle of the laser is low, the amount of heat dissipated from the nanoparticle into the surrounding medium is insufficient to raise tissue temperature. Last, since our control group (optoBBTB+Vehicle) did not show a reduction of tumor volume or an increased survival rate, and the local temperature was not enhanced after laser excitation, we believe that the reduced tumor volume is mainly mediated by the extravasation of Taxol rather than by other mechanisms such as the activation of the immune system by the heating effect. Recently, we unveiled the mechanisms of the optical BBB modulation using an in vitro BBB model established with human cerebral microvascular endothelial cells. We showed that the picosecond laser excitation of vascular-targeting AuNPs produced nanoscale mechanical perturbation, which triggers several mechanobiological responses, including (1) actin polymerization that leads to the cytoskeletal contraction, (2) Ca²⁺-influx including from mechanosensitive ion channels (such as TRPV4 and Piezo1), and (3) the activation of Inositol Trisphosphate (IP3) pathway and the Ca²⁺ release from the endoplasmic reticulum. The elevation of Ca²⁺ from the extracellular influx and intracellular IP3 pathway activates ERK1/2 phosphorylation. These effects led to a mechanobiological modulation of the BBB and increased paracellular permeability (35).

We compared the optoBBTB in these two GEMMs regarding nanoparticle targeting and the BBTB opening efficiency. Our results showed that after nanoparticle administration, there was a significantly higher gold accumulation (%ID/g) in the tumor core of 73C GEMM compared with PS5A1 GEMM. Therefore, the different blood vessel phenotypes might influence the nanoparticle binding efficiency, probably due to the increased JAM-A expression in the angiogenic 73C GEMM. However, our results show that optoBBTB in PS5A1 GEMM was more efficient than in 73C GEMM, regardless of the nanoparticle targets, for example increasing Taxol delivery by 16-fold vs 5-fold in these two models, respectively. We hypothesize that the blood vessel phenotypes may also respond differently to the mechanobiological activation of the BBB (35). Therefore, PS5A1 GEMM with normal microvasculature might demonstrate a higher optoBBTB efficiency than angiogenic 73C GEMM. Further work is warranted to investigate the effect of vascular phenotype in optoBBTB.

Reference:

35. Li, X., Cai, Q., Wilson, B. A., Fan, H., Dave, H., Giannotta, M., Bachoo, R., & Qin, Z. Mechanobiological modulation of blood-brain barrier permeability by laser stimulation of endothelial-targeted nanoparticles. *Nanoscale*. **15**, 3387–3397 (2023).
59. Yuan, H., Wilson, C. M., Xia, J., Doyle, S. L., Li, S., Fales, A. M., Liu, Y., Ozaki, E., Mulfaul, K., Hanna, G., Palmer, G. M., Wang, L. V., Grant, G. A., & Vo-Dinh, T. Plasmonics-enhanced and optically modulated delivery of gold nanostars into brain tumor. *Nanoscale*. **6**, 4078–4082 (2014).
60. Praça, C., Rai, A., Santos, T., Cristovão, A. C., Pinho, S. L., Cecchelli, R., Dehouck, M. P., Bernardino, L., & Ferreira, L. S. A nanoformulation for the preferential accumulation in adult neurogenic niches. *J. Control. Release*. **284**, 57–72 (2018).
61. Choi, M., Ku, T., Chong, K., Yoon, J., & Choi, C. Minimally invasive molecular delivery into the brain using optical modulation of vascular permeability. *Proc. Natl. Acad. Sci. U. S. A.* **108**, 9256–9261. (2011).
62. Kang, P., Wang, Y., Wilson, B. A., Liu, Y., Dawkrajai, N., Randrianalisoa, J., & Qin, Z. Nanoparticle Fragmentation Below the Melting Point Under Single Picosecond Laser Pulse Stimulation. *J. Phys. Chem. C. Nanomater. Interfaces*. **125**, 26718–26730 (2021).

Below are the point-by-point responses to the comments. We hope that the revised version clarifies the concerns.

1. Plasmonic nanocarriers with and without antibodies that target JAM-A. The authors have used JAM-A targeted nanoparticles but it is not clear the importance of JAM-A targeting for the overall effect reported by the authors. It will be important to show whether gold nanoparticles without JAM-A conjugation will have the same effect in the extravasation of taxol. In addition, it is not clear what is the temperature that the laser irradiated region reaches with the 40 mJ/cm². Moreover, the authors should also evaluate what is the accumulation of taxol in animals irradiated without the administration of the JAM-A targeted nanoparticles.

Response: We thank the reviewer for this question.

We agree that it is important to show whether AuNPs without JAM-A conjugation will have the same effect in BBTB modulation and Taxol delivery. In the revised manuscript, we used AuNP-PEG as a control nanoparticle to test this hypothesis. The Taxol extravasation was analyzed using both fluorescent imaging and HPLC-MS. Briefly, the AuNPs were synthesized and functionalized with mPEG (1 kDa), and the same nanoparticle dose (18.5 µg/g and 37 µg/g for PS5A1 and 73C GEMM, respectively) were delivered to the mice intravenously. One hour later, we applied laser on the tumor area using the same laser parameters (40 mJ/cm², 1 pulse), followed by the delivery of either fluorescent Taxol646 (for fluorescent imaging) or non-fluorescent Taxol (for concentration measurement using HPLC-MS) by intravenously injection (12.5 mg/kg). For fluorescent imaging, the mice were perfused after 30 min and the brains were extracted and cryosectioned to 20 µm thick coronal slices to analyze the fluorescent Taxol646 extravasation. The results show no Taxol646 leakage in the tumor area (updated Fig. 3c and Fig. 4d). For HPLC-MS analysis, blood was collected at 1 hour after Taxol injection, and the mice were perfused with PBS. The tumors were collected to analyze the Taxol concentration. The results (Fig. S6d, S7d) show that there was no significant difference of the Taxol concentration in the tumor after optoBBTB with AuNP-PEG compared with the mice without optoBBTB treatment (Taxol only). In summary, gold nanoparticles without JAM-A conjugation will not have the same effect in the extravasation of taxol. We updated the manuscript to include this result.

To measure the temperature change in the laser irradiated region, we used a FLIR ONE Thermal Imaging Camera for smartphones, and the temperature before and after optoBBTB was recorded using Vernier Thermal Analysis™ Plus. Briefly, a region of interest (ROI) that covered the laser irradiation region was

manually selected on the app. We recorded a temperature baseline (1 min), followed by applying the laser (40 mJ/cm², 1 pulse), and then recorded for 4 min. The results show that for PS5A1 GEMM, the average temperature before and after optoBBTB was 32.1±0.1 °C and 32.3±0.2 °C. For 73C GEMM, it was 32.1±0.2 °C and 31.9±0.1°C, respectively. There was no apparent temperature increase after optoBBTB (updated Fig. S6c and Fig. S7e). We updated the manuscript to include these results.

Revision:

A. Fig. 3c and 4d, updated.

Fig. R2. OptoBBTB using AuNP-PEG does not improve the delivery of fluorescent Taxol (Taxol646) in PS5A1 (Fig. 3c) and 73C GEMM (Fig. 4d). The tumor is indicated by GFP fluorescent in Fig. 3c and Hoechst staining in Fig.4d (arrow). The scale bar represents 1 mm.

B. Figure S6d and S7d, updated.

Fig. R3. The analysis of Taxol concentration in the plasma and tumor in PS5A1 (S6d) and 73C (S7d). The results showed that after optoBBTB with AuNP-PEG followed by Taxol administration, there was no significant difference of the Taxol concentration in the tumor compared with the mice without optoBBTB treatment (Taxol only). N=3 mice. Data are expressed as Mean ± SD. The quantification analysis was performed by unpaired Student's two-sided *t* test (S6d) or One-way ANOVA followed by Tukey's multiple comparisons test (S7d), respectively.

C. Page 41, method, updated to include the local temperature measurement.

Revised: Local temperature measurement before and after optoBBTB

To measure the temperature change in the laser irradiated region, we used a FLIR ONE Thermal Imaging Camera for smartphones, and the temperature before and after optoBBTB was recorded using Vernier Thermal Analysis™ Plus. Briefly, a region of interest (ROI) that covered the laser irradiation region was manually selected on the app. We recorded a temperature baseline (1 min), followed by applying the laser (40 mJ/cm², 1 pulse), and then recorded continuously for 4 min.

D. Fig. S6c and S7e, updated

Fig. R5. Temperature change after optoBBTB in PS5A1 GEMM (S6c) and 73C GEMM (S7e). The AuNP-BV11 injection dose was 18.5 µg/g and 37 µg/g for PS5A1 and 73C GEMM, respectively. The laser fluence was 40 mJ/cm² (1 pulse). The data are expressed as Mean ± SD. N=3 replicates.

E. Page 13, updated the text.

- (a) Original: We selected 18.5 µg/g of AuNP-BV11 injection followed by 40 mJ/cm² laser fluence (1 pulse) for BBTB opening since it showed high opening efficacy with minimized nanoparticle injection (**Fig. S2A**). We further demonstrated that optoBBTB modulation allows the delivery of molecules of different sizes, such as EZ-link biotin (660 Da) and Evans blue/albumin (66 kDa) (**Fig. 3B**). The BBTB modulation was reversible and largely recovered in 1 day (**Fig. S2B**).
- (b) Revised: We selected 18.5 µg/g of AuNP-BV11 injection followed by 40 mJ/cm² laser fluence (1 pulse) for BBTB opening since it showed high opening efficacy with minimized nanoparticle injection (**Fig. S6a**). We further demonstrated that optoBBTB modulation allows the delivery of molecules of different sizes in PS5A1 GEMM, such as EZ-link biotin (660 Da) and Evans blue/albumin (66 kDa) (**Fig. 3b**). The BBTB modulation was reversible and largely recovered in 1 day (**Fig. S6b**). To investigate if there is a laser-induced heating effect in the tumor area, we recorded the local temperature change using a FLIR ONE Thermal Imaging Camera before and after optoBBTB. The results show that the average temperature before and after optoBBTB was 32.1±0.1 °C and 32.3±0.2 °C in PS5A1 GEMM (**Fig. S6c**), suggesting no apparent temperature increase after optoBBTB.

Since most chemotherapy drugs are administered over multiple doses with intervals for recovery, it is important to assess the feasibility of multiple BBTB modulations for drug delivery. Taxol is a microtubule-stabilizing drug approved by the FDA for the treatment of ovarian, breast, and lung cancer, as well as Kaposi's sarcoma (21). Following the failure of Taxol to show efficacy in an early-phase clinical trial for GBM, further testing was abandoned. However, Taxol cannot pass through the BBB, which may partly account for the lack of clinical efficacy. To investigate the effectiveness of optoBBTB in Taxol delivery, we first demonstrated that optoBBTB using AuNP-PEG with no specific targeting to TJ protein did not improve the delivery of Taxol Janelia Fluor 646 (Taxol646) into the tumor region (**Fig. 3c, Fig. S6d**). Next, we performed optoBBTB with

AuNP-BV11, followed by the administration of Taxol646 to PS5A1 GEMM for 3 times with 3 days between treatments, to investigate the impact of multiple BBTB openings during tumor treatment.

F. Page 16, updated the text.

- (a) Original: The highest BBTB opening level was achieved by injecting 36 $\mu\text{g/g}$ AuNP-BV11, followed by 40 mJ/cm^2 laser excitation (1 pulse). The BBTB recovered within 1 day, and no dye leakage into the brain was observed afterward (Fig. S6C).
- (b) Revised: The highest BBTB opening level was achieved by injecting 36 $\mu\text{g/g}$ AuNP-BV11 and 40 mJ/cm^2 laser excitation (1 pulse). The BBTB recovered within 1 day, and no dye leakage into the brain was observed afterward (Fig. S7c). Since the BBTB in 73C GEMM remained intact at 7 dpi, optoBBTB significantly improved the delivery of both small molecules (EZ-link biotin, 660 Da) and large molecules (Evans blue, 66 kDa) after i.v. injection (Fig. 4c), while BBTB modulation using AuNP-PEG did not increase the Taxol646 delivery into the tumor (Fig. 4d, Fig. S7d). The local temperature measurement shows that the average temperature before and after laser excitation was 32.1 ± 0.2 °C and 31.9 ± 0.1 °C (Fig. S7e), indicating no temperature increase after optoBBTB in 73C GEMM.

2. Accumulation of the nanoparticles in the brain and in other organs after multiple administration. In the PS5A1 GBM-bearing mice, it is not clear what is the accumulation of nanoparticles after each laser stimuli and how different is the accumulation profile to other regions in the brain. In addition, the authors do not present any data about the effect of the accumulation of the nanoparticles after multiple administrations, in particular in the liver and spleen, which according to the results presented in Fig. S1 are the organs showing higher accumulation of the nanomaterials. It is also not clear what is the degradation profile of the nanoparticles accumulated in those organs.

Response: We thank the reviewer for this question.

In the revised manuscript, we provided the gold particle accumulation analysis in the tumor and healthy brain in PS5A1 GEMM after each laser stimulation by ICP-MS analysis. Briefly, PS5A1 GEMM (n=3 mice) received three injections of AuNP-BV11 (18.5 $\mu\text{g/g}$ body weight) at 14 dpi, 18 dpi, and 22 dpi. 1 hour after the nanoparticle injection, the mice also received picosecond laser excitation (40 mJ/cm^2 , 1 pulse). 30 min later, the mice were perfused with PBS and the organs were collected for ICP-MS analysis. The updated Fig. S6e and Table S2 show the AuNP accumulation in the healthy brain and tumor after each laser stimulation. There was an increase in gold accumulation in the brain and the tumor, i.e., from 0.9 ± 0.5 $\mu\text{g/g}$ to 4 ± 1 $\mu\text{g/g}$ in the brain, and from 1.3 ± 0.4 $\mu\text{g/g}$ to 3.6 ± 1.3 $\mu\text{g/g}$ in the tumor. There was no significant difference of the gold concentration in the tumor and healthy brain (unpaired Student's two-sided *t* test, n=3).

We also collected other organs (lung, heart, kidney, liver and spleen) at the timepoints described above to analyze the nanoparticle accumulation after multiple administrations. The gold concentrations are shown in the updated Fig. S6f and Table S2. Although there was an increase in the gold accumulation in these organs, especially in the liver and spleen, we did not observe toxicity side effect such as body weight loss (Fig. 5l).

Since there was no difference of the gold accumulation in the tumor and healthy brain in PS5A1 GEMM, we used healthy Nu/J mice to study the nanoparticle degradation profile, in order to track the long-term NP degradation. Nu/J mice (n=3) received 3 AuNP-BV11 injections (18.5 $\mu\text{g/g}$, with 3 days interval) and laser excitation (40 mJ/cm^2 , 1 pulse) as described above. The mice were perfused at 60 days after the third

nanoparticle injection and laser treatment. The organs were collected and digested for ICP-MS analysis. The results (Fig. S6f) show that there was a slightly decrease in the gold concentration in most of the organs, for example, the gold accumulation in the liver was $960 \pm 180 \mu\text{g/g}$ after the 3rd optoBBTB, and 60 days later it decreased to $840 \pm 70 \mu\text{g/g}$. The slow gold clearance profile in mice is in agreement with other literature (*Nanomedicine*. 2009, 5(2), 162–169; *Proc Natl Acad Sci U S A*. 2017, 114(15), E3110; *Part Fibre Toxicol*. 2014, 30;11:26.). We updated Fig. S6e-f, and added Table S2 to include these results.

Revision:

A. Fig. S6E and S6F updated.

Fig. R5. AuNP biodistribution analysis in PS5A1 GEMM. (S6e) The brain and tumor accumulation of AuNP-BV11 in PS5A1 GEMM after each optoBBTB analyzed by Inductively Coupled Plasma Mass Spectroscopy (ICP-MS). N=3 mice, data are expressed as Mean \pm SD. No significant difference was found between the brain and the tumor, analyzed by unpaired Student’s two-sided *t* test. (S6f) The biodistribution of AuNP-BV11 in PS5A1 GEMM after each optoBBTB and the nanoparticle degradation analyzed by ICP-MS. N=3 mice, data are expressed as Mean \pm SD.

B. Table S2, updated.

Table S2. Gold concentration in PS5A1 GEMM after three optoBBTB. Unit: μg (Au)/g (tissue).

	Brain	Tumor	Blood	Kidney	Heart	Lung	Spleen	Liver
1 st optoBBTB	0.9 \pm 0.5	1.3 \pm 0.4	13 \pm 9	2.0 \pm 0.7	6 \pm 3	18 \pm 4	250 \pm 40	370 \pm 50
2 nd optoBBTB	2.6 \pm 0.9	2 \pm 1	13.9 \pm 1.4	3.4 \pm 0.6	13.5 \pm 1.8	32 \pm 3	370 \pm 50	670 \pm 30
3 rd optoBBTB	4 \pm 1	3.6 \pm 1.3	19 \pm 14	5.4 \pm 1.2	25 \pm 3	38 \pm 16	610 \pm 160	960 \pm 180
60 days after the 3 rd optoBBTB	2.3 \pm 1.1	N/A	1.3 \pm 0.5	7.4 \pm 2.2	18 \pm 4	32 \pm 5	620 \pm 80	840 \pm 70

C. Page 14, result, updated the text.

- (a) Original: Each BBTB modulation resulted in Taxol delivery in the tumor core and margin regions. Notably, there was no evidence of fluorescent Taxol leakage in the contralateral hemisphere, which serves as an internal control and reconfirms the inability of this drug to pass through the normal BBB. Therefore, optoBBTB can be repeated and allows a multiple-cycle treatment regimen.
- (b) Revised: Each BBTB modulation resulted in Taxol delivery in the tumor core and margin regions. Notably, there was no evidence of fluorescent Taxol leakage in the contralateral hemisphere, which reconfirmed the inability of this drug to pass through the normal BBB. We further analyzed the bioaccumulation and biodegradation of the gold nanoparticles in the tumor and healthy brain in PS5A1 GEMM after each optoBBTB by Inductively Coupled Plasma Mass Spectrometry (ICP-MS). The result showed that there was an increased gold accumulation in the brain and the tumor, i.e., from $0.9 \pm 0.5 \mu\text{g/g}$ to $4 \pm 1 \mu\text{g/g}$ in the brain and from $1.3 \pm 0.4 \mu\text{g/g}$ to $3.6 \pm 1.3 \mu\text{g/g}$ in the tumor. No significant difference in the gold concentration was observed in the tumor and healthy brain (Fig. S6e, Table S2). Moreover, the slow gold clearance profile in mice is in agreement with the literature (Fig. S6f) (32-34). In summary, optoBBTB can be repeated and allows a multiple-cycle treatment regimen in PS5A1 GEMM that recapitulates the tumor margin histopathological characteristics.

D. Page 40, method, update the text.

- (a) Original: ICP-MS was used to determine the biodistribution of AuNP-BV11 ($37 \mu\text{g/g}$) after intravenous (i.v.) injection to the 73C glioma-bearing mice (7 dpi). 1 hour after nanoparticle injection, the mice were perfused with ice-cold PBS, and the main organs were collected. The tissue was then digested in fresh aqua regia until the tissue was fully dissolved. Then the solution was centrifuged at 5000 g for 10 min, and the supernatant was collected and diluted with ultrapure water for ICP-MS analysis (Agilent 7900).
- (b) Revised: We used ICP-MS to measure the AuNP-BV11 ($18.5 \mu\text{g/g}$) accumulation in PS5A1 GEMM (14 dpi) after each optoBBTB (40 mJ/cm^2 , 1 pulse, repeated 3 times at 14, 18, and 22 dpi). We also studied the nanoparticle degradation profile using healthy Nu/J mice. The mice received 3 nanoparticle injections and laser treatments with 3 days interval, and the gold concentration in each organ were measured at 60 days after the third nanoparticle injection and laser excitation. ICP-MS was also used to determine the biodistribution of AuNP-BV11 ($37 \mu\text{g/g}$) after intravenous (i.v.) injection to the 73C glioma-bearing mice (7 dpi). 1 hour after nanoparticle injection, the mice were perfused with ice-cold PBS, and the main organs were collected. The tissue was then digested in fresh aqua regia until the tissue was fully dissolved. Then the solution was centrifuged at 5000 g for 10 min, and the supernatant was collected and diluted with ultrapure water for ICP-MS analysis (Agilent 7900).

3. Accumulation of taxol in the brain and the mechanism underlying the reduction of tumor volume.

Although the authors quantify the taxol that extravasates by fluorescence, it is not clear the concentration of the drug that indeed crosses the BBB. Perhaps the authors can quantify the concentration by HPLC or other methodologies. The authors should also clarify whether the reduction of tumor volume is only mediated by the extravasated taxol or by other mechanisms (e.g. activation of the immune system by the heating effect).

Response: We thank the reviewer for this question.

In the revised manuscript, we provided the quantification of Taxol concentration in the tumor without or with optoBBTB using HPLC-MS, in both PS5A1 and 73C GEMMs. The results show that in the PS5A1

GEMM (14 dpi), the Taxol concentration in the tumor without or with optoBBTB was 12 ± 15 ng/g and 185 ± 92 ng/g, respectively, indicating a 16-fold concentration increase after optoBBTB. In 73C GEMM (7 dpi), the Taxol concentration in the tumor without or with optoBBTB is 240 ± 168 ng/g and 1206 ± 1094 ng/g, respectively, indicating a 5-fold concentration increase after optoBBTB.

Since our control groups (optoBBTB+Vehicle) did not show a reduction of tumor volume (Fig. 5-6), and the local temperature was not enhanced after optoBBTB (Fig. S6c and Fig. S7e), we believe that the reduced tumor volume is mainly mediated by the extravasation of Taxol. We have updated the manuscript to clarify this finding.

Revision:

A. Fig. 5d and 6d, updated.

Fig. R6. The analysis of Taxol concentration in the tumor without optoBBTB or with optoBBTB in PS5A1 (5d) and 73C (6d), N=3 mice. Data are expressed as Mean \pm SD. The quantification analysis was performed by unpaired Student's two-sided *t* test.

B. Page 21, result, added text.

- (a) Original: The data shows that optoBBTB greatly enhanced the delivery of fluorescent Taxol646 in the tumor core and margin compared to no optoBBTB treatment (**Fig. 5B-C**). These tumors showed no T1-weighted contrast enhancement by MRI consistent with an intact BBTB and minimal T2-weighted hyperintensity (**Fig. S10A**).
- (b) Revised: The data shows that optoBBTB greatly enhanced the delivery of fluorescent Taxol646 in the tumor core and margin compared with no optoBBTB treatment (**Fig. 5b, c**). **The Taxol concentration in the tumor without or with optoBBTB was 12 ± 15 ng/g and 185 ± 92 ng/g, respectively, indicating a 16-fold concentration increase after optoBBTB (Fig. 5d).** These tumors show no T1-weighted contrast enhancement by MRI consistent with an intact BBTB and minimal T2-weighted hyperintensity (**Fig. S10a**).

C. Page 25, result, added text.

- (a) Original: **Fig. 6B-C** shows that a single dose of optoBBTB enhanced the delivery of Taxol to the tumor core and margin compared to no optoBBTB treatment. This enhanced Taxol delivery produced a statistically significant difference in slowing the tumor progression and increased survival (**Fig. 6E-I**).

- (b) Revised: **Fig. 6b, c** show that a single dose of optoBBTB enhances the delivery of Taxol to the tumor core and margin compared with no optoBBTB treatment. **The Taxol concentration in the tumor without or with optoBBTB was 240 ± 168 ng/g and 1206 ± 1094 ng/g, indicating a 5-fold concentration increase after optoBBTB (Fig. 6d).** The enhanced Taxol delivery produced a statistically significant difference in slowing the tumor progression and increasing survival (**Fig. 6e-i**).

D. Page 31, discussion, added text.

Revised: Although modulation of the BBB permeability using the laser with/without plasmonic nanoparticles has been reported elsewhere (59-61), our method has three critical differences compared with these strategies. First, our AuNP-BV11 nanoparticles binds to the vasculature and optoBBB modulation approach involves the increase of paracellular permeability without causing cell damage. Second, optoBBTB exploits nanoscale mechanical perturbation rather than the heating effect (62). Since the duty cycle of the laser is low, the amount of heat dissipated from the nanoparticle into the surrounding medium is insufficient to raise tissue temperature. Last, since our control groups (optoBBTB+Vehicle) do not show a reduction of tumor volume or an increased survival rate, and the local temperature was not enhanced after laser excitation, we believe that the reduced tumor volume is mainly mediated by the extravasation of Taxol rather than by other mechanisms such as the activation of the immune system by the heating effect. Recently, we unveiled the mechanisms of the optical BBB modulation using an in vitro BBB model established with human cerebral microvascular endothelial cells. We showed that the picosecond laser excitation of vascular-targeting AuNPs produced nanoscale mechanical perturbation, which triggers several mechanobiological responses, including (1) actin polymerization that leads to the cytoskeletal contraction, (2) Ca^{2+} -influx including from mechanosensitive ion channels (such as TRPV4 and Piezo1), and (3) the activation of Inositol Trisphosphate (IP3) pathway and the Ca^{2+} release from the endoplasmic reticulum. The elevation of Ca^{2+} from the extracellular influx and intracellular IP3 pathway activates ERK1/2 phosphorylation. These effects led to a mechanobiological modulation of the BBB and increased paracellular permeability (35).

4. BBB maturity and extravasation profile in the 73C GEMM mouse model. I wonder if the authors could clarify whether occludin expression is reduced in the tumor core as they observed in human biopsies (Fig. 1). The authors show that the blood vessels in the tumor core are immature likely due to alterations in the expression of ZO-1 at protein level. I wonder if the low staining is due to alterations in the structure of ZO-1 protein or potential artifacts. To further confirm this effect, the reviewer suggests the authors to confirm the decrease in ZO-1 expression by transcriptomic analyses. Moreover, if the blood vessels in this mouse model are immature, and thus leakier, I wonder what is the mechanism behind the opto-activation of the nanoparticles.

Response:

We thank the reviewer for this comment.

In the revised manuscript, we provided analysis on occludin expression in 73C GEMM by fluorescent imaging. 73C tumor-bearing mice were sacrificed at 7, 14 and 21 dpi, the brains were collected and cryosectioned to 20 μm thickness coronal slices for IHC staining. The blood vessels were co-stained with CD31. We observed the occludin immunofluorescent at these timepoints, and there is no difference of the occludin coverage on blood vessels (occludin/CD31) in the tumor core, margin and the contralateral side. We updated Fig. S4 to include this result. Since there is no evidence shows that the protein undergoes a post translational modification of the antigen recognition site, therefore it is unlikely that the decrease in

the ZO-1 immunofluorescence is due to the structure alteration. The fact that ZO-1 immunofluorescence is clearly visible in the tumor margin and contralateral side reduces the possibility of artifact.

To study the mechanisms of the BBTB modulation in 73C GEMM, we analyzed the changes in the blood vessels after optoBBTB using IHC staining. Briefly, AuNP-BV11 (37 $\mu\text{g/g}$) was intravenously delivered to the 73C tumor-bearing mice. 1 hour later, the mice received a single pulse of picosecond laser excitation (40 mJ/cm^2). 30 min later, the brains were collected and cryosectioned to 20 μm thick coronal slices. The blood vessels were stained with CD31, and the blood vessel coverage in the tumor area was visualized using a 10x objective. The results showed that before and after optoBBTB, there was no significant difference of the blood vessel density in tumor core and margin (updated Fig. S3a, N=15 images from 3 mice, unpaired Student's two-sided *t* test). We further investigated the influence on other key junctional proteins such as Claudin-5, VE-Cadherin, Occludin, and JAM-A by IHC staining at 30 min after optoBBTB. As shown in updated Fig. S4b, there was no significant difference of the area fraction ratio of protein over blood vessel (i.e., CLDN5/CD31, VE-Cad/CD31, Occludin/CD31, and JAM-A/CD31) before and after treatment in tumor core and margin, comparing to contralateral side (N=15 images from 3 mice, unpaired Student's two-sided *t* test). Therefore, these results suggest that there are no changes in the density or the immunofluorescent of the junctional protein after optoBBTB.

According to our recent work, we hypothesize that the increase in BBTB permeability is due to the Ca^{2+} -mediated activation of mechanobiological pathways and the re-organization of the cellular skeleton (*Nanoscale*, 2023,15, 3387-3397). Our ongoing work is focusing on investigating how the blood vessel phenotypes respond to the optoBBTB.

Revision:

A. Fig. S3a, updated.

Fig. S3a. IHC staining and quantification of blood vessels in 73C GEMM using CD31 at 7-21 dpi. The cell nuclei are labeled with Hoechst staining (HOE). The scale bars represent 100 μm . Quantification of blood vessel coverage was performed by CD31 area fraction. N=15 images from 3 mice. Data in the box and whisker plots are given from the minima to maxima, the bounds of the box represent the 25th percentile and 75th percentile, and the middle line of the box is the median. Quantification analysis was performed with One-way ANOVA followed by Tukey's multiple comparisons test. Source data are available as a Source Data file.

B. Fig. S4b, updated.

Fig. S4b. The quantification analysis of the expression of junctional proteins over CD31 before and after optoBBTB by area fraction. N=15 images from 3 mice. Data in the box and whisker plots are given from the minima to maxima, the bounds of the box represent the 25th percentile and 75th percentile, and the middle line of the box is the median. Quantification analysis was performed with unpaired Student's two-sided *t* test. Source data are available as a Source Data file.

C. Page 38, method, updated.

- (a) Original: To immunostaining vascular biomarker (CD31) and junctional proteins (i.e., Claudin-5, ZO-1, VE-cadherin, and JAM-A), the mice brains were snap-frozen on dry ice once quickly removed from the skull and cut to 20 μm thick coronal slices on a cryostat. The brain slices were fixed for 10 min using ice-cold methanol at -20 $^{\circ}\text{C}$.
- (b) Revised: To immunostaining vascular biomarker (CD31) and junctional proteins (i.e., Claudin-5, ZO-1, VE-cadherin, Occludin, and JAM-A), the mice brains were snap-frozen on dry ice once quickly removed from the skull and cut to 20 μm thick coronal slices on a cryostat. To analyze the influence of optoBBTB on vessel density and the immunofluorescent of the junctional proteins, the brains were collected at 30 min after the optoBBTB, followed by cryosectioned to 20 μm thick coronal slices. The brain slices were fixed for 10 min using ice-cold methanol at -20 $^{\circ}\text{C}$.

D. Page 9, result, updated the text.

- (a) Original: The irregular structure of the microvasculature associated with poor hemodynamics and high metabolic demands of the tumor mass, creates an environment of relative hypoxia which contributes to tumor angiogenesis and often regions of necrosis, a pathognomonic histologic feature of GBM. IHC staining of junctional proteins showed that the immunofluorescence of Claudin-5, VE-Cadherin, and JAM-A persisted during 7-21 dpi at both tumor core and margin (Fig. S5D, Fig. 5C). However, there was a significantly lower level of ZO-1 expression at the tumor core at 14 and 21 dpi (Fig. 5D), consistent with the observation in human GBM (Fig. 1E).
- (b) Revised: The irregular microvasculature structure, associated with poor hemodynamics and high metabolic demands of the tumor mass, creates an environment of relative hypoxia which contributes to tumor angiogenesis and often regions of necrosis, a pathognomonic histologic feature of GBM. IHC staining of junctional proteins showed that the immunofluorescent of

Claudin-5, VE-Cadherin, JAM-A, and Occludin persisted during 7-21 dpi at both tumor core and margin (Fig. S4a, Fig. 2c). However, there was a significantly lower level of ZO-1 expression at the tumor core at 14 and 21 dpi (Fig. 2d). Further quantification analysis of the area fraction ratio of protein over blood vessel (CD31) suggested that the relative protein coverage ratio for Claudin-5, VE-Cadherin, JAM-A, and Occludin was comparable at the tumor core, margin, and contralateral side during 7-21 dpi. Although no apparent change in the ZO-1/CD31 was observed at 7 dpi, there was a significant decrease in this ratio at the tumor core and margin at 14 and 21 dpi (Fig. 2d). We speculate that the BBTB disruption in the tumor core during disease progression is partially due to the loss of ZO-1 coverage on immature newly formed vessels.

E. Page 17, added text.

Revised: We noted that the BBTB modulation displayed a higher efficiency in the PS5A1 GEMM than in the 73C GEMM (Fig. 3, 4, S6, S7), although there was a significantly higher AuNP-BV11 accumulation in the tumor core of 73C GEMM compared with PS5A1 GEMM (0.32 ± 0.06 %ID/g, and 0.18 ± 0.03 %ID/g, respectively, Table S4). To increase the BBTB opening efficiency in the 73C GEMM, we attempted to functionalize AuNPs with other vasculature targets, such as the anti-vascular endothelial growth factor 2 (VEGFR2) antibody and the anti-transferrin receptor (TfR) antibody, since VEGFR2 and TfR were overexpressed in 73C GEMM (Fig. S8a, b). However, these nanoparticles did not improve BBTB opening efficiency compared with AuNP-BV11 (Fig. S8c, d). To probe the mechanisms of the optoBBTB, we analyzed the changes in the irregular blood vessels in 73C GEMM after laser stimulation using IHC staining. The blood vessel density analysis show that optoBBTB did not influence the vessel coverage percentages in the tumor core and margin (Fig. S3a). Moreover, no significant difference in the immunofluorescent of junctional protein was observed before and after optoBBTB (Fig. S4b). These results suggest that optoBBTB in 73C GEMM did not influence the density or the junctional protein immunofluorescent of the angiogenic blood vessels. In our recent work (31), we demonstrated that laser excitation of vascular-targeting AuNPs was associated with a transient elevation and propagation of Ca^{2+} , actin polymerization, and phosphorylation of ERK1/2 (extracellular signal-regulated protein kinase). They collectively activated the cytoskeleton resulting in increased paracellular permeability. We hypothesize that the increased barrier permeability after optoBBTB is due to the Ca^{2+} -mediated activation of the mechanobiological pathways and the re-arrangement of the cytoskeleton. Moreover, angiogenic blood vessels may respond differently to optoBBTB than normal brain microvasculature. Further investigation may be focused on examining how the blood vessel phenotypes respond to optoBBTB and change the barrier permeability.

5. Leakage of taxol from tumor blood vessels in the absence of BBTB modulation (73C GEMM mouse model). The authors demonstrate in Fig. 5A and 5B that BBTB at day 14 dpi is leaky and allows the extravasation of EZ-link biotin (600 Da) and Evans blue (66 kDa) into the brain parenchyma. Thus, it is not clear why in Fig. S6C the authors do not observe dye (Biotin) leakage into the tumor without BBTB modulation. The authors should also quantify the leakage of taxol in conditions without BBTB modulation (i.e. without laser activation).

Response: We thank the reviewer for this comment.

We repeated the experiment in the original Fig. S6c (tumor was injected to both sides of the brain, optoBBTB was applied to the left hemisphere, followed by i.v. injection of Taxol). Although we observed smaller tumor size in the left hemisphere (with optoBBTB and systematic Taxol delivery) at 12 dpi, we occasionally observed biotin leakage in the right hemisphere as well (with tumor but without optoBBTB). Since this is an extremely aggressive tumor, it is likely that the BBTB permeability is in a transient stage

during this time period. However, it does not influence our conclusion on the efficiency of optoBBTB in GBM treatment. We removed Fig. S6c from our revised manuscript.

In the revised manuscript, we provided the quantification of Taxol concentration in 73C GEMM, without BBTB modulation at 7 and 14 dpi, by HPLC-MS (Taxol only). The results show that the Taxol concentration in the tumor was 240 ± 168 ng/g at 7 dpi, and it was 1148 ± 577 ng/g at 14 dpi.

Revision:

A. Fig. S7d, updated.

Fig. S7d. The analysis of Taxol concentration in 73C GEMM. N=3 mice. Data are expressed as Mean \pm SD. The quantification analysis was performed by One-way ANOVA followed by Tukey's multiple comparisons test.

6. In the discussion section, the authors should clarify what are the advantages and limitations of this opto-activation strategy relatively to ultrasound and other approaches documented in the literature to open the BBB at specific sites.

Response: We thank the reviewer for this comment. The advantages of optoBBTB compared with other methods has been added to the discussion. We also mentioned in the discussion that due to the limitation of the light penetration depth in the brain, our method has the advantages in targeting superficial brain tumors and CNS regions with easier fiberoptic access such as spinal cord. We provided alternative solutions to optimize the optoBBTB for treating brain tumors in deep brain.

Revision:

A. Page 31, updated the text.

- Original: Since the intercellular TJs represent a formidable barrier against paracellular drug delivery at the BBB (36, 37), approaches have been developed to modulate the TJ to enhance the delivery across the BBB, including co-administration of siRNA against claudin-5 and occludin, as well as exploiting claudins or cadherin inhibitory peptides (38-41). However, a lack of a robust delivery system in human, poor targeting efficacy, or a lack of site-specificity impedes the successful translation of these approaches. Here we demonstrated that optoBBTB specifically targeted the JAM-A component of the TJ to modulate the BBTB locally and reversibly, and multiple openings could be achieved for anticancer drug delivery.
- Revised: Since the intercellular TJs represent a formidable barrier against paracellular drug delivery at the BBB (13, 52), approaches have been developed to modulate the TJ to enhance the delivery across the BBB, including co-administration of siRNA against claudin-5 and occludin, as

well as exploiting claudins or cadherin inhibitory peptides (53-56). However, a lack of a robust delivery system in human, poor targeting efficacy, or a lack of site-specificity impedes the successful translation of these approaches. Here we demonstrated that optoBBTB specifically targeted the JAM-A component of the TJ to modulate the BBTB locally and reversibly, and multiple openings could be achieved for anticancer drug delivery. Compared with the above-mentioned TJ modulation approaches, optoBBTB demonstrates advantages such as high targeting efficiency and site-specificity. Focused ultrasound (FUS) with circulating microbubbles (MB) is an emerging approach to modulate BBB permeability non-invasively and reversibly (16, 57). OptoBBTB is a complimentary approach and offers the feasibility to tune the laser beam size, enabling the BBB modulation in larger or smaller areas. Furthermore, it may be utilized to open CNS barriers such as in the spinal cord that has been challenging for ultrasound penetration due to the complicated bone structures, but is straightforward with fiberoptic light delivery (58).

Reference:

57. Chen, K. T., Lin, Y. J., Chai, W. Y., Lin, C. J., Chen, P. Y., Huang, C. Y., Kuo, J. S., Liu, H. L., & Wei, K. C. Neuronavigation-guided focused ultrasound (NaviFUS) for transcranial blood-brain barrier opening in recurrent glioblastoma patients: clinical trial protocol. *Ann. Trans. Med.* **8**, 673 (2020).
58. Busch, D. R., Davis, J., Kogler, A., Galler, R. M., Parthasarathy, A. B., Yodh, A. G., & Floyd, T. F. Laser safety in fiber-optic monitoring of spinal cord hemodynamics: a preclinical evaluation. *J. Biomed. Opt.* **23**, 1–9 (2018).

B. Page 33, updated the text.

- (a) Original: Among the various methods to change the BBTB permeability, optoBBTB presents unique opportunities for further development and clinical translation. First, near-infrared light-absorbing nanoparticles can be exploited since the light in this region exhibits deeper tissue penetration to cover the tumor margin in a larger animal model. Second, while light can be delivered transcranially in the mouse brain, fiber delivery to the human brain is envisioned, especially after surgical removal of the primary tumor. Placing an optical fiber in the tumor surgical cavity would allow the delivery of side-emitting light to the tumor margin (**Fig. S13**). Moreover, extracranial light delivery is within the realm of implementation with a transparent cranial window to replace a portion of the skull (54). All these developments will facilitate the next-stage translation of optoBBTB for GBM treatment going forward.
- (b) Revised: Among the various methods to change the BBTB permeability, optoBBTB presents unique opportunities for further preclinical and clinical investigations. In preclinical settings, our method can be useful as a drug development and screening platform (optoBBTB and GEMMs) for testing a class of potent anticancer drugs. In terms of clinical investigations, there are several opportunities for further development. First, the 532 nm light exploited in the current study enables light delivery to the mouse cortex and therefore, the treatment of cortically-located tumors. However, to treat brain tumor in the deep brain region, near-infrared light-absorbing nanoparticles can be exploited since the light in this region exhibits deeper tissue penetration to cover the tumor margin in a larger animal model. Second, while light can be delivered transcranially in the mouse brain, fiber delivery to the human brain is envisioned, especially after the surgical removal of the primary tumor. Placing an optical fiber in the tumor surgical cavity would allow the delivery of side-emitting light to the tumor margin far from the tumor mass (**Fig. S13**). Moreover, since human skulls are significantly thicker than mice skulls, extracranial light delivery is within the realm of implementation with a transparent cranial window to replace a portion of the skull (63). Furthermore, our recent work investigated opening the blood-spinal cord

which represent another important application of the technology and has advantages compared with the state-of-the-art methods (64). Further work is ongoing to investigate tumor treatment in this area.

Minor issues:

7. In page 4 (first paragraph), I have the impression that reference 13 should be replaced by reference 17.

Response:

We thank the reviewer for this comment. The reference list has been checked and updated.

8. I could not find reference to the wavelength of the laser used.

Response:

We used 532 nm picosecond laser for BBTB modulation. In this study, we used 50 nm spherical gold nanoparticles. After conjugating with the anti-JAM-A antibody (BV11), the nanoparticle shows a surface plasmon resonance peak around 530 nm, which matches well with our 532 nm picosecond laser. This information has been added to the revised manuscript.

Revision:

Page 13, updated the text.

- (a) Original: These nanoparticles were i.v. injected into a tumor-bearing mouse, followed by the delivery of a transcranial picosecond laser pulse to the tumor region to stimulate the AuNPs for BBTB modulation (optoBBTB, **Fig. 3A**). Fluorescent dyes or therapeutics were then administrated to assess the BBTB permeability and the brain uptake. To optimize the optoBBTB, a series of nanoparticle doses and laser fluences were tested (**Table S1**).
- (b) Revised: These nanoparticles were i.v. injected into a tumor-bearing mouse, followed by the delivery of a transcranial 532 nm picosecond laser pulse to the tumor region to stimulate the AuNPs for BBTB modulation (optoBBTB, **Fig. 3a**). The 532 nm picosecond laser was exploited for optoBBTB since the wavelength matches well with the surface plasmon resonance peak of the 50 nm spherical gold nanoparticles (530 nm). Fluorescent dyes or therapeutics were then delivered to assess the BBTB permeability and brain uptake. To optimize the optoBBTB, a series of nanoparticle doses and laser fluences were tested (**Table S1**).

REVIEWERS' COMMENTS

Reviewer #1 (Remarks to the Author):

The authors have addressed all of the comments of this reviewer and should be congratulated for an exciting study that will move the field of brain tumour treatment research forward.

Reviewer #2 (Remarks to the Author):

Qin et al. have provided detailed responses to my queries. Concerning the questions posed originally by myself, they have been addressed.

However, the central weakness of this manuscript is that the approach only works for small superficial rodent tumors. Most glioblastomas are deep tumors. Thus, the technique shown here will not be applicable to real human tumors. The authors try to argue the point that modifications of this technique may have broader treatment capabilities. This argument is irrelevant to the technique proposed in this manuscript. I regret to say that this manuscript is about a technique that has no prospect of ever being used in human patients. As such I cannot recommend further consideration of this manuscript.

Reviewer #3 (Remarks to the Author):

Comments for Author

Key Results

The authors report the use of an optical method called 'optoBBTB' to noninvasively increase BBB permeability in a local region of the brain near the surface. The optoBBTB method was described in a previous publication and is comprised of transcranial pulsed laser excitation of gold nanoparticles. The noteworthy results in this study are the application of optoBBTB in two mouse models of GBM, and in the revised manuscript this aspect is made much more clear. I appreciate the change to make comparison between the two models more readily apparent, and the characterization of the models, including methods and phenotype, adds a great deal to the manuscript's findings.

The addition of statistical analyses and improved presentation of fluorescence images has strengthened the manuscript as well. The discussion is quite long and winding, and not all the references and avenues are particularly relevant. Shortening the discussion to directly relate to the data presented and discussing limitations candidly would improve the readability.

Overall, my primary concerns with the initial manuscript have been addressed, and the manuscript is significantly improved.

However, there remain a few important points that require clarification or toning down the claims:

- The authors were asked to provide evidence that the PS5A1 and 73C cells used to generate the mouse models were genotyped and retain the BRAFV600E and other alterations and I was not able to find this in the new manuscript or SI. This is a key piece of information to enable reproducibility, and should be readily accessible since the cells were manipulated with Lenti-GFP and presumably maintained in culture prior to orthotopic injection.
- It appears that additional studies were performed to investigate the impact of optoBBTB on junctional proteins and the authors conclude that there was no impact. This result should be qualified in the discussion to note that this was assessed at one time point, 30 minutes after optoBBB, which may not be enough time to see protein changes depending on the mechanism of action. The authors' claim that the BBB opening is reversible and does not cause cell damage needs to be toned down .
- A section in the discussion on limitations of this technology should be more explicit – lines 565-589 are vague and wordy and It is better to address head on that this technology is currently limited to superficial tumors. The discussion of spinal cord does not seem particularly relevant.
- In regards to potential mechanisms, the authors' addition in the discussion of new information they recently published elsewhere (547-557) using an in vitro model does not add significantly to this paper – unless they investigate these mechanisms in the current dataset –and could be shortened.

Minor points:

- There is inconsistency in labeling of figure panels, particularly in regards to the box-and-whisker plots which are sometimes labeled as a separate panels but often not, making it difficult for the reader to quickly determine what each panel is referring to in the caption.
- Figure 5c and 6c– what do the gray versus green bars refer to? They do not appear to be labeled in the figure or discussed in the legend. In figure 5b and 5f, the color for vehicle and taxol are inconsistent – was this purposeful? It will aid readers to be consistent in labeling each group when they are comparable.
- Line 132-135 – The rationale for using immunodeficient mice is not strong, as it does not appear the authors making direct comparisons with their data to other GBM treatment studies? The choice of nude mice does not invalidate the findings of the study, but may warrant mentioning in the discussion as a limitation to extending these results to immunocompetent models without further study. There is mention in the discussion (line 547) noting the lack of immune response in this model which should be qualified further.

- Lines 252-255 – The addition of temperature acquisition is important, and a welcomed addition, but it is not clear how this was measured – is the camera able to detect temperature changes at the level of skin, bone, or underlying brain tissue?
- Further analysis of the biodistribution of gold NPs in each model is appreciated, but is there a reason that data is presented in different units for each model (ug/g tissue versus %ID?)
- Line 548: there appears to be a typo “unrevealed”

Reviewer #4 (Remarks to the Author):

The paper entitled “Optical Blood-Brain-Tumor Barrier Modulation Enhances Drug Penetration and Therapeutic Outcome in Clinically Relevant infiltrative and Angiogenic Glioblastoma Models” is an interesting paper which describes the modulation of the blood brain barrier using plasmonic carriers to facilitate the extravasation of anticancer drugs. The authors have done a relevant effort to address the points raised in a previous round of revisions. There are still relevant cavities in the current version related with the BBB opening mechanism by the formulation and whether the concentration of taxol that accumulates in the tumor (ng of taxol per g of animal) is indeed relevant for a biological action.

Response to reviewers

Reviewer #1 (Remarks to the Author):

The authors have addressed all of the comments of this reviewer and should be congratulated for an exciting study that will move the field of brain tumour treatment research forward.

Response: We thank the reviewer for their recommendation.

Reviewer #2 (Remarks to the Author):

Qin et al. have provided detailed responses to my queries. Concerning the questions posed originally by myself, they have been addressed.

However, the central weakness of this manuscript is that the approach only works for small superficial rodent tumors. Most glioblastomas are deep tumors. Thus, the technique shown here will not be applicable to real human tumors. The authors try to argue the point that modifications of this technique may have broader treatment capabilities. This argument is irrelevant to the technique proposed in this manuscript. I regret to say that this manuscript is about a technique that has no prospect of ever being used in human patients. As such I cannot recommend further consideration of this manuscript.

Response: We thank the reviewer for their comment. We revised our discussion to address the limitations of our mouse models and light detection depth. We also toned down the perspectives on translational relevance accordingly.

Revision:

Page 20, discussion, update the text:

- (a) Original: Among the various methods to change the BBTB permeability, optoBBTB presents unique opportunities for further preclinical and clinical investigations. In preclinical settings, our method can be useful as a drug development and screening platform (optoBBTB and GEMMs) for testing a class of potent anticancer drugs. In terms of clinical investigations, there are several opportunities for further development. First, the 532 nm light exploited in the current study enables light delivery to the mouse cortex and therefore, the treatment of cortically-located tumors. However, to treat brain tumor in the deep brain region, near-infrared light-absorbing nanoparticles can be exploited since the light in this region exhibits deeper tissue penetration to cover the tumor margin in a larger animal model. Second, while light can be delivered transcranially in the mouse brain, fiber delivery to the human brain is envisioned, especially after the surgical removal of the primary tumor. Placing an optical fiber in the tumor surgical cavity would allow the delivery of side-emitting light to the tumor margin far from the tumor mass (Fig. S13). Moreover, since human skulls are significantly thicker than mice skulls, extracranial light delivery is within the realm of implementation with a transparent cranial window to replace a portion of the skull (63). Furthermore, our recent work investigated opening the blood-spinal cord which represent another important application of the technology and has advantages compared

with the state-of-the-art methods (64). Further work is ongoing to investigate tumor treatment in this area.

- (b) Revised: Among the various methods to change the BBTB permeability, optoBBTB presents unique opportunities for further investigations. The 532 nm light exploited in the current study enables light delivery to the mouse cortex, and therefore, our method can be useful as a drug development and screening platform (optoBBTB and GEMMs) for testing a class of potent anticancer drugs for superficial tumors. Several approaches can be exploited to further advance the technique, such as utilizing near-infrared laser and near-infrared light absorbing nanoparticles to improve the light penetration depth in the tissue, or using optical fiber in the tumor surgical cavity for light delivery into deeper brain regions.

Reviewer #3 (Remarks to the Author):

Comments for Author

Key Results

The authors report the use of an optical method called ‘optoBBTB’ to noninvasively increase BBB permeability in a local region of the brain near the surface. The optoBBTB method was described in a previous publication and is comprised of transcranial pulsed laser excitation of gold nanoparticles. The noteworthy results in this study are the application of optoBBTB in two mouse models of GBM, and in the revised manuscript this aspect is made much more clear. I appreciate the change to make comparison between the two models more readily apparent, and the characterization of the models, including methods and phenotype, adds a great deal to the manuscript’s findings.

The addition of statistical analyses and improved presentation of fluorescence images has strengthened the manuscript as well. The discussion is quite long and winding, and not all the references and avenues are particularly relevant. Shortening the discussion to directly relate to the data presented and discussing limitations candidly would improve the readability.

Response: We thank the reviewer for the suggestion. The discussion has been shortened. Specifically, we addressed the limitations of our mouse models and light detection depth. We also toned down the perspectives on translational relevance. The changes are listed in the responses below.

Overall, my primary concerns with the initial manuscript have been addressed, and the manuscript is significantly improved.

However, there remain a few important points that require clarification or toning down the claims:

- The authors were asked to provide evidence that the PS5A1 and 73C cells used to generate the mouse models were genotyped and retain the BRAFV600E and other alterations and I was not able to find this in the new manuscript or SI. This is a key piece of information to enable reproducibility, and should be readily accessible since the cells were manipulated with Lenti-GFP and presumably maintained in culture prior to orthotopic injection.

Response: We thank the reviewer for the suggestion. The genotyping results have been provided in Supplementary Fig. 2.

Revision:

Supplementary Fig. S2. Genotyping of the cell lines by Polymerase chain reaction (PCR). **a** PS5A1 cell line carried conditional floxed tumor suppressor genes PTEN^{f/f} and INK4a/b.Ar^{f/f} along with the conditional (lox-stop-lox) Braf^{V600E f/+}. **b** 73C cell line carried (lox-stop-lox) Braf^{V600E f/+}, P53^{f/f} and PTEN^{f/f}.

- It appears that additional studies were performed to investigate the impact of optoBBTB on junctional proteins and the authors conclude that there was no impact. This result should be qualified in the discussion to note that this was assessed at one time point, 30 minutes after optoBBB, which may not be enough time to see protein changes depending on the mechanism of action. The authors' claim that the BBB opening is reversible and does not cause cell damage needs to be toned down.

Response: We thank the reviewer for the suggestion. The discussion has been updated to include this information.

Revision:

Page 18, discussion, updated the text.

- (a) Original: First, our AuNP-BV11 nanoparticles binds to the vasculature and optoBBB modulation approach involves the increase of paracellular permeability without causing cell damage.
- (b) Revised: First, our AuNP-BV11 nanoparticles bind to the vasculature, and the optoBBB modulation approach involves the increase of paracellular permeability without causing damage to the blood vessels and junctional proteins under the conditions investigated in this work. It is worth noticing that these changes were assessed 30 minutes after optoBBTB, and further examination should be conducted at longer time points to obtain a comprehensive assessment.

- A section in the discussion on limitations of this technology should be more explicit – lines 565-589 are vague and wordy and It is better to address head on that this technology is currently limited to superficial tumors. The discussion of spinal cord does not seem particularly relevant.

Response: We thank the reviewer for this point. In the revised manuscript, we addressed the current technology limited to superficial tumors, and the discussion of the spinal cord has been removed.

Revision:

Page 20, updated the text.

- (a) Original: Among the various methods to change the BBTB permeability, optoBBTB presents unique opportunities for further preclinical and clinical investigations. In preclinical settings, our method can be useful as a drug development and screening platform (optoBBTB and GEMMs) for testing a class of potent anticancer drugs. In terms of clinical investigations, there are several opportunities for further development. First, the 532 nm light exploited in the current study enables light delivery to the mouse cortex and therefore, the treatment of cortically-located tumors. However, to treat brain tumor in the deep brain region, near-infrared light-absorbing nanoparticles can be exploited since the light in this region exhibits deeper tissue penetration to cover the tumor margin in a larger animal model. Second, while light can be delivered transcranially in the mouse brain, fiber delivery to the human brain is envisioned, especially after the surgical removal of the primary tumor. Placing an optical fiber in the tumor surgical cavity would allow the delivery of side-emitting light to the tumor margin far from the tumor mass (Fig. S13). Moreover, since human skulls are significantly thicker than mice skulls, extracranial light delivery is within the realm of implementation with a transparent cranial window to replace a portion of the skull (63). Furthermore, our recent work investigated opening the blood-spinal cord which represent another important application of the technology and has advantages compared with the state-of-the-art methods (64). Further work is ongoing to investigate tumor treatment in this area.
- (b) Revised: Among the various methods to change the BBTB permeability, optoBBTB presents unique opportunities for further investigations. The 532 nm light exploited in the current study enables light delivery to the mouse cortex, and therefore, our method can be useful as a drug development and screening platform (optoBBTB and GEMMs) for testing a class of potent anticancer drugs for superficial tumors. Several approaches can be exploited to further advance the technique, such as utilizing near-infrared laser and near-infrared light absorbing nanoparticles to improve the light penetration depth in the tissue, or using optical fiber in the tumor surgical cavity for light delivery into deeper brain regions.

- In regards to potential mechanisms, the authors' addition in the discussion of new information they recently published elsewhere (547-557) using an in vitro model does not add significantly to this paper – unless they investigate these mechanisms in the current dataset –and could be shortened.

Response: We thank the reviewer for this suggestion. The discussion of mechanisms has been shortened.

Revision:

Page 19, updated text.

- (a) Original: Recently, we unveiled the mechanisms of the optical BBB modulation using an in vitro BBB model established with human cerebral microvascular endothelial cells. We showed that the picosecond laser excitation of vascular-targeting AuNPs produced nanoscale mechanical perturbation, which triggers several mechanobiological responses, including (1) actin polymerization that leads to the cytoskeletal contraction, (2) Ca²⁺-influx including from mechanosensitive ion channels (such as TRPV4 and Piezo1), and (3) the activation of Inositol Trisphosphate (IP3) pathway and the Ca²⁺ release from the endoplasmic reticulum. The elevation of Ca²⁺ from the extracellular influx and intracellular IP3 pathway activates ERK1/2

phosphorylation. These effects led to a mechanobiological modulation of the BBB and increased paracellular permeability (35).

- (b) Revised: Recently, we revealed the mechanisms of the optical BBB modulation using an in vitro BBB model established with human cerebral microvascular endothelial cells. We showed that the picosecond laser excitation of vascular-targeting AuNPs produced nanoscale mechanical perturbation, which triggers several mechanobiological responses that lead to increased paracellular permeability (35). While it may contribute to the increased drug accumulation in tumors after optoBBTB, further work is warranted to investigate the BBTB opening mechanisms in the current models.

Minor points:

- There is inconsistency in labeling of figure panels, particularly in regards to the box-and-whisker plots which are sometimes labeled as a separate panels but often not, making it difficult for the reader to quickly determine what each panel is referring to in the caption.

Response: We thank the reviewer for this comment. The figures and legends have been updated in the revised manuscript. In the grouped box-and-whisker plots, each color has been labeled with the group name.

- Figure 5c and 6c– what do the gray versus green bars refer to? They do not appear to be labeled in the figure or discussed in the legend. In figure 5f and 6f, the color for vehicle and taxol are inconsistent – was this purposeful? It will aid readers to be consistent in labeling each group when they are comparable.

Response: We thank the reviewer for the question. Figures 5c and 6c are quantification of 5b and 6b (Taxol leakage) into tumor core (black) and tumor margin (blue) by area fraction. We updated the figure legends to describe these panels. The colors for vehicle and taxol in Figures 5f and 6f are revised to be consistent.

Revision:

- A. Figure 5c and 6c, updated the legend.

Fig. 5 and 6. b, c OptoBBTB facilitates the delivery of fluorescent Taxol646 to the tumor core and margin. The scale bar represents 20 μm . The quantification of Taxol delivery was performed by analyzing fluorescent area fraction. For each group, N=10 images from 3 mice. Data are expressed as Mean \pm SD.

- B. Figure 5f and 6f, updated.

Fig. 5f, 6f: Kaplan-Meier survival analysis in PS5A1 GEMM (5f) and 73C GEMM (6f), respectively. N=7 mice in each group. Data were analyzed by logrank test.

- Line 132-135 – The rationale for using immunodeficient mice is not strong, as it does not appear the authors making direct comparisons with their data to other GBM treatment studies? The choice of nude mice does not invalidate the findings of the study, but may warrant mentioning in the discussion as a limitation to extending these results to immunocompetent models without further study. There is mention in the discussion (line 547) noting the lack of immune response in this model which should be qualified further.

Response: We thank the reviewer for this comment. We updated the discussion to include these points.

Revision:

A. Page 16, discussion, updated the text.

- (a) Original: We demonstrated that optoBBTB significantly increased Taxol delivery to both GEMMs at doses that significantly suppressed tumor growth by reducing tumor cell proliferation and inducing cell death, which further prolonged the survival of tumor-bearing mice without causing adverse effects. These results demonstrate that optoBBTB is effective for drug delivery and GBM treatment in two preclinical GEMMs.
- (b) Revised: We demonstrated that optoBBTB significantly increased Taxol delivery to both GEMMs at doses that significantly suppressed tumor growth by reducing tumor cell proliferation and inducing cell death, which further prolonged the survival of tumor-bearing mice without causing adverse effects. These results demonstrate that optoBBTB is effective for drug delivery and GBM treatment in two preclinical GEMMs. One limitation of our current study is that we tested the treatment using GEMMs in immunodeficient mice. It is important to further assess the efficacy of optoBBTB in immunocompetent models.

B. Page 19, discussion, updated the text.

- (a) Original: since our control group (optoBBTB+Vehicle) did not show a reduction of tumor volume or an increased survival rate, and the local temperature was not enhanced after laser excitation, we believe that the reduced tumor volume is mainly mediated by the extravasation of Taxol rather than by other mechanisms such as the activation of the immune system by the heating effect.
- (b) Revised: since our control group (optoBBTB+Vehicle) did not show a reduction of tumor volume or an increased survival rate, and the local temperature was not enhanced after laser excitation, we believe that the reduced tumor volume is mainly mediated by the extravasation of Taxol rather than by other mechanisms such as the heating effect.

- Lines 252-255 – The addition of temperature acquisition is important, and a welcomed addition, but it is not clear how this was measured – is the camera able to detect temperature changes at the level of skin, bone, or underlying brain tissue?

Response: We appreciate the comment. The infrared thermometer (FLIR thermal camera) detects thermal radiation to identify the surface temperature of objects. Therefore, the camera detects the temperature changes on the mouse's skull. We revised the manuscript to clarify this point.

Revision:

Page 9, result, updated the text.

- (a) Original: To investigate if there is a laser-induced heating effect in the tumor area, we recorded the local temperature change using a FLIR ONE Thermal Imaging Camera before and after optoBBTB.
- (b) Revised: To investigate if there is a laser-induced heating effect in the tumor area, we recorded the local temperature change using a FLIR ONE Thermal Imaging Camera before and after optoBBTB on the mouse's skull.

- Further analysis of the biodistribution of gold NPs in each model is appreciated, but is there a reason that data is presented in different units for each model ($\mu\text{g/g}$ tissue versus %ID?)

Response: We thank the reviewer for this comment. We updated the figures to use $\mu\text{g/g}$ to present the data.

Revision:

Fig. S8a, updated.

Fig. S8a. The biodistribution of AuNP-BV11 in 73C GEMM at 7 days post injection (dpi). The AuNP-BV11 injection dose was $37 \mu\text{g/g}$. $N=3$ mice, data are expressed as Mean \pm SD. Data were analyzed by unpaired Student's two-sided t test.

- Line 548: there appears to be a typo "unrevealed"

Response: We thank the reviewer for pointing it out. The typo has been corrected.

Reviewer #4 (Remarks to the Author):

The paper entitled "Optical Blood-Brain-Tumor Barrier Modulation Enhances Drug Penetration and Therapeutic Outcome in Clinically Relevant infiltrative and Angiogenic Glioblastoma Models" is an interesting paper which describes the modulation of the blood brain barrier using plasmonic carriers to facilitate the extravasation of anticancer drugs. The authors have done a relevant effort to address the points raised in a previous round of revisions. There are still relevant cavities in the current version related with the BBB opening mechanism by the formulation and whether the concentration of taxol that accumulates in the tumor (ng of taxol per g of animal) is indeed relevant for a biological action.

Response: We thank the reviewer for their comments. The BBB opening mechanism was investigated in vitro using human cerebral microvascular endothelial cells, while it may contribute to the increased drug

delivery to the tumor after optoBBTB, future work is needed to investigate the mechanism of optoBBTB in the current models. We revised the discussion to include these points.

The measured Taxol concentration in tumor is 185 ng/g and 1206 ng/g for PS5A1 GEMM and for 73C GEMM, which is approximately ng/ml if assume 1 g of tissue contains >90% water. The molecular weight of Taxol is 854 g/mol. Therefore, the Taxol concentration after optoBBTB is around 200 nM (PS5A1) and 1400 nM (73C GEMM), which surpassed the IC50 for PS5A1 GEMM (7 nM) and 73C GEMM (11 nM).

Revision:

Page 19, updated text.

- (a) Original: Recently, we unrevealed the mechanisms of the optical BBB modulation using an in vitro BBB model established with human cerebral microvascular endothelial cells. We showed that the picosecond laser excitation of vascular-targeting AuNPs produced nanoscale mechanical perturbation, which triggers several mechanobiological responses, including (1) actin polymerization that leads to the cytoskeletal contraction, (2) Ca²⁺-influx including from mechanosensitive ion channels (such as TRPV4 and Piezo1), and (3) the activation of Inositol Trisphosphate (IP3) pathway and the Ca²⁺ release from the endoplasmic reticulum. The elevation of Ca²⁺ from the extracellular influx and intracellular IP3 pathway activates ERK1/2 phosphorylation. These effects led to a mechanobiological modulation of the BBB and increased paracellular permeability (35).
- (b) Revised: Recently, we **revealed** the mechanisms of the optical BBB modulation using an in vitro BBB model established with human cerebral microvascular endothelial cells. **We showed that the picosecond laser excitation of vascular-targeting AuNPs produced nanoscale mechanical perturbation, which triggers several mechanobiological responses that lead to increased paracellular permeability (35). While it may contribute to the increased drug accumulation in tumors after optoBBTB, further work is warranted to investigate the BBTB opening mechanisms in the current models.**